# A Single-Loop Stochastic Gradient Algorithm for Minimax Optimization with Nonlinear Coupled Constraints

## Abstract

In this paper, we propose a single-loop stochastic gradient algorithm for solving stochastic nonconvex-concave minimax optimization with nonlinear convex coupled constraints (MCC). The proposed method, SPACO (Stochastic Penalty-based Algorithm for minimax optimization with COupled constraints), is built upon a penalty-based smooth approximation framework for MCC. This framework integrates a quadratic penalty scheme with regularization to yield a continuously differentiable approximation of the MCC problem. We provide theoretical convergence guarantees for this smoothing framework. Furthermore, we establish non-asymptotic complexity bounds and provide an asymptotic analysis characterizing the stationarity of accumulation points for the iterates generated by SPACO. Experimental results on synthetic examples and practical machine learning tasks demonstrate the effectiveness and efficiency of the proposed method.

## 1. Introduction

In this paper, we consider stochastic minimax optimization with coupled constraints (MCC), formulated as

$$\min_{x \in X} \max_{y \in Y} \{f(x,y) \mid c(x,y) \le 0\}, \tag{1}$$

where $f(x,y) := \mathbb{E}_{\xi \sim D}\big[F(x,y;\xi)\big]$ is a stochastic objective defined by the expectation over a distribution $D$, and $F(x,y;\xi)$ denotes a stochastic realization associated with a random sample $\xi \sim D$. Here, $X \subset \mathbb{R}^n$ and $Y \subset \mathbb{R}^m$ are nonempty, convex, and compact sets. The coupled constraint function $c : X \times Y \to \mathbb{R}^p$ is continuously differentiable and potentially nonlinear. We assume that $f(x,y)$ is concave in $y$ and that $c(x,y)$ is convex in $y$.

[1]Anonymous Institution, Anonymous City, Anonymous Region, Anonymous Country. Correspondence to: Anonymous Author <anon.email@domain.com>.

Preliminary work. Under review by the International Conference on Machine Learning (ICML). Do not distribute.

When the coupled constraint $c(x,y) \le 0$ is absent, problem (1) reduces to the well-studied classical minimax optimization problem, which has been extensively studied due to its broad applicability in many applications including robust optimization (Patel et al., 2024; Wang et al., 2024), adversarial learning (Madry et al., 2018; Zhang et al., 2019), and reinforcement learning (Pinto et al., 2017; Zhang et al., 2020a). Consequently, numerous algorithms have been developed for solving these minimax problems in both deterministic (Yoon & Ryu, 2025; Lee et al., 2024; Li et al., 2025) and stochastic settings (Lin et al., 2020a; Zhang et al., 2025; Xian et al., 2024).

Despite this substantial progress, the development of algorithms for minimax optimization with coupled constraints, i.e., MCC (1), remains nascent. The inclusion of coupled constraints provides a more powerful modeling framework for a wide range of challenging applications, including constrained generative adversarial network (GAN) training (Chao et al., 2021), perceptual adversarial robustness (Laidlaw et al., 2021), adversarial attacks in network flow problems (Tsaknakis et al., 2023), and linear projection equations (Dai et al., 2024). However, from a computational perspective, coupled constraints introduce significant complexity. Even when the objective is strongly convex–strongly concave and the coupled constraints are linear, solving the MCC problem is NP-hard (Tsaknakis et al., 2023).

Recently, several algorithms have been proposed for minimax optimization with linear coupled constraints (Tsaknakis et al., 2023; Dai et al., 2024; Zhang et al., 2024; Zhang & Xu, 2024); however, extending these methods to nonlinear constraints is not straightforward. More recently, (Lu & Mei, 2025; Hu et al., 2024) investigated deterministic minimax optimization with nonlinear coupled constraints, proposing effective algorithms based on dual multiplier techniques. Nevertheless, to the best of our knowledge, the literature on *stochastic* minimax optimization with nonlinear coupled constraints remains very limited. Since stochastic formulations are essential in many machine learning applications to model large-scale data and uncertainty (Chao et al., 2021; Laidlaw et al., 2021), scalable and computationally efficient stochastic algorithms are of practical importance. This naturally leads to the following question:

**Can one design an efficient stochastic gradient-type algorithm for stochastic minimax optimization with nonlinear coupled constraints?**

Most existing numerical approaches for solving deterministic MCC rely on duality-based techniques. In particular, problem (1) can be reformulated via the value function

$$\min_{x \in X} \varphi(x), \quad \text{where} \quad \varphi(x) := \max_{y \in Y}\{f(x,y) \mid c(x,y) \leq 0\}.$$

Observe that the value function $\varphi(x)$ is defined by the optimal value of a constrained inner subproblem. By introducing Lagrange multipliers for the coupled constraints and assuming strong duality holds for this inner problem, the value function admits a representation relying on the Lagrangian function:

$$\varphi(x) = \max_{y \in Y} \min_{\lambda \in \mathbb{R}^p_+} L(x,y,\lambda) = \min_{\lambda \in \mathbb{R}^p_+} \max_{y \in Y} L(x,y,\lambda),$$

where $L(x,y,\lambda) := f(x,y) - \lambda^\top c(x,y)$ is the Lagrangian for the inner maximization subproblem. This leads to a min–min–max reformulation of the MCC:

$$\min_{x \in X, \lambda \in \mathbb{R}^p_+} \max_{y \in Y} L(x,y,\lambda), \tag{2}$$

where the multiplier $\lambda$ is introduced as an additional optimization variable. A key advantage of this reformulation is that it allows for the application of various existing algorithmic techniques designed for min-max optimization. In particular, several gradient-based descent–ascent methods have been proposed and well-studied in this context; see, e.g., (Tsaknakis et al., 2023; Hu et al., 2024; Zhang et al., 2024; Zhang & Xu, 2024).

In this work, in contrast to existing algorithmic studies based on the min–min–max reformulation, we propose a fundamentally different approach to handling the coupled constraints. Beyond Lagrangian multiplier methods, penalty methods are another classical and powerful technique for constrained optimization (Nocedal & Wright, 2006; Bertsekas, 1999). Here, we utilize a classical quadratic penalty for the constraints to construct a smooth approximation of the value function $\varphi(x)$. This yields a sequence of smooth approximation problems for the MCC, which constitutes our proposed Penalty-based Smooth Approximation Framework.

This framework enables the application of efficient gradient-based optimization methods. However, while this smooth approximation enjoys favorable differentiability properties, computing its gradient requires solving an inner maximization problem. This is computationally expensive, particularly when the objective is stochastic. To overcome this challenge, we propose a single-step stochastic gradient ascent procedure to obtain an inexact solution to the inner

problem, followed by an inexact gradient descent update on $x$. Building on this strategy, we develop a single-loop stochastic gradient algorithm for solving MCC: the Stochastic Penalty-based Algorithm for minimax optimization with COupled constraints (SPACO).

Additionally, through a simple illustrative example in Section 4, we provide intuition demonstrating that the min–min–max reformulation may be trapped in spurious stationary points that do not correspond to local optima of the original MCC. In contrast, our proposed penalty-based approximation approach helps avoid such undesired solutions.

### 1.1. Contribution

This paper makes the following contributions to the study of algorithms for solving MCC:

**Penalty-based Smooth Approximation Framework for MCC.** We propose a novel penalty-based smooth approximation framework for the MCC. By incorporating constraints into the objective via a quadratic penalty term and employing a regularization technique, we derive a smooth approximation of the MCC value function and construct a sequence of smooth approximation problems. We validate this framework by proving that any accumulation point of the minimizers of the smooth approximation is a minimizer of the original MCC. Furthermore, we establish that any accumulation point of the stationary points of the smooth approximation is a stationary point of the original MCC.

**Single-Loop Stochastic Gradient Algorithm with Convergence Analysis.** Building on the proposed framework, we develop a single-loop stochastic gradient algorithm, SPACO, for solving the MCC. The algorithm is simple to implement, avoids nested optimization loops, and does not require exact subproblem solutions. We provide a rigorous convergence analysis for SPACO, establishing non-asymptotic convergence rates. Additionally, under suitable constraint qualification conditions, we give an asymptotic analysis that characterizes the stationarity of accumulation points of the iterates generated by SPACO.

**Empirical Validation.** We empirically validate the effectiveness and efficiency of SPACO on both synthetic examples and real-world applications, including generative adversarial networks and fairness-aware classification. The experimental results consistently demonstrate the superior performance of the proposed SPACO.

### 1.2. Related Work

We provide a brief review of works closely related to ours, with a comprehensive review deferred to Appendix A.

From a theoretical perspective, optimality and stationarity conditions for MCC have been investigated by (Dai &

Zhang, 2020; Guo et al., 2024). On the algorithmic side, existing approaches can be categorized based on the nature of the constraints. For minimax optimization with linear coupled constraints, (Dai et al., 2024) proposed an alternating coordinate method. Additionally, (Tsaknakis et al., 2023) introduced the min–min–max reformulation and designed a multiplier gradient descent algorithm. Building on this reformulation, (Zhang & Xu, 2024; Zhang et al., 2024) developed a class of efficient single-loop primal–dual gradient methods for both deterministic and stochastic settings. For deterministic MCC with nonlinear coupled constraints, (Goktas & Greenwald, 2021) proposed max-oracle and nested gradient descent methods. (Dai & Zhang, 2024) proposed an augmented Lagrangian method for minimax optimization with equality constraints and analyzed the convergence rate. (Lu & Mei, 2025) first studied MCC with non-smooth objective and nonlinear inequality constraints and proposed a new first-order algorithm based on the augmented Lagrangian framework. More recently, (Hu et al., 2024) reformulated MCC as a minimization optimization problem by using Moreau envelope techniques and proposed a novel subgraident method.

Despite these advances, to the best of our knowledge, algorithmic development for stochastic MCC with nonlinear coupled constraints, particularly efficient single-loop stochastic gradient algorithms, remains limited.

## 2. Penalty-based Smooth Approximation

As discussed previously, solving MCC can be reformulated as minimizing the value function:

$$\min_{x \in X} \varphi(x), \text{ where } \varphi(x) := \max_{y \in Y} \{f(x,y) \mid c(x,y) \le 0\}.$$

Although this reformulation is concise, a major challenge in minimizing $\varphi(x)$ lies in its potential nonsmoothness. This nonsmoothness arises from both the coupled constraints $c(x,y) \le 0$ and the potential non-uniqueness of the inner maximizer in the inner maximization problem defining $\varphi(x)$ (see, e.g., (Guo et al., 2024)).

To address this, we propose a penalty-based smooth approximation framework for the MCC. The proofs for all results in this section are provided in Appendix C. Throughout the paper, we adopt the following standing assumptions.

**Assumption 2.1.** The following conditions hold:

1. $X$ and $Y$ are nonempty, convex, and compact sets. The function $f(x,y)$ is continuously differentiable with a Lipschitz continuous gradient on $X \times Y$. For any fixed $x \in X$, $f(x,y)$ is concave with respect to $y$ on $Y$.

2. Each component $c_i(x,y)$ of $c(x,y)$ is continuously differentiable on $X \times Y$ with a Lipschitz continuous gradient. For any fixed $x \in X$, the mapping $c_i(x, \cdot)$ is con-

vex on $Y$. Moreover, for every $x \in X$, the feasible set $\{y : c(x,y) \le 0\} \cap Y$ is nonempty.

### 2.1. Smooth Approximation Construction

To handle the nonsmoothness caused by the nonlinear coupled constraints, we employ a classical quadratic penalty strategy to approximate the inner maximization. This leads to the following approximation of the value function:

$$\max_{y \in Y} \left\{ f(x,y) - \frac{\rho}{2} \left\| [c(x,y)]_+ \right\|^2 \right\},$$

where $\rho > 0$ is a penalty parameter and $[\cdot]_+ = \max\{\cdot, 0\}$ is applied component-wise. While the quadratic penalty smooths the effect of the coupled constraints, the value function may still be nonsmooth due to the potential non-uniqueness of the maximizer of the inner problem. To overcome this issue, we further introduce a quadratic regularization term in the inner maximization problem. This leads to the following regularized penalized objective:

$$\psi_{\rho,\sigma}(x,y) := f(x,y) - \frac{\rho}{2} \left\| [c(x,y)]_+ \right\|^2 - \frac{\sigma}{2} \|y\|^2,$$

where $\sigma > 0$ is a regularization parameter. This forms the core of our proposed smooth approximation of $\varphi(x)$, defined as:

$$\varphi_{\rho,\sigma}(x) := \max_{y \in Y} \psi_{\rho,\sigma}(x,y). \tag{3}$$

A key property of $\varphi_{\rho,\sigma}(x)$ is its smoothness, as established below.

**Proposition 2.2.** *For any constant $\rho > 0$ and $\sigma > 0$, the function $\varphi_{\rho,\sigma} : X \to \mathbb{R}$ is differentiable. Its gradient is given by:*

$$\nabla \varphi_{\rho,\sigma}(x) = \nabla_x \psi_{\rho,\sigma}\big(x, y^*_{\rho,\sigma}(x)\big), \tag{4}$$

*where $y^*_{\rho,\sigma}(x)$ denotes the unique maximizer of the inner problem, i.e., $y^*_{\rho,\sigma}(x) := \arg\max_{y \in Y} \psi_{\rho,\sigma}(x,y)$.*

### 2.2. Convergence of Approximation

The proposed penalty-based smooth approximation framework allows us to solve the MCC (1) approximately by considering a sequence of smooth problems of the form

$$\min_{x \in X} \varphi_{\rho_k,\sigma_k}(x), \tag{5}$$

with parameter sequences $\rho_k \to \infty$ and $\sigma_k \to 0$. In this part, we justify the validity of this approximation from two distinct perspectives: (i) the asymptotic convergence of global minimizers and (ii) the asymptotic convergence of stationary points.

We first analyze the convergence of global minimizers. Our analysis relies on the theory of epi-convergence, specifically the result that the epi-convergence of a sequence of

functions implies the convergence of their minimizers (see, e.g., (Bonnans & Shapiro, 2013, Proposition 4.6)). Accordingly, we first establish two auxiliary lemmas concerning the epi-convergence of $\varphi_{\rho_k, \sigma_k}$.

**Lemma 2.3.** *Let $\rho_k \to \infty$ and $\sigma_k \to 0$ as $k \to \infty$. Then, for any fixed $x \in X$,*

$$\limsup_{k \to \infty} \varphi_{\rho_k, \sigma_k}(x) \leq \varphi(x).$$

To obtain a matching lower bound, we require the following continuity assumption on the value function.

**Assumption 2.4.** The value function $\varphi : X \to \mathbb{R}$ is lower semi-continuous on $X$, i.e., for any $\bar{x} \in X$ and any sequence $\{x_k\} \subset X$ converging to $\bar{x}$, $\liminf_{k \to \infty}, \varphi(x_k) \geq \varphi(\bar{x})$.

Lower semi-continuity is crucial for ensuring the existence of minimizers of a function over compact sets (see, e.g., (Rockafellar & Wets, 2009, Theorem 1.9)). The inner semi-continuity of the feasible-set mapping $\Theta(x) := \{y \in Y : c(x, y) \leq 0\}$ provides a sufficient condition for the lower semi-continuity of $\varphi$ (see Lemma C.3).

**Lemma 2.5.** *Suppose $\varphi$ is lower semi-continuous on $X$. Let $\rho_k \to \infty$ and $\sigma_k \to 0$ as $k \to \infty$. Then, for any sequence $\{x_k\} \subset X$ with $x_k \to \bar{x}$, we have:*

$$\liminf_{k \to \infty} \varphi_{\rho_k, \sigma_k}(x_k) \geq \varphi(\bar{x}). \tag{6}$$

Combining Lemma 2.3 and 2.5, we obtain the epi-convergence of $\varphi_{\rho_k, \sigma_k}$ to $\varphi$, which yields the convergence of global minimizers.

**Theorem 2.6.** *Assume $\varphi(x)$ is lower semi-continuous on $X$. Let $\rho_k \to \infty$ and $\sigma_k \to 0$ as $k \to \infty$, and let $x_k \in \arg\min_{x \in X} \varphi_{\rho_k, \sigma_k}(x)$. Then, any accumulation point $\bar{x}$ of the sequence $\{x_k\}$ is an optimal solution to the MCC (1), i.e., $\bar{x} \in \arg\min_{x \in X} \varphi(x)$. Moreover, any accumulation point $\bar{y}$ of the sequence $\{y^*_{\rho_k, \sigma_k}(x_k)\}$ satisfies $\arg\max_{y \in Y}\{f(\bar{x}, y) \mid c(\bar{x}, y) \leq 0\}$.*

We next analyze the asymptotic convergence of stationary points. Since the approximation problem (5) is smooth, gradient-based algorithms can be applied to compute its solutions. However, because $\varphi_{\rho_k, \sigma_k}$ is generally nonconvex, such methods typically converge only to stationary points. It is therefore important to understand the relationship between the stationary points of the smooth approximations and those of the original MCC problem (1).

We consider the stationary (KKT) points for the MCC problem (1) as defined below (see, e.g., (Lu & Mei, 2025, Definition 2) and (Hu et al., 2024) for further discussion).

**Definition 2.7.** A pair $(x, y)$ is called a stationary (or KKT) point of MCC (1) if there exist a multiplier $\lambda \in \mathbb{R}^p_+$ such

that the following conditions hold:

$$\begin{cases} 0 \in \nabla_x L(x, y, \lambda) + \mathcal{N}_X(x), \\ 0 \in -\nabla_y L(x, y, \lambda) + \mathcal{N}_Y(y), \\ c(x, y) \leq 0, \ \lambda^\top c(x, y) = 0, \end{cases} \tag{7}$$

where $L(x, y, \lambda) := f(x, y) - \lambda^\top c(x, y)$ is the Lagrangian for the inner maximization subproblem, and $\mathcal{N}_C(\cdot)$ denotes the normal cone to a convex set $C$. Specifically, for $z \in C$, the normal cone is given by $\mathcal{N}_C(z) = \{v : \langle v, z' - z \rangle \leq 0, \forall z' \in C\}$, and $\mathcal{N}_C(z) = \emptyset$ if $z \notin C$.

For the coupled constraints, we introduce the following constraint qualification. This is a generalized version of the Polyak-Łojasiewicz Constraint Qualification (PŁCQ) (see, e.g., (Andreani et al., 2025)), extended to be uniform with respect to the variable $x$. This extension shares the spirit of the consistent Slater condition used in (Lu & Mei, 2025) and the Stable parametric CQ used in (Guo et al., 2024).

**Definition 2.8.** We say that the Generalized uniform Polyak-Łojasiewicz Constraint Qualification (GPŁCQ) holds at $(\bar{x}, \bar{y})$ if there exist constants $\delta > 0$, $\beta > 0$ and $\gamma > 0$ such that for all $x \in B_\delta(\bar{x}) \cap X$ and $y \in B_\delta(\bar{y}) \cap Y$ (where $B_\delta(\bar{z})$ denotes the closed ball centered at $\bar{z}$ with radius $\delta$), the following inequality holds:

$$\sqrt{p(x, y)} \leq \frac{\gamma}{\beta} \left\| \begin{bmatrix} x \\ y \end{bmatrix} - \mathcal{P}_{X \times Y}\left( \begin{bmatrix} x \\ y \end{bmatrix} - \beta \begin{bmatrix} \nabla_x p(x, y) \\ -\nabla_y p(x, y) \end{bmatrix} \right) \right\|,$$

where $p(x, y) := \frac{1}{2}\|[c(x, y)]_+\|^2$, and $\mathcal{P}_{X \times Y}$ denotes the Euclidean projection onto $X \times Y$.

The consistent Slater condition employed in (Lu & Mei, 2025) implies the GPŁCQ when $X$ and $Y$ are the whole space (see Lemma C.4). With this qualification, we can now characterize the limit points of stationary points of the approximation problem (5).

**Theorem 2.9.** *Let $\rho_k \to \infty$ and $\sigma_k \to 0$ as $k \to \infty$, and suppose $x_k$ is a stationary point of $\min_{x \in X} \varphi_{\rho_k, \sigma_k}(x)$, i.e., $0 \in \nabla \varphi_{\rho_k, \sigma_k}(x_k) + \mathcal{N}_X(x_k)$. Assume $x_k \to \bar{x}$ and let $\bar{y}$ be an accumulation point of $\{y^*_{\rho_k, \sigma_k}(x_k)\}$. If GPŁCQ holds at $(\bar{x}, \bar{y})$, then $(\bar{x}, \bar{y})$ is a stationary (KKT) point of the original MCC problem (1).*

## 3. Single-Loop Stochastic Gradient Algorithm

In the previous section, we introduced a sequence of smooth problems $\min_{x \in X} \varphi_{\rho_k, \sigma_k}(x)$ to approximate the MCC (1). The smoothness of $\varphi_{\rho_k, \sigma_k}(x)$ facilitates the application of gradient-based optimization methods. Building on this smooth approximation framework, we now develop a practical single-loop stochastic gradient algorithm, named SPACO, to solve the stochastic MCC (1).

### 3.1. Algorithm Description

Although $\varphi_{\rho_k,\sigma_k}(x)$ is differentiable, designing an efficient stochastic gradient algorithm based on this smooth approximation presents nontrivial challenges. First, as demonstrated in Proposition 2.2, computing $\nabla_x \varphi_{\rho_k,\sigma_k}(x)$ requires the exact maximizer $y^*_{\rho_k,\sigma_k}(x)$ of the inner problem, which is generally unavailable in closed form and expensive to obtain, especially in stochastic settings.

To ensure practical implementability, we employ an inexact gradient update for $x$. Specifically, the iterate $y^k$ serves as an approximation of the exact maximizer $y^*_{\rho_k,\sigma_k}(x^k)$. At each iteration $k$, given the current iterate $(x^k, y^k)$ and parameters $(\rho_k, \sigma_k)$, we first perform a one-step stochastic projected gradient ascent update on the inner maximization problem (3). Specifically, we draw an independent sample $\xi_k^y \sim D$ and compute a stochastic estimate of $\nabla_y \psi_{\rho_k,\sigma_k}(x^k, y^k)$:

$$d_y^k = \nabla_y \Psi_k(x^k, y^k; \xi_k^y), \qquad (8)$$

where

$$\Psi_k(x, y; \xi) := F(x, y; \xi) - \frac{\rho_k}{2}\|[c(x,y)]_+\|^2 - \frac{\sigma_k}{2}\|y\|^2,$$

denotes the stochastic estimator of $\psi_{\rho_k,\sigma_k}(x, y)$ constructed from a random sample $\xi$. The variable $y$ is then updated via a projected gradient ascent step

$$y^{k+1} = \mathcal{P}_Y\left(y^k + \beta_k d_y^k\right),$$

where $\beta_k > 0$ is the stepsize.

Next, we use $y^{k+1}$ to approximate $y^*_{\rho_k,\sigma_k}(x^k)$ within the gradient formula (4) of $\nabla \varphi_{\rho_k,\sigma_k}(x^k)$ to construct an approximate gradient for the $x$-update. Since the variable $x$ is constrained to the set $X$, directly applying stochastic projected gradient methods may result in oscillatory behavior due to stochastic noise. To mitigate this and stabilize convergence, we incorporate a momentum-based variance reduction technique (Cutkosky & Orabona, 2019). The update direction $d_x^k$ is defined recursively as

$$d_x^k = (1 - \eta_k)\left(d_x^{k-1} - \nabla_x \Psi_{k-1}(x^{k-1}, y^k; \xi_k^x)\right) + \nabla_x \Psi_k(x^k, y^{k+1}; \xi_k^x), \qquad k \geq 1, \qquad (9)$$

where $\eta_k \in (0, 1]$ is the momentum parameter, $\xi_k^x \sim D$ is a sample drawn independently from $\xi_k^y$, and we initialize $d_x^0 = \nabla_x \Psi_0(x^0, y^1; \xi_0^x)$. The variable $x$ is then updated via a projected gradient descent step:

$$x^{k+1} = \mathcal{P}_X\left(x^k - \alpha_k d_x^k\right),$$

where $\alpha_k > 0$ is the stepsize.

As discussed in the previous section, ensuring that the approximated problem $\min_{x \in X} \varphi_{\rho_k,\sigma_k}(x)$ provides a faithful approximation to the original MCC (1) requires the

parameters to satisfy $\rho_k \to \infty$ and $\sigma_k \to 0$ as $k \to \infty$. Consequently, the geometric properties (e.g., curvature) of $\varphi_{\rho_k,\sigma_k}(x)$ evolve across iterations, necessitating a carefully designed strategy for selecting the stepsizes $\alpha_k, \beta_k$ and momentum parameters $\eta_k$. We provide a concrete selection strategy and the corresponding convergence analysis in the subsequent subsection.

Integrating the above inexact gradient and momentum strategies, we present the Stochastic Penalty-based Algorithm for minimax optimization with Coupled constraints (SPACO), summarized in Algorithm 1.

---

**Algorithm 1** **S**tochastic **P**enalty-based **A**lgorithm for minimax optimization with **CO**upled constraints (SPACO)

---

**Input:** Initial points $(x^0, y^0) \in X \times Y$ and, penalty parameters $\{\rho_k\}$, regularization parameters $\{\sigma_k\}$, stepsizes $\alpha_k, \beta_k > 0$ and momentum parameters $\{\eta_k\}$.

**for** $k = 0, 1, \ldots$ **do**

    Sample $\xi_k^y \sim D$ , compute direction $d_y^k$ via (8) and update:

$$y^{k+1} = \mathcal{P}_Y(y^k + \beta_k d_y^k).$$

    Sample $\xi_k^x \sim D$ (independent of $\xi_k^y$), compute direction $d_x^k$ via (9) and update:

$$x^{k+1} = \mathcal{P}_X(x^k - \alpha_k d_x^k).$$

---

### 3.2. Convergence Analysis

In this part, we provide a non-asymptotic complexity analysis for the proposed SPACO and an asymptotic analysis characterizing the stationarity of the accumulation points of the generated iterates. Detailed proofs are deferred to Appendix D.

To simplify notation, throughout this section we denote $\psi_k(x, y) := \psi_{\rho_k,\sigma_k}(x, y)$, $\varphi_k(x) := \varphi_{\rho_k,\sigma_k}(x)$, and $y_k^*(x) := y^*_{\rho_k,\sigma_k}(x)$. To formalize the stochasticity, let $\mathcal{F}_k$ denote the $\sigma$-algebras generated by the samples up to step $k$, and $\mathcal{F}_{k+\frac{1}{2}}$ denote the $\sigma$-algebras generated by $\mathcal{F}_k$ and the sample $\xi_k^y$:

$$\mathcal{F}_k = \sigma\{\xi_0^y, \xi_0^x, \ldots, \xi_{k-1}^y, \xi_{k-1}^x\}, \quad \mathcal{F}_{k+\frac{1}{2}} = \sigma\{\mathcal{F}_k, \xi_k^y\}.$$

We next impose standard assumptions on the stochastic oracles employed in SPACO.

**Assumption 3.1.** The stochastic oracles $\nabla_x F(x, y; \xi)$ and $\nabla_y F(x, y; \xi)$ satisfy the following conditions:

1. Unbiasedness: $\xi_k^y$ (resp. $\xi_k^x$) is independent of $\mathcal{F}_k$ (resp. $\mathcal{F}_{k+\frac{1}{2}}$), and

$$\mathbb{E}_{\xi \sim D}[\nabla_x F(x, y; \xi)] = \nabla_x f(x, y),$$
$$\mathbb{E}_{\xi \sim D}[\nabla_y F(x, y; \xi)] = \nabla_y f(x, y).$$

2. Bounded variance: there exists $\delta > 0$ such that

$$\mathbb{E}_{\xi \sim D} \left[ \|\nabla_x F(x, y; \xi) - \nabla_x f(x, y)\|^2 \right] \leq \delta^2,$$
$$\mathbb{E}_{\xi \sim D} \left[ \|\nabla_y F(x, y; \xi) - \nabla_y f(x, y)\|^2 \right] \leq \delta^2.$$

**Assumption 3.2.** The stochastic oracle $\nabla_x F(x, y; \xi)$ allows for simultaneous queries; that is, the algorithm can evaluate unbiased gradient estimators at two distinct points $(x_1, y_1)$ and $(x_2, y_2)$ using the same sample $\xi$. Furthermore, the oracle satisfies the following mean-squared smoothness condition for some constant $L_F > 0$:

$$\mathbb{E}_\xi \left[ \|\nabla_x F(x_1, y_1; \xi) - \nabla_x F(x_2, y_2; \xi)\|^2 \right]$$
$$\leq L_F^2 \left( \|x_1 - x_2\|^2 + \|y_1 - y_2\|^2 \right).$$

We begin with a lemma characterizing the smoothness of $\psi_k(x, y)$ and the boundedness of $\varphi_k(x)$.

**Lemma 3.3.** Let $\rho_k \to \infty$ and $\sigma_k \to 0$ as $k \to \infty$. Then, $\nabla \psi_k(x, y)$ is $\bar{L}_k$-Lipschitz continuous on $X \times Y$ with constant $\bar{L}_k = \mathcal{O}(\rho_k)$. Furthermore, $\varphi_k(x)$ is uniformly lower bounded with respect to $k$ on $X$ by a constant $\underline{\varphi}$.

To analyze the convergence of SPACO, we introduce the following merit function:

$$V_k = a_k(\varphi_k(x^k) - \underline{\varphi}) + b_k\|y^k - y_k^*(x^k)\|^2$$
$$+ c_k\|e_x^{k-1}\|^2 + d_k\|x^k - x^{k-1}\|^2, \tag{10}$$

where $a_k, b_k, c_k, d_k > 0$ are iteration-varying coefficients (specified later), $e_x^k := d_k^x - \nabla_x \psi_k(x^k, y^{k+1})$ is the error term induced by the inexact gradient, and $\underline{\varphi}$ is the uniform lower bound established above.

With suitable parameter choices, we obtain the following descent property.

**Proposition 3.4.** Let $\{(x^k, y^k)\}$ be generated by SPACO (Algorithm 1) with penalty and regualrization parameters selected as: $\sigma_k = \sigma_0 k^{-t}, \rho_k = \rho_0 k^t$, and the stepsizes and momentum parameter selected as:

$$\alpha_k = \alpha_0 k^{-6t-s}, \quad \beta_k = \beta_0 k^{-t-s}, \quad \eta_k = \eta_0 k^{-s}$$

where $\alpha_0, \beta_0, \eta_0, \sigma_0, \rho_0, t, s > 0$. Set the varying coefficients in $V_k$ as $a_k = k^{-2t}, b_k = k^{-3t}, c_k = k^{-7t}, d_k = k^{-4t}$. If $0 < t, s < 1$, $s > 3t$ and $8t + s < 1$, and if $\beta_0/\sigma_0$ is sufficiently small, then for all sufficiently large $k$, the following inequality holds:

$$\mathbb{E}[V_{k+1} \mid \mathcal{F}_k] - V_k$$
$$\leq -\frac{a_k \alpha_k}{24} \mathbb{E}[\|\mathcal{G}_k(x^k)\|^2 \mid \mathcal{F}_k] - \frac{b_k \beta_k \sigma_k}{4} \|y^k - y_k^*(x^k)\|^2$$
$$+ 2b_k \beta_k^2 \delta^2 + \zeta_k.$$

where $\mathcal{G}_k(x) = \frac{1}{\alpha_k}(x - P_X(x - \alpha_k \nabla \varphi_k(x)))$ is the generalized gradient residual for $\min_{x \in X} \varphi_k(x)$ and $\{\zeta_k\}$ is a summable nonnegative sequence.

Using the descent property of $V_k$, we establish the following non-asymptotic convergence rates for the generalized gradient residual $\|\mathcal{G}_k(x_k)\|$ and the tracking error $\|y^k - y_k^*(x^k)\|$.

**Theorem 3.5.** Let $\{(x^k, y^k)\}$ be the sequence generated by SPACO (Algorithm 1) with parameters selected as in Proposition 3.4. Then,

$$\min_{0 \leq k \leq K} \{\mathbb{E}[\|\mathcal{G}_k(x^k)\|^2]\} = \mathcal{O}(\frac{1}{K^{1-8t-s}}) + \mathcal{O}(\frac{1}{K^{s-3t}}),$$

$$\min_{0 \leq k \leq K} \{\mathbb{E}[\|y^k - y_k^*(x^k)\|^2]\} = \mathcal{O}(\frac{1}{K^{1-5t-s}}) + \mathcal{O}(\frac{1}{K^s}).$$

Finally, we characterize the asymptotic stationarity of the accumulation points of the generated sequence.

**Theorem 3.6.** Let $\{(x^k, y^k)\}$ be the sequence generated by SPACO (Algorithm 1) with parameters selected as in Proposition 3.4, and further assume that $2s + 5t > 1$. Then, almost surely, there exists a convergent subsequence $\{(x^{k_j}, y^{k_j})\}$ such that $\|\mathcal{G}_{k_j}(x^{k_j})\| \to 0$ and its limit point $(\bar{x}, \bar{y})$ is a stationary (KKT) point of the MCC (1) defined in (7), provided GPŁCQ holds at $(\bar{x}, \bar{y})$.

# 4. An Example: Escaping Spurious Stationary Points via Penalty-Based Approximation

In this section, we present a toy example to illustrate an advantage of the penalty-based smooth approximation over the min–min–max reformulation of the MCC: its ability to avoid convergence to "spurious" stationary points that do not correspond to local solutions of the original MCC.

We consider the following deterministic MCC:

$$\min_{x \in X} \max_{\substack{y \in Y \\ \mathbf{e}^\top y - \|x\|^2 \leq 0}} \left( \frac{\|x\|^2}{2} - 1 \right)^2 - \frac{\|y - \mathbf{e}\|^2}{2} + \frac{x^\top y}{2}, \tag{11}$$

where $X = [-\frac{3}{4}, \frac{5}{4}]^2 \subset \mathbb{R}^2$, $Y = [-10, 10]^2 \subset \mathbb{R}^2$ and $\mathbf{e} = (1, 1)$. For any fixed $x$, the unique inner maximization admits the closed-form solution $\frac{2\|x\|^2 - \mathbf{e}^\top x}{4}\mathbf{e} + \frac{x}{2}$. Substituting this into the objective yields the value function $\varphi(x) = \frac{\|x\|^2}{8} + \frac{\mathbf{e}^\top x}{4}\left(\|x\|^2 - \frac{\mathbf{e}^\top x}{4}\right)$. Consequently, the unique solution to this problem is $(x^*, y^*) = (-\frac{3}{4}\mathbf{e}, \frac{9}{16}\mathbf{e})$.

Examining the stationary (KKT) conditions defined in (7), we observe that, in addition to the triplet $(x^*, y^*, \lambda^*) = (-\frac{3}{4}\mathbf{e}, \frac{9}{16}\mathbf{e}, \frac{1}{16})$ associated with the true solution, there exists an additional stationary point $(x', y', \lambda') = (\mathbf{0}, \mathbf{0}, 1)$. Since this point is not a local extremum, we classify it as a spurious stationary point.

Next, we numerically evaluate the proposed SPACO against the min–min–max reformulation approach. We assess the convergence limits of the iterates generated by both approaches using multiple initial points, with $x^0$ uniformly

drawn from the grid $[-\frac{3}{4}, \frac{3}{4}] \times [-\frac{3}{4}, \frac{3}{4}]$ and fixed $y^0 = \mathbf{0}$. The min–min–max reformulation approach of (11) results in the following min–max problem:

$$\min_{\substack{x \in X \\ \lambda \in \mathbb{R}_+}} \max_{y \in Y} (\frac{\|x\|^2}{2} - 1)^2 - \frac{\|y - \mathbf{e}\|^2}{2} + \frac{x^\top y}{2} - \lambda(\mathbf{e}^\top y - \|x\|^2).$$

We solve this reformulation using the min–max optimization solver SciPy (Virtanen et al., 2020). The results are presented in Figure 1, and experimental details are provided in Appendix E.1. As illustrated in Figure 1, the min–min–max reformulation converges to the spurious stationary point for a large portion of the initialization space. In contrast, SPACO converges to the true solution from all tested initial points. This suggests that penalty-based approximations may effectively reduce the likelihood of convergence to undesirable stationary points.

We now provide an intuitive explanation for this behavior. Although our convergence results (Theorem 3.6) demonstrate that iterates generated by SPACO asymptotically satisfy the same stationary conditions as the min–min–max reformulation, the optimization landscapes differ. The penalty-based approximation discourages constraint violations through the objective function, thereby improving the geometry of the landscape. This alteration may help iterates escape spurious regions. The following lemma formalizes this intuition for our example.

**Lemma 4.1.** *Consider the penalty-based smooth approximation of (11):*

$$\min_{x \in X} \max_{y \in Y} (\frac{\|x\|^2}{2} - 1)^2 - \frac{\|y - \mathbf{e}\|^2}{2} + \frac{x^\top y}{2} - \rho[\mathbf{e}^\top y - \|x\|^2]_+^2.$$

*For any $\rho = \rho_k > 1$, this approximation admits a unique stationary point $(x_k^*, y_k^*) = \left(-\frac{3}{4}\mathbf{e}, \frac{10 + 18\rho_k}{16 + 32\rho_k}\mathbf{e}\right)$, which converges to $(-\frac{3}{4}\mathbf{e}, \frac{9}{16}\mathbf{e})$ as $\rho_k \to 0$.*

This lemma demonstrates that the penalty-based smooth approximation of (11) generates a sequence of points converging uniquely to the true solution, effectively excluding the spurious stationary point $(\mathbf{0}, \mathbf{0})$ from the approximation path. Additional examples exhibiting similar behavior are provided in Table 5 in the appendix.

# 5. Numerical Experiments

In this section, we empirically evaluate the practical performance of SPACO using a collection of synthetic experiments and real-world applications. Implementation details and experimental configurations for all evaluated methods are provided in Appendix B.

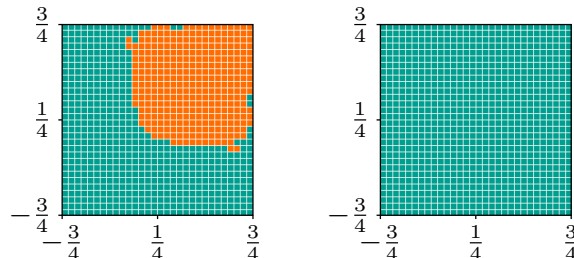

*(a)* Min–min–max reformulation  *(b)* Penalty-based approximation

*Figure 1.* Convergence behavior of the penalty-based approximation approach (SPACO) versus the min–min–max reformulation (solved with SciPy), shown for different initializations $x^0$. Green regions ■ indicate convergence to the true solution, while orange regions ■ indicate convergence to a spurious stationary point.

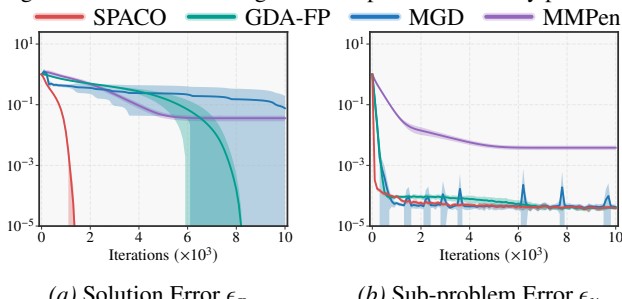

*(a)* Solution Error $\epsilon_x$  *(b)* Sub-problem Error $\epsilon_y$

*Figure 2.* Convergence curve for solving (12).

## 5.1. Synthetic Example

We consider the following stochastic minimax optimization with nonlinear constraint extended by (11):

$$\min_{x \in X} \max_{y \in \Theta(x)} \mathbb{E}_w \left[ \frac{n}{2}(\frac{\|x\|^2}{n} - 1)^2 - \frac{\|y - \mathbf{e}\|^2}{2} + \frac{x^\top(y + w)}{2} \right],$$
(12)

where $X = [-\frac{3}{4}, \frac{5}{4}]^n$, $\Theta(x) = \{y \in [-10, 10]^n \mid \mathbf{e}^\top y - \|x\|^2 \le 0\}$, $w$ is a Gaussian random vector with independent entries $w_i \sim \mathcal{N}(0, \delta^2)$ and $\mathbf{e} \in \mathbb{R}^n$ denotes the all-ones vector. With simple calculation, for a fixed $x$, the expected inner sub-problem solution is given by $y^*(x) = \frac{2\|x\|^2 - \mathbf{e}^\top x}{2n}\mathbf{e} + \frac{x}{2}$ and the whole expected optimal solution $(x^*, y^*)$ is given by $(x^*, y^*) = (-\frac{3}{4}\mathbf{e}, \frac{9}{16}\mathbf{e})$. The performance is assessed by the solution error $\epsilon_x = \frac{\|x - x^*\|^2}{\|x^0 - x^*\|^2 + 1}$ and the sub-problem error $\epsilon_y = \frac{\|y - y^*(x)\|^2}{\|y^0 - y^*(x)\|^2 + 1}$. We fix $n = 100, \delta = 1$ and compare SPACO with MGD (Tsaknakis et al., 2023), MMPen (Hu et al., 2024) and gradient descent-ascent with fixed penalty parameter (GDA-FP). Figure 1 shows the convergence curve for both errors over 10 runs. We additionally test the performance on a example with linear constraint and the robustness of SPACO on hyperparameters in Appendix B.1. Further, We evaluate SPACO's robustness to hyperparameters (stepsizes $\alpha_0, \beta_0$, initial parameter $\rho_0$ and factors $t, s$) by measuring iterations required to achieve the tolerance $\max\{\epsilon_x, \epsilon_y\} \le 10^{-4}$, where the results are summarized in Table 1.

*Table 1.* Ablation analysis for hyperparameter.

| $\alpha_0$ | $\beta_0$ | $\rho_0$ | $(t, s)$ | Iterations |
|---|---|---|---|---|
| 0.1 | 0.1 | 10 | $(0.05, 0.2)$ | $1171_{\pm 52}$ |
| **0.05** | 0.1 | 10 | $(0.05, 0.2)$ | $4299_{\pm 230}$ |
| **0.2** | 0.1 | 10 | $(0.05, 0.2)$ | $452_{\pm 29}$ |
| 0.1 | **0.05** | 10 | $(0.05, 0.2)$ | $1349_{\pm 70}$ |
| 0.1 | **0.2** | 10 | $(0.05, 0.2)$ | $1280_{\pm 118}$ |
| 0.1 | 0.1 | **5** | $(0.05, 0.2)$ | $1194_{\pm 68}$ |
| 0.1 | 0.1 | **20** | $(0.05, 0.2)$ | $1199_{\pm 25}$ |
| 0.1 | 0.1 | 10 | $(\mathbf{0.03}, \mathbf{0.12})$ | $749_{\pm 292}$ |
| 0.1 | 0.1 | 10 | $(\mathbf{0.07}, \mathbf{0.28})$ | $21852_{\pm 927}$ |

*Table 2.* Comparison of fairness-utility trade-offs. We report predictive accuracy and fairness metrics, including DPD and EOD over three runs, while subscripts indicate standard deviations. For the Nonconvex Regime, we report the best model that minimizes DPD subject to the requirement of predictive accuracy $\geq 90\%$.

| Dataset | Method | Acc ($\uparrow$) | DPD ($\downarrow$) | EOD ($\downarrow$) |
|---|---|---|---|---|
| *Convex-Concave Regime: logistic regression* | | | | |
| | EG | $84.3_{\pm 0.0}$ | $0.191_{\pm 0.001}$ | $0.282_{\pm 0.008}$ |
| Adult | LEN | $\mathbf{84.9}_{\pm 0.0}$ | $0.184_{\pm 0.000}$ | $0.212_{\pm 0.000}$ |
| | **SPACO** | $84.8_{\pm 0.0}$ | $\mathbf{0.177}_{\pm 0.001}$ | $\mathbf{0.195}_{\pm 0.002}$ |
| *Nonconvex Regime: deep adversarial learning* | | | | |
| | Vanilla | $\mathbf{96.0}_{\pm 0.0}$ | $0.188_{\pm 0.003}$ | $0.506_{\pm 0.016}$ |
| CelebA | Adversarial | $91.3_{\pm 1.5}$ | $0.080_{\pm 0.023}$ | $0.273_{\pm 0.021}$ |
| | **SPACO** | $90.8_{\pm 0.3}$ | $\mathbf{0.057}_{\pm 0.010}$ | $\mathbf{0.076}_{\pm 0.020}$ |

## 5.2. Fairness-aware Classification

Adversarial debiasing (Zhang et al., 2018) aims to learn predictive representations invariant to sensitive attributes while maintaining predictive utility. Recently, (Chen et al., 2025) reformulated this task into a minimax optimization, which relies on a prior weight $\beta$ to govern the trade-off between predictive utility and privacy protection, rendering the solution highly sensitive to hyperparameter tuning. To mitigate the sensitivity, we adopt a CCM formulation:

$$\min_{\theta} \max_{\varphi} \left\{ \mathcal{L}_{\text{pred}}(\theta) - \beta \mathcal{L}_{\text{adv}}(\theta, \varphi) \mid \mathcal{L}_{\text{adv}}(\theta, \varphi) \leq \kappa \right\}.$$

Here, the constraint $\mathcal{L}_{\text{adv}} \leq \kappa$ enforces a mandatory competency level on the adversary, ensuring the predictor is optimized against a sufficiently strong opponent.

We evaluate SPACO across two distinct regimes to assess both optimization efficiency and scalability. In the Convex-Concave Regime, following (Chen et al., 2025), we utilize the UCI Adult dataset (Chang & Lin, 2011) and solve an $\ell_2$-regularized logistic regression game, which is a convex-concave minimax problem. In this setting, we compare SPACO, solving the CCM formulation, against unconstrained case obtained by the classical first-order solver ExtraGradient (EG) (Korpelevich, 1976) and the advanced second-order solver LEN (Chen et al., 2025).

In the Nonconvex Regime, following standard adversarial debiasing protocols, we utilize the CelebA dataset (Liu et al.,

*Table 3.* Quantitative results on GAN. We report FID and IS scores over three runs, while subscripts indicate standard deviations.

| Method | CIFAR-10 | | AFHQ-v2 | |
|---|---|---|---|---|
| | FID ($\downarrow$) | IS ($\uparrow$) | FID ($\downarrow$) | IS ($\uparrow$) |
| GAN | $35.33_{\pm 2.58}$ | $6.23_{\pm 0.24}$ | $28.50_{\pm 1.52}$ | $6.47_{\pm 0.14}$ |
| GAN-C | $21.19_{\pm 1.34}$ | $7.39_{\pm 0.07}$ | $26.46_{\pm 0.74}$ | $6.62_{\pm 0.15}$ |
| **SPACO** | $\mathbf{18.78}_{\pm 1.41}$ | $\mathbf{7.73}_{\pm 0.15}$ | $\mathbf{24.44}_{\pm 0.95}$ | $\mathbf{6.89}_{\pm 0.09}$ |

2015) and solve a fairness learning task using a ResNet-18 predictor and an MLP adversary. In this setting, we compare SPACO against the Vanilla classification (standard training without adversarial debiasing), as well as unconstrained solutions obtained by the seminal projection-based heuristic Adversarial Debiasing (Adversarial) (Zhang et al., 2018).

Performance across both regimes is assessed by predictive Accuracy (Acc) and fairness metrics, including Demographic Parity Difference (DPD) (Feldman et al., 2015) and Equalized Odds Difference (EOD) (Hardt et al., 2016). Detailed numerical results are reported in Table 2. The results demonstrate that SPACO consistently achieves the lowest bias metrics while maintaining predictive accuracy.

## 5.3. Generative Adversarial Networks

To further demonstrate the efficacy of SPACO on large-scale real-world applications, we apply it to the training of Constrained Generative Adversarial Networks (GAN-C) (Chao et al., 2021). GAN-C introduces a coupled constraint to mitigate training instability by bounding the discrepancy between the discriminator's outputs on real and generated samples, which is formulated as:

$$\min_{D} \max_{G, c(G,D) \leq 0} \mathbb{E}_{x_r, z} \left[ \log D(x_r) + \log \left( 1 - D(G(z)) \right) \right],$$

where $c(G, D) := \mathbb{E}_{x_r, z} \left[ \log D(x_r) - \log D(G(z)) \right]^2 - \epsilon$. Here $x_r$ denotes real data, $z$ is the latent noise, and $\epsilon \geq 0$ is a tolerance parameter. This constraint prevents the discriminator from dominating the generator prematurely, thereby promoting stable learning dynamics. We compare SPACO with the unconstrained GAN baseline and the original GAN-C method (Chao et al., 2021) on two benchmark datasets, CIFAR-10 (Krizhevsky, 2009) and AFHQ-v2 (Choi et al., 2020). All methods are implemented using the same Spectral Normalization GAN architecture (Miyato et al., 2018) to ensure a fair comparison. Performance is evaluated using the Fréchet Inception Distance (FID) (Heusel et al., 2017) and Inception Score (IS) (Salimans et al., 2016). As reported in Table 3, SPACO consistently yields the best FID and IS scores compared to the baselines, demonstrating the practical effectiveness of the proposed algorithm.

## Impact Statement

This paper presents work whose goal is to advance the field of Machine Learning. There are many potential societal consequences of our work, none which we feel must be specifically highlighted here.

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

## A. Related work

**Minimax optimization without coupled constraints.** Most existing work on minimax optimization focuses on minimax problems without coupled constraints. According to the properties of the objective function, these works are usually classified into convex–concave, nonconvex–strongly concave, nonconvex–concave, and nonconvex–nonconcave settings. In particular, convex–concave minimax problems can also be viewed as a class of variational inequality problems (Nemirovski, 2004; Kinderlehrer & Stampacchia, 2000; Gidel et al., 2019). Classical solution methods include the extragradient method (Nesterov, 2007; Monteiro & Svaiter, 2011) and gradient descent–ascent algorithms (Nedić & Ozdaglar, 2009; Mokhtari et al., 2020). Stochastic variants of these methods have also been developed accordingly (Nemirovski et al., 2009; Juditsky et al., 2011; Chen et al., 2014). In addition, for a special linearly coupled objective structure, namely $f(x, y) = f_1(x) + \langle Ax, y \rangle - f_2(y)$, (Chambolle & Pock, 2011) introduced a primal–dual variant of the proximal point method, which was further extended to nonlinear coupling structures by (Valkonen, 2014). For the nonconvex–strongly concave setting, existing studies often adopt a value-function perspective, using the stability properties of the value function as convergence criteria, and employ gradient descent–ascent schemes, including both deterministic and stochastic algorithms (Lin et al., 2020a; Sanjabi et al., 2018; Lin et al., 2020b). For the nonconvex–concave setting, various nested-loop algorithms (Nouiehed et al., 2019; Thekumparampil et al., 2019; Kong & Monteiro, 2021) and single-loop algorithms (Lu et al., 2020; Xu et al., 2023; Zhang et al., 2020b) have been proposed, covering both deterministic and stochastic variants. For the more general nonconvex–nonconcave setting, existing work typically relies on additional convex–concave–type assumptions, such as Stampacchia or Minty variational inequality conditions (Zhou et al., 2017), the Polyak–Łojasiewicz condition (Nouiehed et al., 2019), or weakly convex–weakly concave structures (Grimmer et al., 2023).

**Minimax optimization with coupled constraints.** MCC has found applications in a variety of domains, including constrained generative adversarial network training (Chao et al., 2021), perceptual adversarial robustness (Laidlaw et al., 2021), adversarial attacks in network flow problems (Tsaknakis et al., 2023), as well as absolute value equations and linear projection equations (Dai et al., 2024). From a theoretical perspective, optimality conditions for MCC have been investigated in (Dai & Zhang, 2020; Guo et al., 2024; Ma & Ye, 2025). On the algorithmic side, for minimax optimization with linear coupled constraints, (Dai et al., 2024) proposed an alternating coordinate method, for a special linearly coupled objective structure, namely, $f(x, y) = f_1(x) + \langle Ax, y \rangle - f_2(y)$. (Tsaknakis et al., 2023) exploited a dual reformulation that converts MCC into a min–min–max structure and introduced a double loop Multiplier Gradient Descent algorithm. Building upon this min–min–max reformulation, (Zhang & Xu, 2024; Zhang et al., 2024) developed a class of single-loop primal–dual gradient methods for both deterministic and stochastic objectives. However, when linear constraints are extended to nonlinear coupled constraints, the boundedness of the multiplier sequence becomes difficult to guarantee, which in turn affects the Lipschitz continuity of the gradient of the reformulated problem. Whether primal–dual gradient methods with convergence guarantees can be developed for nonlinear coupled constraints remains an open question.

For deterministic minimax optimization with nonlinear coupled constraints, the existing literature is considerably more limited. (Goktas & Greenwald, 2021) proposed max-oracle gradient descent and nested gradient methods, under the assumptions that the value function $\varphi(x)$ is convex and that a Lagrange multiplier associated with the constraints is accessible. (Dai & Zhang, 2024) proposed an augmented Lagrangian method for minimax optimization with equality constraints with the assumption that the corresponding subproblems can be solved and analyzed the convergence rate. (Lu & Mei, 2025) first studied MCC with non-smooth objective and nonlinear inequality constraints and proposed a new first-order algorithm based on the augmented Lagrangian framework. More recently, (Hu et al., 2024) established the equivalence between the min–min–max reformulation and the original MCC problem in terms of first-order minimax stationary points, and designed a subgradient method by reformulating MCC as a standard minimax optimization problem by Moreau envelope.

## B. Details of Experiments

### B.1. Synthetic Example

As some existing algorithms focus on minimax optimization with linear coupled constraints, besides the nonlinear constrained problem (12), we additionally consider the following stochastic minimax problem with linear constraint:

$$\min_{x \in X} \max_{y \in Y} \left\{ \mathbb{E}_W \left[ \frac{1}{2} x^\top (\bar{A} + W)x - \left( \frac{1}{2} \|y_1\|^2 - x^\top y_1 + e^\top y_2 + \frac{1}{2} \|y_2 + 2x + e\|^2 \right) \right] \mid e^\top x + e^\top y_1 + e^\top y_2 = 0 \right\},$$

(13)

where $X = [-10, 10]^n$, $Y = [-20, 20]^{2n}$ and $y_1, y_2 \in \mathbb{R}^n$ $\bar{A} \in \mathbb{R}^{n \times n}$ is a fixed matrix such that $\bar{A} + I$ is positive definite, and $W$ is a Gaussian random matrix with independent entries $W_{ij} \sim \mathcal{N}(0, \delta^2)$. With simple calculation, for a fixed $x$, the solution for the maximization subproblem is $(y_1^*(x), y_2^*(x)) = (x + e, -2x - e)$. And the expected optimal solution should be $(x^*, y_1^*, y_2^*) = (-2(\bar{A} + I)^{-1}e, x^* + e, -2x^* - e)$. The convergence curve for both errors over 10 runs are summarized in Figure 3.

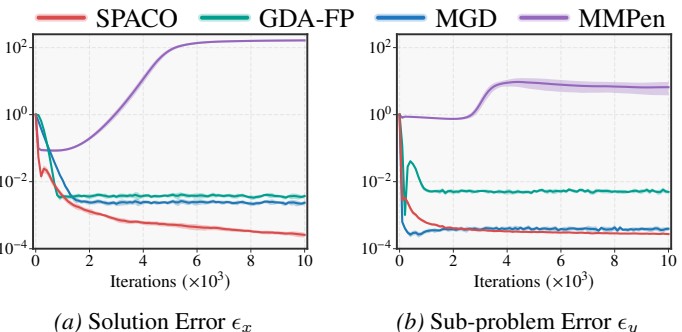

*(a)* Solution Error $\epsilon_x$       *(b)* Sub-problem Error $\epsilon_y$

*Figure 3.* Convergence curve for solving (13)

### B.1.1. EXPERIMENTAL SETUP AND HYPERPARAMETER TUNING

We evaluate the proposed algorithms on two synthetic stochastic minimax optimization: a nonlinear constrained problem (12) and a linear constrained problem (13). For both experimental settings, the problem dimension is fixed at $n = 100$, and the noise variance is set to $\delta^2 = 1$. Regarding initialization, the starting points $z_0 = (x_0, y_0)$ are sampled uniformly at random from the feasible region for each independent run. Specifically for the linear case (13), the fixed matrix $\bar{A}$ is generated as $\bar{A} = \frac{1}{n} M M^\top + I_n$ where entries of $M \in \mathbb{R}^{n \times n}$ are drawn from $\mathcal{N}(0, 1)$, ensuring that $\bar{A} + I \succ 0$.

Due to the inherent variance in stochastic gradients, computing the exact stationarity gap is computationally intractable. To reliably approximate this metric under noise, we employ the large-sample estimation technique described in (Ghadimi & Lan, 2013). Specifically, we denote the generalized gradient as

$$\mathcal{G}(z_k) = P_Z(z^k - \nabla l(z_k)),$$

where $Z$ is the feasible set, $l$ is the loss function and $z^k$ are the iterates. For example, for SPACO,

$$\mathcal{G}(z_k) = \left\| \begin{pmatrix} x_k - P_X(x_k - \nabla_x \psi_k(x^k, y^k)) \\ y_k - P_Y(y_k + \nabla_y \psi_k(x^k, y^k)) \end{pmatrix} \right\|.$$

We estimate the generalized gradient as squared norm of the expected generalized gradient $\mathcal{G}(z_k; \xi_i)$ using a fixed set of $T = 1000$ samples $\{\xi_i\}_{t=1}^T$:

$$\hat{\mathcal{E}}_k := \|\mathcal{G}_T(z_k)\|^2 \approx \left\| \frac{1}{T} \sum_{i=1}^T \mathcal{G}(z_k; \xi_i) \right\|^2,$$

Incorporating this estimator, we adopt a robust fixed-budget verification strategy to mitigate the sensitivity of minimax problems to initialization. For each hyperparameter configuration, we conduct 10 independent runs with random initializations, each lasting for a fixed budget of $K = 10000$ iterations. To capture asymptotic stability, we record the estimated stationarity gap $\hat{\mathcal{E}}_k$ every 100 iterations during the final 1000 steps of each run. The final performance score is reported as the average of these recorded values across all independent runs.

With the experimental verification metric established, we performed a grid search to identify the configuration that minimizes the average stationarity gap. The search spaces were consistent across both linear and nonlinear constrained problems:

- SPACO & GDA-FP: Penalty $\rho_0 \in \{5, 10, 20, 50\}$, stepsizes $\alpha_0, \beta_0 \in \{0.1, 0.01, 0.001\}$. For SPACO, we additionally tuned proximal $\sigma_0 \in \{10^{-3}, 10^{-4}, 10^{-5}\}$ and set $t = 0.05$ and $s = 0.2$.

- MGD: Primal stepsizes $\alpha, \beta \in \{0.1, 0.01, 0.001\}$, dual stepsize $\gamma \in \{1.0, 0.1, 0.01\}$, and inner loop steps $L \in \{1, 5, 10\}$.

- MMPen: Penalty coefficient $\alpha_{\text{pen}} \in \{20, 100, 500\}$ and stepsize $\eta \in \{0.5, 1.0, 2.0\}$.

The configuration yielding the lowest average stationarity gap was selected for the reported numerical results. The specific optimal hyperparameters derived from this search are summarized in Table 4.

*Table 4.* Optimal hyperparameters for synthetic examples.

| Algorithm | Nonlinear Constrained Problem (12) | Linear Constrained Problem (13) |
|---|---|---|
| **SPACO** | $\rho_0 = 10, \alpha_0 = 0.1, \beta_0 = 0.1, \sigma_0 = 10^{-4}$ | $\rho_0 = 20, \alpha_0 = 0.01, \beta_0 = 0.1, \sigma_0 = 10^{-4}$ |
| **GDA-FP** | $\rho_0 = 20, \alpha_0 = 0.001, \beta_0 = 0.01$ | $\rho_0 = 20, \alpha_0 = 0.001, \beta_0 = 0.01$ |
| **MGD** | $\alpha = 0.01, \beta = 0.01, \gamma = 0.1, L = 1$ | $\alpha = 0.001, \beta = 0.1, \gamma = 0.1, L = 1$ |
| **MMPen** | $\alpha_{\text{pen}} = 20, \eta = 1.0$ | $\alpha_{\text{pen}} = 100, \eta = 1.0$ |

### B.2. Fairness-aware Classification

#### B.2.1. COMPETENCY-CONSTRAINED MINIMAX (CCM)

The deployment of machine learning models in high-stakes domains has necessitated the development of algorithms that ensure fairness alongside predictive utility. A seminal framework in this domain is Adversarial Debiasing, first formalized by (Zhang et al., 2018). Consider a dataset comprised of tuples $(X, Y, Z)$, where $X$ represents the input features, $Y$ is the target label, and $Z$ is a protected attribute (e.g., gender or race). This framework introduces a predictor parameterized by $\theta$ to perform the target task and an adversary parameterized by $\varphi$ to recover the sensitive attribute $Z$ from the predictor's representations. The goal is to learn the predictor parameters $\theta$ that minimize the predictive loss $\mathcal{L}_{\text{pred}}(\theta)$ while maximizing the adversarial loss $\mathcal{L}_{\text{adv}}(\theta, \varphi)$ against an adversary $\varphi$ trained to recover $Z$, thereby minimizing the leakage of information regarding the sensitive attribute.

(Zhang et al., 2018) proposed solving this via a gradient projection mechanism to prevent the encoding of sensitive information. Specifically, they modify the standard gradient for the predictor $\theta$ by removing the component parallel to the adversary's gradient. The effective update direction $g_\theta$ is formulated as:

$$g_\theta = \nabla_\theta \mathcal{L}_{\text{pred}}(\theta) - \frac{\langle \nabla_\theta \mathcal{L}_{\text{pred}}(\theta), \nabla_\theta \mathcal{L}_{\text{adv}}(\theta, \varphi) \rangle}{\|\nabla_\theta \mathcal{L}_{\text{adv}}(\theta)\|^2} \nabla_\theta \mathcal{L}_{\text{adv}} - \alpha \nabla_\theta \mathcal{L}_{\text{adv}}(\theta, \varphi), \tag{14}$$

where $\alpha$ determines the adversarial strength. This approach essentially simulates a multi-objective optimization process by aggregating the gradients of objectives, while pioneering but may resulting in training instability and sensitivity to the hyperparameter $\alpha$.

Recently, (Chen et al., 2025) reformulated this task through the lens of minimax optimization, aiming to solve

$$\min_\theta \max_\varphi \mathcal{L}_{\text{pred}}(\theta) - \beta \mathcal{L}_{\text{adv}}(\theta, \varphi) \tag{15}$$

where the scalar $\beta > 0$ governs the critical trade-off between predictive utility and privacy. While (Chen et al., 2025) utilized a Lazy Hessian strategy to accelerate computation, the reliance on second-order information incurs prohibitive computational overhead for large-scale deep neural networks. Furthermore, this unconstrained objective remains highly sensitive to $\lambda$, where an improper static weight often leads to either insufficient debiasing or severe utility collapse.

To address these limitations, we propose the **Competency-Constrained Minimax (CCM)** framework. We augment the adversarial minimax problem (15) with an explicit competency constraint:

$$\min_\theta \max_\varphi \left\{ \mathcal{L}_{\text{pred}}(\theta) - \beta \mathcal{L}_{\text{adv}}(\theta, \varphi) \mid \mathcal{L}_{\text{adv}}(\theta, \varphi) \leq \kappa \right\}. \tag{16}$$

This MCC formulation enforces a mandatory *competency requirement* on the adversary. While $-\beta \mathcal{L}_{\text{adv}}(\theta, \varphi)$ provides continuous gradient signals to prevent stagnation, the constraint ensures the adversary maintains a loss below $\kappa$. Intuitively, since a random guess on a binary attribute yields a cross-entropy loss of $\ln 2 \approx 0.69$, setting $\kappa$ slightly below this theoretical baseline (e.g., $\kappa = 0.65$) serves as a natural and sufficient criterion for competency. This restricts the game to a regime where the adversary is knowledgeable, preventing the "lazy adversary" phenomenon and ensuring the predictor is optimized against a consistently strong opponent.

### B.2.2. EXPERIMENTAL SETUP

We structure our evaluation into two distinct regimes: the **Convex-Concave Regime** for benchmarking against established minimax solvers, and the **Nonconvex Regime** for demonstrating scalability on large-scale deep learning tasks. Across both regimes, we assess performance using predictive Accuracy (Acc) for utility, and Demographic Parity Difference (DPD) (Feldman et al., 2015) and Equalized Odds Difference (EOD) (Hardt et al., 2016) for fairness. Lower DPD and EOD values indicate better fairness, with 0 being the ideal state. Detailed definitions are provided at the end of this section.

**1. Convex-Concave Regime: Fairness-aware Logistic Regression.** To evaluate optimization efficacy in a controlled setting, we adopt the convex-concave protocol from (Chen et al., 2025). We use the UCI Adult dataset (Chang & Lin, 2011) to predict annual income ($> \$50k$) with gender as the protected attribute. The problem is formulated as an $\ell_2$-regularized logistic regression game:

$$\min_{\mathbf{x}\in\mathbb{R}^d} \max_{y\in\mathbb{R}} \frac{1}{n} \sum_{i=1}^{n} \ell(b_i \mathbf{a}_i^\top \mathbf{x}) - \beta\ell(c_i y \mathbf{a}_i^\top \mathbf{x}) + \lambda\|\mathbf{x}\|^2 - \gamma y^2, \tag{17}$$

where $d$ is the feature dimension, and logit function $\ell(t) = \log(1 + \exp(-t))$. We set adversarial weight $\beta = 0.5$ and regularization parameters $\lambda = \gamma = 10^{-4}$. We compare SPACO against the classic convex-concave minimax solver ExtraGradient (EG) (Korpelevich, 1976) and an advanced Newton-type solver LEN (Chen et al., 2025).

All algorithms are run for a fixed duration of 100 seconds with stepsizes of 0.1 for both $x$ and $y$. For LEN, we adopt the recommended Lazy Hessian frequency $m = 10$. For SPACO, we configure the competency requirement $\kappa = 0.65$, initial penalty $\rho_0 = 50$, batch size 512, and factors $t = 0.01, s = 0.04$.

**2. Nonconvex Regime: Deep Fairness Learning.** To demonstrate scalability, we perform fairness-aware classification on the CelebA dataset (Liu et al., 2015), aiming to classify *Blond Hair* while mitigating bias related to *Gender*. Images are center-cropped to $178 \times 178$ and resized to $224 \times 224$. We employ a pretrained ResNet-18 backbone as the predictor and a MLP as the adversary. We compare SPACO against Vanilla, the standard classification training without adversarial debiasing, and Adversarial Debiasing (Adversarial) (Zhang et al., 2018), the seminal projection-based heuristic for deep fairness learning.

We use a batch size of 256 and train for 20 epochs. The stepsizes for both the predictor $\theta$ and adversary $\phi$ are set to 0.002. For SPACO, the hyperparameters are set as follows: competency requirement $\kappa = 0.65$, adversarial scale $\beta = 4.0$, initial penalty $\rho_0 = 20$, initial proximal parameter $\sigma_0 = 10^{-6}$, and factors $t = 0.01, s = 0.04$.

To ensure a rigorous comparison, we adopt a threshold-based model selection criterion. We evaluate checkpoints at the end of every epoch and identify the "best model" as the one achieving the lowest fairness violation (minimum DPD) subject to a utility constraint (Acc $\geq 90\%$). For Adversarial Debiasing, We perform a grid search for the adversarial strength $\alpha \in \{0.1, 0.3, 0.5, 0.7, 0.9\}$. As illustrated in Figure 5a, we identify $\alpha = 0.3$ as the optimal setting that yields the lowest DPD under the 90% Acc requirement.

**Metric Definitions.** Let $Y, \hat{Y} \in \{0, 1\}$ denote the ground-truth and predicted labels, and $Z \in \{0, 1\}$ the sensitive attribute. DPD measures the absolute discrepancy in positive prediction rates between groups: DPD $= |P(\hat{Y} = 1 \mid Z = 0) - P(\hat{Y} = 1 \mid Z = 1)|$. EOD quantifies the disparity in error rates: EOD $= |\text{TPR}_0 - \text{TPR}_1| + |\text{FPR}_0 - \text{FPR}_1|$, where $\text{TPR}_z = P(\hat{Y} = 1 \mid Y = 1, Z = z)$ and $\text{FPR}_z = P(\hat{Y} = 1 \mid Y = 0, Z = z)$ denote True Positive Rate and False Positive Rate, respectively.

### B.2.3. RESULTS AND DISCUSSION

To substantiate the superiority of SPACO in both efficiency and effectiveness, we provide a detailed visualization of the training dynamics in the convex-concave regime. Figure 4 presents the evolution of Predictive Accuracy, DPD, and EOD with respect to wall-clock time on the Adult dataset. SPACO demonstrates the fastest convergence speed, rapidly reducing both DPD and EOD metrics to stabilize at the lowest bias levels (DPD $\approx 0.177$, EOD $\approx 0.195$) at least $5\times$ faster. This highlights the superior ability of SPACO in solving the CCM problem to mitigate privacy risk efficiently.

Figure 5a reveals the instability of standard Adversarial Debiasing: increasing the adversarial strength $\alpha$ leads to a catastrophic drop in predictive accuracy (solid lines), forcing a difficult trade-off selection. In contrast, SPACO demonstrates remarkable robustness. As shown in Figures 5b and 5c, the algorithm consistently converges to a low-bias state (dashed lines)

while maintaining high predictive utility (solid lines) across a wide range of adversarial scales $\beta$ and penalty parameters $\rho_0$. This confirms that the competency-constrained formulation effectively prevents the adversary from destroying predictive features, ensuring robust and stable training in deep adversarial learning.

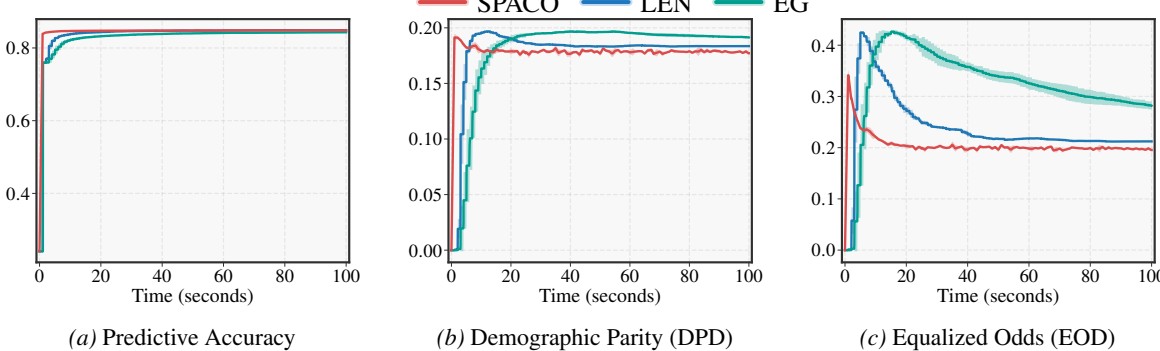

*(a)* Predictive Accuracy      *(b)* Demographic Parity (DPD)      *(c)* Equalized Odds (EOD)

*Figure 4.* Training dynamics on the Convex-Concave Regime: Logistic regression on the Adult dataset.

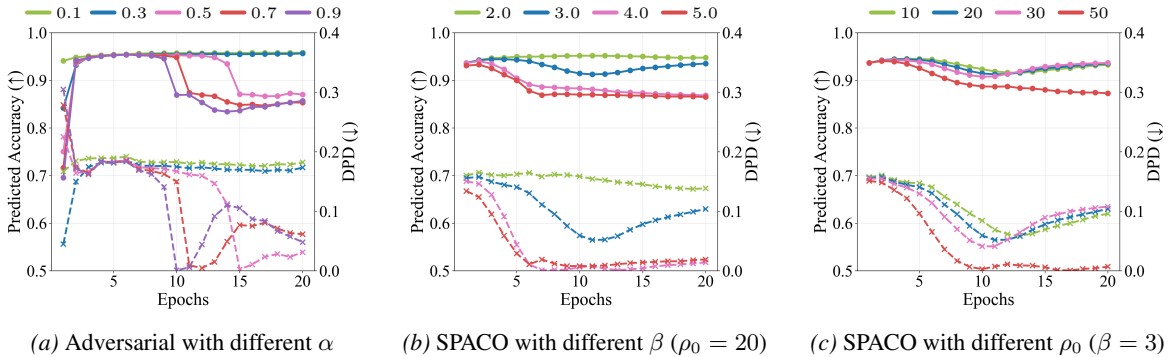

*(a)* Adversarial with different $\alpha$    *(b)* SPACO with different $\beta$ ($\rho_0 = 20$)    *(c)* SPACO with different $\rho_0$ ($\beta = 3$)

*Figure 5.* Training dynamics on the Nonconvex Regime: Deep fairness learning on the CelebA dataset. Solid lines ($\bullet$) and dashed lines ($\times$) indicate predicted accuracy and demographic parity difference (DPD), respectively.

## B.3. Generative Adversarial Networks

To further assess the practical efficiency of SPACO in real-world large-scale applications, we apply SPACO to the training of Generative Adversarial Networks (GANs).

### B.3.1. CONSTRAINED GENERATIVE ADVERSARIAL NETWORK

Standard GANs formulate training as a minimax game between a generator $G$ and a discriminator $D$. Despite their empirical success, unconstrained GANs are well known to suffer from training instability and high sensitivity to hyperparameter choices. To address these issues, (Chao et al., 2021) proposed Constrained Generative Adversarial Networks (GAN-C), which introduce an explicit coupled constraint and are formulated as

$$\min_{G} \max_{D} \ \{\mathcal{L}_{\text{GAN}}(G, D) \mid c(G, D) \leq 0\},$$

where $\mathcal{L}_{\text{GAN}}(G, D) := \mathbb{E}_{x_r, z}\big[\log D(x_r) + \log\big(1 - D(G(z))\big)\big]$ is the standard adversarial loss for GAN training, $x_r \sim p_{\text{data}}$ denotes a real data sample and $z \sim p_z$ is a latent variable drawn from the prior distribution. The coupled constraint $c(G, D)$ is defined as

$$c(G, D) := \mathbb{E}_{x_r, z}\big[\log D(x_r) - \log D(G(z))\big]^2 - \epsilon,$$

with $\epsilon \geq 0$, which enforces a bounded discrepancy between the discriminator's outputs on real and generated samples. This constraint prevents the discriminator from dominating the generator too early during training, thereby promoting more stable learning dynamics.

To solve this constrained formulation, (Chao et al., 2021) relax the hard constraint into a soft regularization term, converting the problem into an unconstrained minimax game. Specifically, they modify the discriminator's objective to maximize a penalized value function:

$$\max_D \left( \mathcal{L}_{\text{GAN}}(G, D) - \lambda \left[ c(G, D) \right]^2 \right),$$

where $\lambda > 0$ is a fixed penalty coefficient. This formulation explicitly compels the discriminator to balance classification performance with the stability constraint.

### B.3.2. EXPERIMENTS

We evaluate our method on two benchmark datasets with varying degrees of complexity to verify the robustness and scalability of SPACO. First, we use CIFAR-10 (Krizhevsky, 2009), resized to $64 \times 64$ resolution, a standard benchmark for measuring image generation performance on diverse object classes. Second, to assess performance on high-fidelity generation tasks, we utilize AFHQ-v2 (Animal Faces-HQ) (Choi et al., 2020), which consists of high-quality images of animal faces, resized to $128 \times 128$ resolution. Figure 6 visualizes representative samples from this dataset, which covers three distinct categories: cat, dog, and wildlife.

To ensure a fair comparison and isolate the efficacy of the optimization algorithms, we employ a consistent network architecture across all baselines. Specifically, we adopt the Spectral Normalization GAN (SN-GAN) (Miyato et al., 2018) for both the generator and discriminator. SN-GAN is chosen for its widespread adoption and ability to stabilize the training of deep generative models.

Across all experiments, we employ the Adam optimizer with learning rates fixed at $2 \times 10^{-4}$ and hyperparameters set to $\beta_1 = 0.0$ and $\beta_2 = 0.9$ across all experiments. We train the models for 100 epochs on both CIFAR-10 and AFHQ-v2 datasets, using batch sizes of 128 and 64, respectively. For evaluation, we employ the Exponential Moving Average (EMA) of the generator weights with a decay rate of 0.999. Regarding the penalty-based formulations, we set $\epsilon = 0$ and set the fixed penalty parameter $\rho = 5$ for the GAN-C baseline and identically initialize the penalty parameter $\rho_0 = 5$ for SPACO. Additionally, we set factors $t = 0.9$ and $s = 0.28$ for SPACO.

We compare SPACO against two primary baselines: the unconstrained GAN (standard minimax formulation) and the original GAN-C solver (Chao et al., 2021). Performance is assessed using two standard metrics:

- Fr'echet Inception Distance (FID) (Heusel et al., 2017): Measures the Wasserstein-2 distance between the feature distributions of real and generated images. Lower FID scores indicate higher fidelity and realism.

- Inception Score (IS) (Salimans et al., 2016): Evaluates the distinctiveness and diversity of the generated samples, with higher scores indicating better performance.

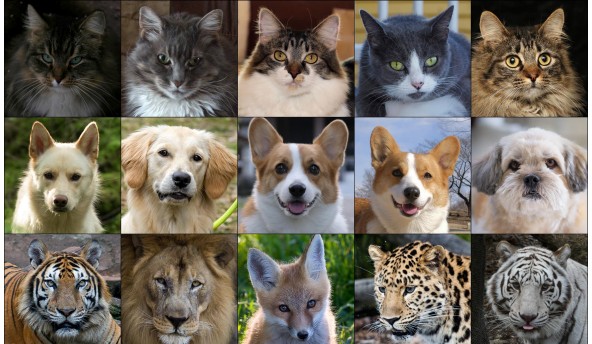 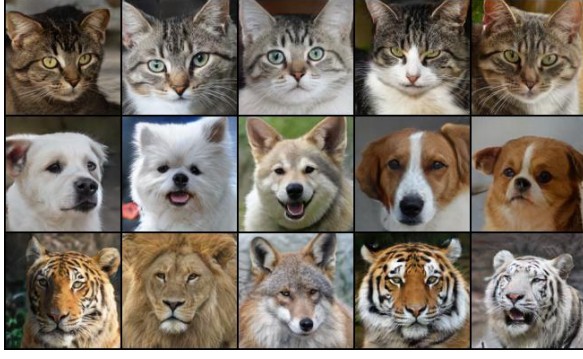

*Figure 6.* **Visual comparison on the AFHQ-v2 dataset.** Left: Representative real samples from the original training set. Right: Samples generated by the SPACO-trained model. The side-by-side comparison demonstrates that SPACO is capable of synthesizing high-fidelity images with visual quality and diversity comparable to the real data.

We report the mean results over three independent runs. and standard deviations averaged over three independent runs with random seeds to ensure statistical reliability. The quantitative results are summarized in Table 3. As illustrated in the table, SPACO achieves the lowest FID score, indicating that the smoothing framework effectively handles the coupled constraint

without compromising the generative quality. Besides, we provide a visual inspection of the generated samples in Figure 6. The generated images demonstrate high fidelity and diversity across different animal categories (cats, dogs, and wildlife).

## C. Proof for Section 2

In this section, we provide a detailed proof for the results proposed for the smooth approximation in Section 2. To simplify notation, we denote $\psi_k(x, y) := \psi_{\rho_k, \sigma_k}(x, y)$, $\varphi_k(x) := \varphi_{\rho_k, \sigma_k}(x)$, and $y_k^*(x) := y_{\rho_k, \sigma_k}^*(x)$.

### C.1. Constraint Qualification Conditions

**Definition C.1.** (Assumption 5 (ii) in (Lu & Mei, 2025)) Suppose that for every $\bar{x} \in X$ with $\Theta(\bar{x}) := \{y \in Y : c(x, y) \leq 0\} \neq \emptyset$, there exist a point $\tilde{y} \in \mathrm{ri}(Y)$ and constants $\gamma > 0$ and $\delta > 0$ such that

$$c(x, \tilde{y}) \leq -\gamma \mathbf{e}, \qquad \forall x \in B_\delta(\bar{x}) \cap X,$$

where $\mathbf{e}$ is the vector of all ones of appropriate dimension, and the inequality holds componentwise. Then, we say that the **consistent Slater condition** holds for the feasible system $X, Y, c(x, y) \leq 0$.

**Definition C.2.** The set-valued mapping $\Theta : X \rightrightarrows Y$ is called **inner semi-continuous** at $(\bar{x}, \bar{y})$ with $\bar{y} \in \Theta(\bar{x})$ if, for every sequence $x_k \in X$ converging to $\bar{x}$, there exists a sequence $y_k \in \Theta(x_k)$ such that $y_k \to \bar{y}$. If this property holds for every $\bar{x} \in X$ and every $\bar{y} \in \Theta(\bar{x})$, we say $\Theta$ is inner semi-continuous on $X$.

**Lemma C.3.** *Suppose that $\Theta(x) := \{y \in Y : c(x, y) \leq 0\}$ is inner semi-continuous on $X$, then $\varphi(x)$ is lower semi-continuous.*

*Proof.* Let $\bar{x} \in X$. Since $\varphi(\bar{x}) = \max_{y \in \Theta(\bar{x})} f(\bar{x}, y)$, there exist $\bar{y} \in \Theta(\bar{x})$ such that $f(\bar{x}, \bar{y}) = \varphi(\bar{x})$. Since $\Theta(x)$ is inner semi-continuous on $X$, then for all $x_k \to \bar{x}$, we can find a sequence $y_k \in \Theta(x_k)$ and $y_k \to \bar{y}$. Consequently,

$$\liminf_{k \to \infty} \varphi(x_k) \geq \liminf_{k \to \infty} f(x_k, y_k) = f(\bar{x}, \bar{y}) = \varphi(\bar{x}). \tag{18}$$

which implies that $\varphi(x)$ is lower semi-continuous. The proof is completed. $\qquad\square$

**Lemma C.4.** *Suppose that consistent Slater condition holds for the feasible system $X, Y, c(x, y) \leq 0$ and $X$ and $Y$ are the whole space , then the GPŁCQ (Definition 2.8) holds at $(x^*, y^*)$.*

*Proof.* Let $(x^*, y^*)$ be a feasible point, i.e., $c(x^*, y^*) \leq 0$. Since $X$ and $Y$ are the whole space, we have

$$P_{X \times Y}\left(\begin{pmatrix} x \\ y \end{pmatrix} - \beta \begin{pmatrix} \nabla_x p(x, y) \\ -\nabla_y p(x, y) \end{pmatrix}\right) = \begin{pmatrix} x \\ y \end{pmatrix} - \beta \begin{pmatrix} \nabla_x p(x, y) \\ -\nabla_y p(x, y) \end{pmatrix}.$$

Thus it suffices to show that there exist $\delta$ and $v > 0$ such that

$$\sqrt{p(x, y)} \leq v \|\nabla p(x, y)\|, \qquad \forall (x, y) \in B_\delta(x^*) \times B_\delta(y^*). \tag{19}$$

Let $\mathcal{A} := \{i \in [p] : c_i(x^*, y^*) = 0\}$ and $J_{\mathcal{A}}^\star \in \mathbb{R}^{|\mathcal{A}| \times (n+m)}$ be the Jacobian of $c_{\mathcal{A}}$ at $(x^*, y^*)$. We claim that

$$(J_{\mathcal{A}}^\star)^\top s = 0, \ s \in \mathbb{R}_+^{|\mathcal{A}|} \implies s = 0. \tag{20}$$

By consistent Slater, the Slater condition holds uniformly for the sliced system $c(x^*, y) \leq 0$. Thus, MFCQ holds for the system $c(x^*, y) \leq 0$ at $y^*$, i.e., there exists $d_y \in \mathbb{R}^m$ such that $\langle \nabla_y c_i(x^*, y^*), d_y \rangle < 0$ for all $i \in \mathcal{A}$. Set $d := (0, d_y) \in \mathbb{R}^{n+m}$, then $(J_{\mathcal{A}}^\star)d < 0$ componentwise. If $(J_{\mathcal{A}}^\star)^\top s = 0$ with $s \geq 0$ and $s \neq 0$, then

$$0 = \langle s, (J_{\mathcal{A}}^\star)d \rangle < 0,$$

a contradiction. Hence (20) holds.

Define the compact set $S := \{s \in \mathbb{R}_+^{|\mathcal{A}|} : \|s\| = 1\}$. By (20), the continuous map $s \mapsto \|(J_{\mathcal{A}}^\star)^\top s\|$ is strictly positive on $S$,

$$\bar{\sigma} := \min_{s \in S} \|(J_{\mathcal{A}}^\star)^\top s\| > 0.$$

By continuity of $J_{\mathcal{A}}(x, y)$, there exists $\delta_1 > 0$ such that for all $(x, y) \in B_{\delta_1}(x^*) \times B_{\delta_1}(y^*)$,

$$\|J_{\mathcal{A}}(x, y)^\top s\| \geq \frac{\bar{\sigma}}{2}\|s\|, \qquad \forall s \in \mathbb{R}_+^{|\mathcal{A}|}. \tag{21}$$

Since $c_i(x^*, y^*) < 0$ for all $i \notin \mathcal{A}$ and $c$ is continuous, there exists $\delta_2 > 0$ such that for all $(x, y) \in B_{\delta_2}(x^*) \times B_{\delta_2}(y^*)$,

$$c_i(x, y) < 0, \quad \forall i \notin \mathcal{A}.$$

Thus the violated set $V(x, y) := \{i : c_i(x, y) > 0\}$ satisfies $V(x, y) \subseteq \mathcal{A}$.

Let $\theta(x, y) := [c(x, y)]_+ \in \mathbb{R}_+^p$ and denote by $\theta_A(x, y) \in \mathbb{R}_+^{|\mathcal{A}|}$ its restriction to $\mathcal{A}$. Then

$$\nabla p(x, y) = J(x, y)^\top \theta(x, y) = J_{\mathcal{A}}(x, y)^\top \theta_A(x, y).$$

Combining with (21) yields, for all $(x, y)$ close enough,

$$\|\nabla p(x, y)\| \geq \frac{\bar{\sigma}}{2}\|\theta_A(x, y)\| = \frac{\bar{\sigma}}{2}\|[c(x, y)]_+\|.$$

Since $\sqrt{p(x, y)} = \frac{1}{\sqrt{2}}\|[c(x, y)]_+\|$, we get

$$\sqrt{p(x, y)} \leq \frac{1}{\sqrt{2}} \cdot \frac{2}{\bar{\sigma}}\|\nabla p(x, y)\| = \frac{\sqrt{2}}{\bar{\sigma}}\|\nabla p(x, y)\|.$$

This proves (19) with $v = \sqrt{2}/\bar{\sigma}$ on a sufficiently small neighborhood, and hence the GPŁCQ inequality in Definition 2.8 follows. $\square$

### C.2. Proof for Proposition 2.2

**Proposition C.5.** *(Proposition 2.2) For any constant $\rho > 0$ and $\sigma > 0$, the function $\varphi_{\rho,\sigma} : X \to \mathbb{R}$ is differentiable. Its gradient is given by:*

$$\nabla \varphi_{\rho,\sigma}(x) = \nabla_x \psi_{\rho,\sigma}\big(x, y_{\rho,\sigma}^*(x)\big), \tag{22}$$

*where $y_{\rho,\sigma}^*(x)$ denotes the unique maximizer of the inner problem, i.e., $y_{\rho,\sigma}^*(x) := \arg\max_{y \in Y} \psi_{\rho,\sigma}(x, y)$.*

*Proof.* Since $\psi$ is continuously differentiable and strongly concave in $y$ for each $x \in X$, and $Y$ is a compact convex set, the maximizer $y_{\rho,\sigma}^*(x) := \arg\max_{y \in Y} \psi_{\rho,\sigma}(x, y)$ is unique and well-defined for each $x \in X$.

To apply (Bonnans & Shapiro, 2013, Theorem 4.13) to $-\psi_{\rho,\sigma}(x, y)$, we need to verify the inf-compactness condition: for any $\bar{x} \in X$, there exist a constant $c \in \mathbb{R}$, a compact set $B_y \subseteq \mathbb{R}^m$, and a neighborhood $B_x$ of $\bar{x}$ such that for all $x \in B_x$,

$$\{y \in Y \mid -\psi_{\rho,\sigma}(x, y) \leq c\} \neq \emptyset \quad \text{and} \quad \{y \in Y \mid -\psi_{\rho,\sigma}(x, y) \leq c\} \subseteq B_y.$$

Fix $\bar{x} \in X$. Let

$$c = -\psi_{\rho,\sigma}(\bar{x}, y_{\rho,\sigma}^*(\bar{x})) + 1.$$

Take $B_y = Y$ (which is compact by assumption). Define the function $h(x) := -\psi_{\rho,\sigma}(x, y_{\rho,\sigma}^*(\bar{x}))$. Since $\psi_{\rho,\sigma}$ is continuous (as $\psi$ is continuously differentiable), $h$ is continuous. Note that

$$h(\bar{x}) = -\psi_{\rho,\sigma}(\bar{x}, y_{\rho,\sigma}^*(\bar{x})) = c - 1 < c.$$

By continuity of $h$, there exists a neighborhood $B_x$ of $\bar{x}$ such that for all $x \in B_x$,

$$h(x) = -\psi_{\rho,\sigma}(x, y_{\rho,\sigma}^*(\bar{x})) < c.$$

This implies that for each $x \in B_x$, $y_{\rho,\sigma}^*(\bar{x}) \in \{y \in Y \mid -\psi_{\rho,\sigma}(x, y) \leq c\}$, so the set is nonempty. Moreover, since $Y$ is compact, the level set is automatically contained in $B_y = Y$. Thus the inf-compactness condition is satisfied. By (Bonnans & Shapiro, 2013, Theorem 4.13, Remark 4.14), we obtain the desired result. $\square$

### C.3. Proof for Lemma 2.3 and Lemma 2.5

Before presenting the proof for Proposition 2.3, we establish the feasibility of the limitation of the sequence $y_k^*(x_k)$.

**Lemma C.6.** *Let $\rho_k \to \infty$ and $\sigma_k \to 0$ as $k \to \infty$. Then, for any sequence $x_k$ such that $x_k \to \bar{x} \in X$, we have that:*

$$\limsup_{k\to\infty} \ [c(x_k, y_k^*(x_k))]_+ = 0. \tag{23}$$

*Consequently, any accumulation point of $(x_k, y_k^*(x_k))$ are feasible.*

*Proof.* Let $(\bar{x}, \bar{y})$ be a feasible point, i.e., $c(\bar{x}, \bar{y}) \le 0$. Recalling that $y_k^*(x_k) \in \arg\max_{y \in Y} \psi_k(x_k, y)$, we have that:

$$f(x_k, y_k^*(x_k)) - \frac{\rho_k}{2}[c(x_k, y_k^*(x_k))]_+^2 - \frac{\sigma_k}{2}\|y_k^*(x_k)\|^2 \ge f(x_k, \bar{y}) - \frac{\rho_k}{2}[c(x_k, \bar{y})]_+^2 - \frac{\sigma_k}{2}\|\bar{y}\|^2.$$

By rearranging this inequality, we have that

$$[c(x_k, y_k^*(x_k))]_+^2 - [c(x_k, \bar{y})]_+^2 \le \frac{2}{\rho_k}\left(f(x_k, y_k^*(x_k)) - f(x_k, \bar{y})\right) - \frac{\sigma_k}{\rho_k}\|y_k^*(x_k)\|^2 + \frac{\sigma_k}{\rho_k}\|\bar{y}\|^2.$$

Since $X$ and $Y$ are bounded, we have that $y_k^*(x_k)$ and $x_k$ are bounded. As $f(x, y)$ is continuous, $f(x_k, y_k^*(x_k)), f(x_k, \bar{y})$ are bounded. As a result, it follows from the continuity of $c(x, y)$ and $\rho_k \to \infty$ that

$$\limsup_{k\to\infty} \ [c(x_k, y_k^*(x_k))]_+^2 - [c(\bar{x}, \bar{y})]_+^2$$
$$\le \limsup_{k\to\infty} \ \left([c(x_k, y_k^*(x_k))]_+^2 - [c(x_k, \bar{y})]_+^2\right) + \limsup_{k\to\infty} \ \left([c(x_k, \bar{y})]_+^2 - [c(\bar{x}, \bar{y})]_+^2\right)$$
$$\le \limsup_{k\to\infty} \frac{2}{\rho_k}\left(f(x_k, y_k^*(x_k)) - f(x_k, \bar{y})\right) - \frac{\sigma_k}{\rho_k}\|y_k^*(x_k)\|^2 + \frac{\sigma_k}{\rho_k}\|\bar{y}\|^2$$
$$= 0,$$

which implies the desired result since $c(\bar{x}, \bar{y}) \le 0$ and $[c(x_k, y_k^*(x_k))]_+ \ge 0$. $\qquad\square$

**Lemma C.7.** *(Lemma 2.3) Let $\rho_k \to \infty$ and $\sigma_k \to 0$ as $k \to \infty$. Then, for any fixed $x \in X$,*

$$\limsup_{k\to\infty} \ \varphi_{\rho_k, \sigma_k}(x) \le \varphi(x).$$

*Proof.* Fix $x \in X$. We prove $\limsup_{k\to\infty} \varphi_k(x) \le \varphi(x)$ by contradiction. Suppose that there exists $\delta > 0$ such that

$$\limsup_{k\to\infty} \varphi_k(x) > \varphi(x) + \delta.$$

Then there exists a subsequence $\{k_i\}$ such that

$$\limsup_{i\to\infty} \varphi_{k_i}(x) > \varphi(x) + \delta.$$

Since $Y$ is compact, the sequence $\{y_{k_i}^*(x)\}$ is bounded. By passing to a further subsequence if necessary, it can be assumed that $y_{k_i}^*(x) \to \hat{y}$ for some $\hat{y} \in Y$. Consequently,

$$\limsup_{i\to\infty} \varphi_{k_i}(x) = \limsup_{i\to\infty}\left(f(x, y_{k_i}^*(x)) - \frac{\rho_{k_i}}{2}[c(x, y_{k_i}^*(x))]_+^2 - \frac{\sigma_{k_i}}{2}\|y_{k_i}^*(x)\|^2\right)$$
$$\le \limsup_{i\to\infty} f(x, y_{k_i}^*(x)) = f(x, \hat{y}), \tag{24}$$

where the inequality follows from the nonnegativity of the penalty terms. On the other hand, it follows from Lemma C.6 that $c(x, \hat{y}) \le 0$. Recalling that

$$\varphi(x) = \max_{y \in Y, \ c(x,y) \le 0} f(x, y),$$

one has $\varphi(x) \ge f(x, \hat{y})$, which contradicts (24). $\qquad\square$

**Lemma C.8.** *(Lemma 2.5) Suppose $\varphi$ is lower semi-continuous on $X$. Let $\rho_k \to \infty$ and $\sigma_k \to 0$ as $k \to \infty$. Then, for any sequence $\{x_k\} \subset X$ with $x_k \to \bar{x}$, we have:*

$$\liminf_{k \to \infty} \varphi_{\rho_k, \sigma_k}(x_k) \geq \varphi(\bar{x}). \tag{25}$$

*Proof.* For each $k$, let

$$y^*(x_k) \in \operatorname*{arg\,max}_{y \in Y,\ c(x_k, y) \leq 0} f(x_k, y),$$

which is well-defined since $Y$ is compact and $-f$ and $c$ are convex and continuous in $y$. By definition of $\varphi_k(x_k)$ and the optimality of $y_k^*(x_k)$, it holds that

$$\varphi_k(x_k) = f(x_k, y_k^*(x_k)) - \frac{\rho_k}{2}[c(x_k, y_k^*(x_k))]_+^2 - \frac{\sigma_k}{2}\|y_k^*(x_k)\|^2$$

$$\geq f(x_k, y^*(x_k)) - \frac{\rho_k}{2}[c(x_k, y^*(x_k))]_+^2 - \frac{\sigma_k}{2}\|y^*(x_k)\|^2.$$

Since $c(x_k, y^*(x_k)) \leq 0$, the penalty term vanishes, and thus

$$\varphi_k(x_k) \geq f(x_k, y^*(x_k)) - \frac{\sigma_k}{2}\|y^*(x_k)\|^2 = \varphi(x_k) - \frac{\sigma_k}{2}\|y^*(x_k)\|^2. \tag{26}$$

Taking $\liminf$ on both sides yields

$$\liminf_{k \to \infty} \varphi_k(x_k) \geq \liminf_{k \to \infty} \varphi(x_k).$$

Since $\varphi$ is lower semicontinuous, it follows that

$$\liminf_{k \to \infty} \varphi(x_k) \geq \varphi(\bar{x}),$$

which completes the proof. $\qquad\square$

### C.4. Proof for Theorem 2.6

**Lemma C.9.** *Suppose that $\varphi$ is lower semi-continuous on $X$. Let $\rho_k \to \infty$ and $\sigma_k \to 0$ as $k \to \infty$. Then, for any sequence $\{x_k\} \subset X$ with $x_k \to \bar{x}$, we have any accumulation point of the sequence $\{y_k^*(x_k)\}$ belongs to the set $\arg\max_{y \in Y}\{f(\bar{x}, y) \mid c(\bar{x}, y) \leq 0\}$.*

*Proof.* First, we show that $\Theta(x) := \{y \in Y : c(x, y) \leq 0\}$ is outer semi-continuous. Let $(x_j, y_j) \to (\tilde{x}, \tilde{y})$ with $y_j \in \Theta(x_j)$. Then $[c(x_j, y_j)]_+ = 0$ for all $j$. Taking $j \to \infty$ and using the continuity of $[c(\cdot, \cdot)]_+$, we obtain $[c(\tilde{x}, \tilde{y})]_+ = 0$, i.e., $\tilde{y} \in \Theta(\tilde{x})$.

Next, recall that $\varphi(x) := \max_{y \in \Theta(x)} f(x, y)$. For each $k$, pick $\hat{y}_k \in \arg\max_{y \in \Theta(x_k)} f(x_k, y)$, so that $\varphi(x_k) = f(x_k, \hat{y}_k)$. Since $Y$ is compact, $\{\hat{y}_k\}$ admits a convergent subsequence $\hat{y}_{k_i} \to \hat{y}$. By the outer semi-continuity of $\Theta$, we have $\hat{y} \in \Theta(\bar{x})$. By continuity of $f$,

$$\lim_{i \to \infty} f(x_{k_i}, \hat{y}_{k_i}) = f(\bar{x}, \hat{y}) \leq \sup_{y \in \Theta(\bar{x})} f(\bar{x}, y) = \varphi(\bar{x}).$$

As a result, we have $\limsup_{k \to \infty} \varphi(x_k) \leq \varphi(\bar{x})$, which implie that $\varphi$ is upper semicontinuous. Combined with the assumed lower semicontinuity, $\varphi$ is continuous on $X$, and hence $\varphi(x_k) \to \varphi(\bar{x})$. Thus, there must exist a sequence $y^*(x_k) \in \arg\max_{y \in \Theta(x_k)} f(x_k, y)$ such that $f(x_k, y^*(x_k)) \to \varphi(\bar{x})$. Recalling (26) and definition of $\varphi_k(x_k)$, we have

$$f(x_k, y_k^*(x_k)) \geq \varphi_k(x_k) \geq f(x_k, y^*(x_k)) - \frac{\sigma_k}{2}\|y^*(x_k)\|^2.$$

Take the limit as $k \to \infty$ on both side and use the fact that $y^*(x_k)$ are bounded:

$$\liminf_{k \to \infty} f(\bar{x}, y_k^*(x_k)) = \liminf_{k \to \infty} f(x_k, y_k^*(x_k)) \geq \lim_{k \to \infty} f(x_k, y^*(x_k)) - \frac{\sigma_k}{2}\|y^*(x_k)\|^2 = \varphi(\bar{x}). \tag{27}$$

Combining the results in Lemma C.6:

$$\limsup_{k\to\infty} [c(x_k, y_k^*(x_k))]_+ = \limsup_{k\to\infty} [c(\bar{x}, y_k^*(x_k))]_+ = 0.$$

This implies that any accumulation point of $y_k^*(x_k)$ is feasible and thus belongs to $\Theta(\bar{x})$. On the other hand, $\varphi(\bar{x}) = \max_{y\in\Theta(\bar{x})} f(\bar{x}, y)$, combining this with (27), we have any accumulation point of the sequence $\{y_k^*(x_k)\}$ belongs to $\arg\max_{y\in\Theta(\bar{x})} f(\bar{x}, y)$. This completes the proof.

$\square$

**Theorem C.10.** *(Theorem 2.6) Assume $\varphi(x)$ is lower semi-continuous on $X$. Let $\rho_k \to \infty$ and $\sigma_k \to 0$ as $k \to \infty$, and let $x_k \in \arg\min_{x\in X} \varphi_{\rho_k,\sigma_k}(x)$. Then, any accumulation point $\bar{x}$ of the sequence $\{x_k\}$ is an optimal solution to the MCC (1), i.e., $\bar{x} \in \arg\min_{x\in X}\varphi(x)$. Moreover, any accumulation point $\bar{y}$ of the sequence $\{y_{\rho_k,\sigma_k}^*(x_k)\}$ satisfies $\arg\max_{y\in Y}\{f(\bar{x}, y) \mid c(x, y) \le 0\}$.*

*Proof.* By applying (Rockafellar & Wets, 2009, Proposition 7.2), the epi-convergence of the sequence of functions $\{\varphi_{\rho_k,\sigma_k}\}$ to $\varphi$ on $X$ as $k \to \infty$ follows by Lemma 2.3 together with Lemma 2.5. Consequently, the subsequential convergence of minimizers follows by (Bonnans & Shapiro, 2013, Proposition 4.6). Furthermore, the convergence results for $\{y_{\rho_k,\sigma_k}^*(x_k)\}$ follows by Lemma C.9.

$\square$

## C.5. Proof for Theorem 2.9

**Theorem C.11.** *(Theorem 2.9) Let $\rho_k \to \infty$ and $\sigma_k \to 0$ as $k \to \infty$, and suppose $x_k$ is a stationary point of $\min_{x\in X} \varphi_{\rho_k,\sigma_k}(x)$, i.e., $0 \in \nabla\varphi_{\rho_k,\sigma_k}(x_k) + \mathcal{N}_X(x_k)$. Assume $x_k \to \bar{x}$ and let $\bar{y}$ be an accumulation point of $\{y_{\rho_k,\sigma_k}^*(x_k)\}$. If GPŁCQ holds at $(\bar{x}, \bar{y})$, then $(\bar{x}, \bar{y})$ is a stationary (KKT) point of the original MCC problem (1).*

*Proof.* First, we introduce $\lambda_k := \rho_k[c(x_k, y_k^*(x_k))]_+$ and prove that $\{\lambda_k\}$ are bounded.

Note that the GPŁCQ holds at $(\bar{x}, \bar{y})$, let $\beta, \delta, \nu$ be the constant in Definition 2.8 and $\beta_k := \min\{1, \frac{\beta}{\rho_k}\}$. Further, the optimality of $y_k^*(x^k)$ and stationarity of $x_k$ implies that

$$0 = \begin{bmatrix} x_k \\ y_k^*(x_k) \end{bmatrix} - \mathcal{P}_{X\times Y}\left(\begin{bmatrix} x_k \\ y_k^*(x_k) \end{bmatrix} - \beta_k \begin{bmatrix} \nabla_x\psi_k(x_k, y_k^*(x_k)) \\ -\nabla_y\psi_k(x_k, y_k^*(x_k)) \end{bmatrix}\right).$$

Recall that $p(x, y) = \frac{1}{2}\|[c(x, y)]_+\|^2$ and $\sigma_k \to 0$, by applying Cauchy-Schwarz inequality and non-expansiveness of projection operator, by denoting that $M := \max\{\sup_{x\in X, y\in Y}\|\nabla f(x, y)\|, \sup_{y\in Y}\|y\|\}$, we have that for sufficiently large $k$:

$$\frac{1}{\beta_k\rho_k}\left\|\begin{bmatrix} x_k \\ y_k^*(x_k) \end{bmatrix} - \mathcal{P}_{X\times Y}\left(\begin{bmatrix} x_k \\ y_k^*(x_k) \end{bmatrix} - \beta_k\rho_k \begin{bmatrix} \nabla_x p(x_k, y_k^*(x_k)) \\ -\nabla_y p(x_k, y_k^*(x_k)) \end{bmatrix}\right)\right\|$$

$$\le \frac{1}{\beta_k\rho_k}\left\|\begin{bmatrix} x_k \\ y_k^*(x_k) \end{bmatrix} - \mathcal{P}_{X\times Y}\left(\begin{bmatrix} x_k \\ y_k^*(x_k) \end{bmatrix} - \beta_k\rho_k \begin{bmatrix} \nabla_x p(x_k, y_k^*(x_k)) \\ -\nabla_y p(x_k, y_k^*(x_k)) \end{bmatrix}\right)\right.$$

$$\left. - \left(\begin{bmatrix} x_k \\ y_k^*(x_k) \end{bmatrix} - \mathcal{P}_{X\times Y}\left(\begin{bmatrix} x_k \\ y_k^*(x_k) \end{bmatrix} - \beta_k \begin{bmatrix} \nabla_x\psi_k(x_k, y_k^*(x_k)) \\ -\nabla_y\psi_k(x_k, y_k^*(x_k)) \end{bmatrix}\right)\right)\right\|$$

$$+ \frac{1}{\beta_k\rho_k}\left\|\begin{bmatrix} x_k \\ y_k^*(x_k) \end{bmatrix} - \mathcal{P}_{X\times Y}\left(\begin{bmatrix} x_k \\ y_k^*(x_k) \end{bmatrix} - \beta_k \begin{bmatrix} \nabla_x\psi_k(x_k, y_k^*(x_k)) \\ -\nabla_y\psi_k(x_k, y_k^*(x_k)) \end{bmatrix}\right)\right\|$$

$$\le \frac{1}{\rho_k}\left\|\begin{bmatrix} -\rho_k\nabla_x p(x_k, y_k^*(x_k)) - \nabla_x\psi_k(x_k, y_k^*(x_k)) \\ \rho_k\nabla_y p(x_k, y_k^*(x_k)) + \nabla_y\psi_k(x_k, y_k^*(x_k)) \end{bmatrix}\right\|$$

$$\le \frac{1}{\rho_k}\left\|\begin{bmatrix} \nabla_x f(x_k, y_k^*(x_k)) \\ -\nabla_y f(x_k, y_k^*(x_k)) + \sigma_k y_k^*(x_k) \end{bmatrix}\right\|$$

$$\le \frac{3}{\rho_k} M,$$

where the first inequality follows from the triangle inequality and the second inequality follows from the non-expansiveness of projection operator. Note that for all $\beta' \leq \beta$, the gradient mapping satisfies (see, e.g., (Beck, 2017, 10.9)):

$$\frac{1}{\beta} \left\| \begin{bmatrix} x_k \\ y_k^*(x_k) \end{bmatrix} - \mathcal{P}_{X \times Y} \left( \begin{bmatrix} x_k \\ y_k^*(x_k) \end{bmatrix} - \beta \begin{bmatrix} \nabla_x p(x_k, y_k^*(x_k)) \\ -\nabla_y p(x_k, y_k^*(x_k)) \end{bmatrix} \right) \right\|$$

$$\leq \frac{1}{\beta'} \left\| \begin{bmatrix} x_k \\ y_k^*(x_k) \end{bmatrix} - \mathcal{P}_{X \times Y} \left( \begin{bmatrix} x_k \\ y_k^*(x_k) \end{bmatrix} - \beta' \begin{bmatrix} \nabla_x p(x_k, y_k^*(x_k)) \\ -\nabla_y p(x_k, y_k^*(x_k)) \end{bmatrix} \right) \right\|.$$

Since $(x_k, y_k^*(x_k)) \to (\bar{x}, \bar{y})$, recall that $\beta_k := \min\{1, \frac{\beta}{\rho_k}\}$, for sufficiently large $k$, we have $x^k \in B_\delta(\bar{x}), y_k^*(x_k) \in B_\delta(\bar{y}), \beta_k \rho_k \leq \beta$. As a result, the following inequality holds:

$$\|\lambda_k\| = \rho_k \|[c(x_k, y_k^*(x_k))]_+\| = \sqrt{2} \rho_k \sqrt{\|p(x_k, y_k^*(x_k))\|}$$

$$\leq \sqrt{2} v \rho_k \frac{1}{\beta} \left\| \begin{bmatrix} x_k \\ y_k^*(x_k) \end{bmatrix} - \mathcal{P}_{X \times Y} \left( \begin{bmatrix} x_k \\ y_k^*(x_k) \end{bmatrix} - \beta \begin{bmatrix} \nabla_x p(x_k, y_k^*(x_k)) \\ -\nabla_y p(x_k, y_k^*(x_k)) \end{bmatrix} \right) \right\|$$

$$\leq \sqrt{2} v \rho_k \frac{1}{\beta_k \rho_k} \left\| \begin{bmatrix} x_k \\ y_k^*(x_k) \end{bmatrix} - \mathcal{P}_{X \times Y} \left( \begin{bmatrix} x_k \\ y_k^*(x_k) \end{bmatrix} - \beta_k \rho_k \begin{bmatrix} \nabla_x p(x_k, y_k^*(x_k)) \\ -\nabla_y p(x_k, y_k^*(x_k)) \end{bmatrix} \right) \right\|$$

$$\leq 3\sqrt{2} v M.$$

Hence, the sequence $\{\lambda_k\}$ is bounded. Next, we show that any accumulation point of the multipliers $\lambda_k$ corresponds to a multiplier $\bar{\lambda}$ for $(\bar{x}, \bar{y})$. We assume without loss of generality that $\lambda_k \to \bar{\lambda}$. Since $x_k$ is the stationary point, combining the optimality condition of $y_k^*(x_k)$, we have that

$$-\nabla_x f(x_k, y_k^*(x_k)) + \lambda_k^\top \nabla_x c(x_k, y_k^*(x_k)) \in N_X(x_k),$$

$$\nabla_y f(x_k, y_k^*(x_k)) - \lambda_k^\top \nabla_y c(x_k, y_k^*(x_k)) - \sigma_k y_k^*(x_k) \in N_Y(y_k^*(x_k)).$$

As $X$ and $Y$ are convex compact sets, $N_X(\cdot)$ and $N_Y(\cdot)$ are outer semi-continuous (Rockafellar & Wets, 2009, Proposition 6.6), which implies that

$$-\nabla_x f(\bar{x}, \bar{y}) + \bar{\lambda}^\top \nabla_x c(\bar{x}, \bar{y}) = \lim_{k \to \infty} -\nabla_x f(x_k, y_k^*(x_k)) + \lambda_k^\top \nabla_x c(x_k, y_k^*(x_k)) \in N_X(\bar{x}),$$

$$\nabla_y f(\bar{x}, \bar{y}) - \bar{\lambda}^\top \nabla_y c(\bar{x}, \bar{y}) = \lim_{k \to \infty} \nabla_y f(x_k, y_k^*(x_k)) - \lambda_k^\top \nabla_y c(x_k, y_k^*(x_k)) - \sigma_k y_k^*(x_k) \in N_Y(\bar{y}).$$

Further, $c(\bar{x}, \bar{y}) \leq 0$ follows immediately by Lemma C.6. For the complementarity, if $c_i(\bar{x}, \bar{y}) < 0$, then we have $c_i(x_k, y_k^*(x_k)) < 0$ for sufficiently large $k$. As a result $[\bar{\lambda}]_i = \lim_{k \to \infty} [\lambda_k]_i = \lim_{k \to \infty} \rho_k [c_i(x_k, y_k^*(x_k))]_+ = 0$. If otherwise $c_i(\bar{x}, \bar{y}) = 0$, $[\bar{\lambda}]_i c_i(\bar{x}, \bar{y}) = 0$ automatically. As a result, we have $[\bar{\lambda}]_i c_i(\bar{x}, \bar{y}) = 0$ for all $i$. Thus, we can conclude that $(\bar{x}, \bar{y})$ is a KKT point defined in (7). $\square$

# D. Proof for Section 3.2

In this section, we present a detailed proof of the convergence results stated in Section 3.2. To streamline the notation in what follows, we let $L_f$ and $L_c$ denote the Lipschitz constants of $\nabla f$ and $\nabla c$ on $X \times Y$, respectively. To avoid notational redundancy, we treat $L_f$ and $L_F$ aas a single constant and consistently use $L_f$ throughout the proof. Since $X$ and $Y$ are bounded, we define

$$M := \max \left\{ \sup_{x \in X, y \in Y} \|\nabla f(x, y)\|, \sup_{x \in X, y \in Y} \|\nabla c(x, y)\|, \sup_{x \in X, y \in Y} \|f(x, y)\|, \sup_{x \in X, y \in Y} \|c(x, y)\|, \sup_{x \in X} \|x\|, \sup_{y \in Y} \|y\| \right\}.$$

Finally, for conciseness we adopt the shorthand $\psi_k(x, y) := \psi_{\rho_k, \sigma_k}(x, y)$, $\varphi_k(x) := \varphi_{\rho_k, \sigma_k}(x)$, and $y_k^*(x) := y_{\rho_k, \sigma_k}^*(x)$. We first establish the smoothness of $\psi_k(x, y)$ and boundedness of $\varphi_k(x)$.

**Lemma D.1.** *(Lemma 3.3) Let $\rho_k \to \infty$ and $\sigma_k \to 0$ as $k \to \infty$, $\nabla \psi_k(x, y)$ is $\bar{L}_k$-Lipschitz continuous on $X \times Y$ with constant $\bar{L}_k = \mathcal{O}(\rho_k)$. Furthermore, $\varphi_k(x)$ is uniformly lower bounded with respect to $k$ on $X$ by a constant $\underline{\varphi}$.*

*Proof.* As $\nabla \psi_k(x,y) = \nabla f(x,y) - \rho_k[c(x,y)]_+ \nabla c(x,y) - \sigma_k y$, by applying Cauchy-Schwarz inequality, we have that

$$\|\nabla \psi_k(x_2, y_2) - \nabla \psi_k(x_1, y_1)\|$$

$$\leq \|\nabla f(x_2, y_2) - \nabla f(x_1, y_1)\| + \rho_k \| \sum_{i=1}^{p} [c_i(x_2, y_2)]_+ \nabla c_i(x_2, y_2) - \sum_{i=1}^{p} [c_i(x_1, y_1)]_+ \nabla c_i(x_1, y_1)\| + \sigma_k \|y_2 - y_1\|$$

$$\leq (L_f + \sigma_k)\|(x_2, y_2) - (x_1, y_1)\| + \rho_k \sum_{i=1}^{p} \|[c_i(x_2, y_2)]_+ \nabla c_i(x_2, y_2) - [c_i(x_1, y_1)]_+ \nabla c_i(x_2, y_2)\|$$

$$+ \rho_k \sum_{i=1}^{p} \|[c_i(x_1, y_1)]_+ \nabla c_i(x_2, y_2) - [c_i(x_1, y_1)]_+ \nabla c_i(x_1, y_1)\|$$

$$\leq (L_f + \sigma_k)\|(x_2, y_2) - (x_1, y_1)\| + \rho_k p M \|c(x_2, y_2) - c(x_1, y_1)\| + \rho_k p M \|\nabla c(x_2, y_2) - \nabla c(x_1, y_1)\|$$

$$\leq (L_f + \rho_k p M L_c + \sigma_k)\|(x_2, y_2) - (x_1, y_1)\| + \rho_k p M \|\nabla c(\bar{x}, \bar{y})[(x_2, y_2) - (x_1, y_1)]\|$$

$$\leq (L_f + \rho_k p M L_c + \rho_k p M^2 + \sigma_k)\|(x_2, y_2) - (x_1, y_1)\|$$

where the fourth inequality follows from Lagrange mean value theorem and $(\bar{x}, \bar{y}) = a(x_2, y_2) + (1 - a)(x_1, y_1)$ for some $0 \leq a \leq 1$. This implies that $\nabla \psi_k(x,y)$ is $\bar{L}_k$-Lipschitiz continuous with $\bar{L}_k = L_f + \rho_k p M L_c + \rho_k p M^2 + \sigma_k$. Furthermore, combining the boundedness of $X, Y$, we have

$$\varphi_k(x) = \max_{y \in Y} \psi_k(x,y) \geq \max_{y \in Y, c(x,y) \leq 0} \psi_k(x^k, y) = \max_{y \in Y, c(x,y) \leq 0} f(x,y) - \frac{\sigma_k}{2}\|y\|^2 \geq \inf_{x \in X, y \in Y} f(x,y) - \frac{\sigma_k}{2}M^2.$$

Thus, $\varphi_k(x)$ are lower bounded as $\lim_{k \to \infty} \sigma_k = 0$. $\qquad \square$

We denote by $\bar{L}_k$ the Lipschitz constant of the gradient of $\psi_k(x,y)$ on $X \times Y$.

### D.1. Framework for Convergence Analysis

In this subsection, we present the overall convergence analysis framework for SPACO. Our analysis is built upon a time-varying merit value function defined as

$$V_k := a_k\big(\varphi_k(x^k) - \underline{\varphi}\big) + b_k\|y^k - y_k^*(x^k)\|^2 + c_k\|e_x^k\|^2 + d_k\|x^k - x^{k-1}\|^2, \tag{28}$$

where $\{a_k, b_k, c_k, d_k\}$ are positive, iteration-dependent coefficients.

Our goal is to establish a recursive inequality of the following form:

$$\gamma_k^1 \mathbb{E}\big[\|\mathcal{G}_k(x^k)\|^2\big] + \gamma_k^2 \mathbb{E}\big[\|y^k - y_k^*(x^k)\|^2\big] \leq \mathbb{E}[V_k] - \mathbb{E}[V_{k+1}] + \gamma_k^3 \delta^2 + \zeta_k, \tag{29}$$

where $\gamma_k^1, \gamma_k^2, \gamma_k^3$ are nonnegative coefficients, and $\{\zeta_k\}$ is a nonnegative summable sequence satisfying $\sum_{k=1}^{\infty} \zeta_k = C_0 < \infty$.

Summing the above inequality from $k = 1$ to $K$ and dividing both sides by $K$, we obtain

$$\frac{1}{K} \sum_{k=1}^{K} \gamma_k^1 \mathbb{E}\big[\|\mathcal{G}_k(x^k)\|^2\big] + \frac{1}{K} \sum_{k=1}^{K} \gamma_k^2 \mathbb{E}\big[\|y^k - y_k^*(x^k)\|^2\big] \leq \frac{V_0}{K} + \frac{1}{K} \sum_{k=1}^{K} \gamma_k^3 \delta^2 + \frac{C_0}{K}.$$

This inequality directly yields non-asymptotic convergence rates for the stationarity measure $\mathbb{E}[\|\mathcal{G}_k(x^k)\|^2]$ as well as the tracking error $\mathbb{E}[\|y^k - y_k^*(x^k)\|^2]$. To derive (29), we divide our proof into four steps:
**Step 1: Controlling the estimation error of $y^k$.**
We first establish a sufficient descent property for the update of $y^{k+1}$, which allows us to quantify the estimation accuracy of $y^{k+1}$ relative to the implicit solution $y_k^*(x^k)$. Specifically, we show that

$$\mathbb{E}[\|y^{k+1} - y_k^*(x^k)\|^2 \mid \mathcal{F}_k] \leq (1 - \sigma_k \beta_k)\|y^k - y_k^*(x^k)\|^2 + \beta_k^2 \delta^2. \tag{30}$$

We further characterize the evolution of the estimation error along the optimization trajectory by accounting for the changes in both the primal variable and the problem parameters. In particular, the following bound holds:

$$
\begin{aligned}
\mathbb{E}[\|y^{k+1} - y_{k+1}^*(x^{k+1})\|^2 \mid \mathcal{F}_k] \leq & (1 - \frac{1}{2}\beta_k\sigma_k)\|y^k - y_k^*(x^k)\|^2 + (1 + \frac{1}{2}\beta_k\sigma_k)\beta_k^2\delta^2 \\
& + 2(1 + \frac{2}{\beta_k\sigma_k})\frac{\bar{L}_k^2}{\sigma_k^2}\mathbb{E}[\|x^k - x^{k+1}\|^2 \mid \mathcal{F}_k] \\
& + 2(1 + \frac{2}{\beta_k\sigma_k})\left(\frac{2(\rho_{k+1} - \rho_k)^2}{\sigma_k^2}M^4 + \frac{2(\sigma_k - \sigma_{k+1})^2}{\sigma_k^2}M^2\right).
\end{aligned}
\tag{31}
$$

The corresponding results of this step are formally provided in Lemma D.2 and Lemma D.7.

**Step 2: Upper bounding the descent of the approximation $\varphi_k(x)$.**
Next, by exploiting the Lipschitz continuity of the gradient of $\varphi_k$ together with the error bound of the inexact gradient estimator, we control the descent behavior of the approximate value function. Specifically, we obtain

$$
\begin{aligned}
\mathbb{E}[\varphi_{k+1}(x^{k+1}) \mid \mathcal{F}_k] - \varphi_k(x^k) + &\left(\frac{1}{4\alpha_k} - \frac{L_{\varphi_k}}{2}\right)\mathbb{E}[\|x^{k+1} - x^k\|^2 \mid \mathcal{F}_k] \\
\leq & \frac{\alpha_k}{2}\bar{L}_k^2\|y^k - y_k^*(x^k)\|^2 + \frac{\alpha_k}{2}\bar{L}_k^2\beta_k^2\delta^2 + \alpha_k\mathbb{E}[\|e_x^k\|^2 \mid \mathcal{F}_k] + (\sigma_k - \sigma_{k+1})M^2.
\end{aligned}
\tag{32}
$$

where the detailed results are given in Lemma D.8. Note that the stationarity measure $\|\mathcal{G}_k(x^k)\|^2$ can be related to the update magnitude $\frac{1}{\alpha_k^2}\|x^{k+1} - x^k\|^2$. However, in the above inequality, the descent term $\frac{1}{\alpha_k^2}\|x^{k+1} - x^k\|^2$ is coupled with the variance term $\mathbb{E}[\|e_x^k\|^2 \mid \mathcal{F}_k]$. Therefore, to close the analysis, it is necessary to further control the evolution of the estimator variance.

**Step 3: Upper bounding the variance of the estimator $d_x^k$.**
Through a careful analysis, we establish the following recursive bound on the variance of the gradient estimator:

$$
\begin{aligned}
\mathbb{E}[\|e_x^k\|^2 \mid \mathcal{F}_k] \leq & (1 - \eta_k)^2\|e_x^{k-1}\|^2 + 2\eta_k^2\delta^2 + 18\bar{L}_k^2\left(\|x^k - x^{k-1}\|^2 + 3\beta_k^2\delta^2 + 3\beta_k^2\bar{L}_k^2\|y^k - y_k^*(x^k)\|^2 \right. \\
& \left. + 3\beta_k^2(\rho_k M + 1 + \sigma_k M)^2M^2\right),
\end{aligned}
\tag{33}
$$

where the results are present in Lemma D.9.

**Step 4: Choosing coefficients to ensure merit value descent.**
Combining the results of the previous three steps, and by selecting the algorithmic parameters and merit coefficients appropriately, we obtain the following descent property of the merit value function:

$$
\mathbb{E}[V_{k+1} \mid \mathcal{F}_k] - V_k \leq -\frac{a_k\alpha_k}{24}\mathbb{E}[\|\mathcal{G}_k(x^k)\|^2 \mid \mathcal{F}_k] - \frac{b_k\beta_k\sigma_k}{4}\|y^k - y_k^*(x^k)\|^2 + \zeta_k + 2b_k\beta_k^2\delta^2,
\tag{34}
$$

where the proof is formally stated in Proposition 3.4. With these results, we finally establish the non-asymptotic convergence guarantees of the proposed algorithm, as formalized in Theorems 3.5 and 3.6.

### D.2. Auxiliary Lemmas

**Lemma D.2.** *Suppose that the step-size sequence $\{\beta_k\}$ satisfies $0 < \beta_k \leq \frac{1}{\bar{L}_k}$ for each $k$. Let $\{(x^k, y^k)\}$ be the sequence generated by Algorithm 1. Then the iterates $y^k$ and $y^{k+1}$ satisfy*

$$
\mathbb{E}[\|y^{k+1} - y_k^*(x^k)\|^2 \mid \mathcal{F}_k] \leq (1 - \sigma_k\beta_k)\|y^k - y_k^*(x^k)\|^2 + \beta_k^2\delta^2.
\tag{35}
$$

*Proof.* We first relate the distance between successive iterates and the current lower-level maximizer using the non-expansiveness of the projection operator. Specifically, since the projection onto a closed convex set is non-expansive, we have

$$
\begin{aligned}
\mathbb{E}[\|y^{k+1} - y_k^*(x^k)\|^2 \mid \mathcal{F}_k] &= \mathbb{E}[\|\mathcal{P}_Y(y^k + \beta_k d_y^k) - \mathcal{P}_Y(y_k^*(x^k))\|^2 \mid \mathcal{F}_k] \\
&\leq \mathbb{E}[\|y^k + \beta_k d_y^k - y_k^*(x^k)\|^2 \mid \mathcal{F}_k].
\end{aligned}
$$

Expanding the squared norm yields

$$\|y^k - y_k^*(x^k)\|^2 + 2\beta_k \mathbb{E}[\langle d_y^k, y^k - y_k^*(x^k)\rangle \mid \mathcal{F}_k] + \beta_k^2 \mathbb{E}[\|d_y^k\|^2 \mid \mathcal{F}_k].$$

Next, we decompose the stochastic gradient term into its mean and variance components. Using the unbiasedness of the stochastic estimator $d_y^k$ and the bounded variance assumption, we obtain

$$\mathbb{E}[\|d_y^k\|^2 \mid \mathcal{F}_k] = \mathbb{E}[\|d_y^k - \nabla_y \psi_k(x^k, y^k) + \nabla_y \psi_k(x^k, y^k)\|^2 \mid \mathcal{F}_k] \leq \|\nabla_y \psi_k(x^k, y^k)\|^2 + \delta^2,$$

and

$$\mathbb{E}[\langle d_y^k, y^k - y_k^*(x^k)\rangle \mid \mathcal{F}_k] = \langle \mathbb{E}[d_y^k \mid \mathcal{F}_k], y^k - y_k^*(x^k)\rangle = \langle \nabla_y \psi_k(x^k, y^k), y^k - y_k^*(x^k)\rangle.$$

Consequently,

$$\mathbb{E}[\|y^{k+1} - y_k^*(x^k)\|^2 \mid \mathcal{F}_k] \leq \|y^k - y_k^*(x^k)\|^2 + 2\beta_k \langle \nabla_y \psi_k(x^k, y^k), y^k - y_k^*(x^k)\rangle + \beta_k^2 \|\nabla_y \psi_k(x^k, y^k)\|^2 + \beta_k^2 \delta^2.$$

We now control the inner-product term using the strong concavity of the lower-level objective. Since $\psi_k(x^k, \cdot)$ is $\sigma_k$-strongly concave, the following inequality holds:

$$\langle \nabla_y \psi_k(x^k, y^k), y^k - y_k^*(x^k)\rangle \leq \psi_k(x^k, y^k) - \psi_k(x^k, y_k^*(x^k)) - \frac{\sigma_k}{2}\|y^k - y_k^*(x^k)\|^2.$$

Multiplying both sides by $2\beta_k$ gives

$$2\beta_k \langle \nabla_y \psi_k(x^k, y^k), y^k - y_k^*(x^k)\rangle \leq 2\beta_k(\psi_k(x^k, y^k) - \psi_k(x^k, y_k^*(x^k))) - \beta_k \sigma_k \|y^k - y_k^*(x^k)\|^2.$$

It remains to control the squared gradient norm. By Lemma 3.3, the function $-\psi_k(x^k, \cdot)$ is convex and $\bar{L}_k$-smooth. Applying the standard descent inequality for smooth and convex functions yields

$$-\psi_k(x^k, y_k^*(x^k)) \leq -\psi_k(x^k, y_k - \frac{1}{\bar{L}_k}\nabla_y \psi_k(x^k, y^k)) \leq -\psi_k(x^k, y^k) - \frac{1}{2\bar{L}_k}\|\nabla_y \psi_k(x^k, y^k)\|^2.$$

Rearranging the above inequality and multiplying both sides by $\beta_k^2$, we obtain

$$\beta_k^2 \|\nabla_y \psi_k(x^k, y^k)\|^2 \leq 2\bar{L}_k \beta_k^2 (\psi_k(x^k, y_k^*(x^k)) - \psi_k(x^k, y^k)).$$

Finally, combining all the above bounds and using the step-size condition $\beta_k \leq 1/\bar{L}_k$, we arrive at

$$\mathbb{E}[\|y^{k+1} - y_k^*(x^k)\|^2 \mid \mathcal{F}_k] \leq (1 - \beta_k \sigma_k)\|y^k - y_k^*(x^k)\|^2 + (2\beta_k - 2\bar{L}_k \beta_k^2)(\psi_k(x^k, y^k) - \psi_k(x^k, y_k^*(x^k))] + \beta_k^2 \delta^2$$
$$\leq (1 - \beta_k \sigma_k)\|y^k - y_k^*(x^k)\|^2 + \beta_k^2 \delta^2,$$

which completes the proof. $\square$

The following lemma characterize the Lipschitz continuity of $y_k^*(\cdot)$, which is similar to (Liu et al., 2023, Lemma D.2).

**Lemma D.3.** *For any $x, x' \in X$, the corresponding points $y_k^*(x)$ and $y_k^*(x')$ satisfy:*

$$\|y_k^*(x') - y_k^*(x)\| \leq \frac{\bar{L}_k}{\sigma_k}\|x - x'\| \tag{36}$$

*Proof.* First, as $y_k^*(x)$ and $y_k^*(x')$ are the solution to problem $\max_{y \in Y} \psi_k(x, y)$, and $\max_{y \in Y} \psi_k(x, y)$, respectively, it follows from the first-order optimality conditions that :

$$0 \in -\nabla_y \psi_k(x, y_k^*(x)) + \mathcal{N}_Y(y_k^*(x)), \tag{37}$$

and

$$0 \in -\nabla_y \psi_k(x', y_k^*(x')) + \mathcal{N}_Y(y_k^*(x'))$$

As a result, we have

$$-\nabla_y \psi_k(x, y_k^*(x')) + \nabla_y \psi_k(x', y_k^*(x')) \in -\nabla_y \psi_k(x, y_k^*(x')) + \mathcal{N}_Y(y_k^*(x')). \tag{38}$$

Note that the normal cone $\mathcal{N}_Y$ satisfies the monotonicity. That is, for all $y_1, y_2 \in Y$, $v_1 \in \mathcal{N}_Y(y_1), v_2 \in \mathcal{N}_Y(y_2)$, the following statement holds:

$$\langle v_1 - v_2, y_1 - y_2 \rangle \ge 0.$$

By applying the $\sigma_k$-strongly concaveness of $\psi_k$ with respect to $y$, along with the monotonicity of the normal cone $\mathcal{N}_Y$, (37) and (38), we have the following inequality:

$$
\begin{aligned}
\sigma_k \|y_k^*(x') &- y_k^*(x)\|^2 \\
&\le \langle \nabla_y \psi_k(x, y_k^*(x')) - \nabla_y \psi_k(x', y_k^*(x')), y_k^*(x') - y_k^*(x) \rangle \\
&\le \|\nabla_y \psi_k(x, y_k^*(x')) - \nabla_y \psi_k(x', y_k^*(x'))\| \|y_k^*(x') - y_k^*(x)\| \\
&\le \bar{L}_k \|x - x'\| \|y_k^*(x') - y_k^*(x)\|
\end{aligned}
$$

where the last equality follows from $\bar{L}_k$ smoothness of $\psi_k$ from Lemma 3.3. (36) follows immediately by rearranging the items. $\qquad \square$

The following lemma characterizes the smoothness of $y_{k+1}^*(x)$ with respect to $\rho_k$ and $\sigma_k$.

**Lemma D.4.** *Let $\{\rho_k\}$ and $\{\sigma_k\}$ be sequences such that $\rho_{k+1} \ge \rho_k > 0$ and $\sigma_k \ge \sigma_{k+1} > 0$. Then, for any fixed $x \in X$, we have*

$$\|y_{k+1}^*(x) - y_k^*(x)\| \le \frac{\rho_{k+1} - \rho_k}{\sigma_k} M^2 + \frac{\sigma_k - \sigma_{k+1}}{\sigma_k} M. \tag{39}$$

*Proof.* The proof follows the same line as Lemma D.3. By the first-order optimality conditions of the lower-level problems, we have

$$0 \in -\nabla_y \psi_k(x, y_k^*(x)) + \mathcal{N}_Y(y_k^*(x)), \tag{40}$$

and

$$0 \in -\nabla_y \psi_{k+1}(x, y_{k+1}^*(x)) + \mathcal{N}_Y(y_{k+1}^*(x)).$$

We first quantify the change in the gradient of the lower-level objective induced by the update of the penalty parameters. Expanding the difference between the gradients of $\psi_k$ and $\psi_{k+1}$ at $y_{k+1}^*(x)$ yields

$$
\begin{aligned}
&- \nabla_y \psi_{k+1}(x, y_{k+1}^*(x)) + \nabla_y \psi_k(x, y_{k+1}^*(x)) \\
&= (\rho_{k+1} - \rho_k)[c(x, y_{k+1}^*(x))]_+ \nabla_y c(x, y_{k+1}^*(x)) + (\sigma_{k+1} - \sigma_k) y_{k+1}^*(x).
\end{aligned}
$$

Recalling that $\max\{\|c(x,y)\|, \|\nabla_y c(x,y)\|, \|y\|\} \le M$ for all $(x, y) \in X \times Y$, we obtain the following bound:

$$\|\nabla_y \psi_{k+1}(x, y_{k+1}^*(x)) - \nabla_y \psi_k(x, y_{k+1}^*(x))\| \le (\rho_{k+1} - \rho_k) M^2 + (\sigma_k - \sigma_{k+1}) M. \tag{41}$$

Next, using the optimality condition (40), together with the $\sigma_k$-strong concavity of $\psi_k$ with respect to $y$ and the monotonicity of the normal cone $\mathcal{N}_Y$, we obtain

$$
\begin{aligned}
\sigma_k \|y_{k+1}^*(x) - y_k^*(x)\|^2 &\le \langle \nabla_y \psi_{k+1}(x, y_{k+1}^*(x)) - \nabla_y \psi_k(x, y_{k+1}^*(x)), \, y_{k+1}^*(x) - y_k^*(x) \rangle \\
&\le \|\nabla_y \psi_{k+1}(x, y_{k+1}^*(x)) - \nabla_y \psi_k(x, y_{k+1}^*(x))\| \|y_{k+1}^*(x) - y_k^*(x)\|.
\end{aligned}
$$

Substituting (41) into the above inequality and rearranging terms yields the desired result. $\qquad \square$

The following lemma characterizes the gradient Lipschitz continuity of $\varphi_k$, and is analogous to (Liu et al., 2023, Lemma D.3).

**Lemma D.5.** *Let $\{\rho_k\}$ and $\{\sigma_k\}$ be sequences such that $\rho_k, \sigma_k > 0$. Then, for any $x, x' \in X$, we have*

$$\|\nabla\varphi_k(x') - \nabla\varphi_k(x)\| \leq L_{\varphi_k}\|x' - x\|, \tag{42}$$

*where $L_{\varphi_k} := \frac{(L_f + \rho_k M L_c + \rho_k M^2)(\bar{L}_k + \sigma_k)}{\sigma_k}$.*

*Proof.* Recall that $\nabla\varphi_k(x) = \nabla_x\psi_k(x, y_k^*(x))$. We bound the difference of $\nabla\varphi_k$ by separating the effects of the smoothness of $\psi_k$ in $(x, y)$ and the variation of the inner solution mapping $y_k^*(\cdot)$.

$$
\begin{aligned}
\|\nabla\varphi_k(x) - \nabla\varphi_k(x')\| &= \|\nabla_x\psi_k(x, y_k^*(x)) - \nabla_x\psi_k(x', y_k^*(x'))\| \\
&\leq \|\nabla_x f(x, y_k^*(x)) - \nabla_x f(x', y_k^*(x'))\| \\
&\quad + \rho_k\|[c(x, y_k^*(x))]_+\nabla_x c(x, y_k^*(x)) - [c(x', y_k^*(x'))]_+\nabla_x c(x', y_k^*(x'))\| \\
&\leq L_f(\|x - x'\| + \|y_k^*(x) - y_k^*(x')\|) \\
&\quad + \rho_k\|[c(x, y_k^*(x))]_+[\nabla_x c(x, y_k^*(x)) - \nabla_x c(x', y_k^*(x'))]\| \\
&\quad + \rho_k\|([c(x, y_k^*(x))]_+ - [c(x', y_k^*(x'))]_+)\nabla_x c(x', y_k^*(x'))\| \\
&\leq (L_f + \rho_k M L_c)(\|x - x'\| + \|y_k^*(x) - y_k^*(x')\|) \\
&\quad + \rho_k M\|c(x, y_k^*(x)) - c(x', y_k^*(x'))\| \\
&\leq (L_f + \rho_k M L_c + \rho_k M^2)(\|x - x'\| + \|y_k^*(x) - y_k^*(x')\|).
\end{aligned}
\tag{43}
$$

Finally, invoking the Lipschitz continuity of the solution mapping $x \mapsto y_k^*(x)$ from Lemma D.3, we obtain

$$\|\nabla\varphi_k(x) - \nabla\varphi_k(x')\| \leq \frac{(L_f + \rho_k M L_c + \rho_k M^2)(\bar{L}_k + \sigma_k)}{\sigma_k}\|x - x'\|,$$

which completes the proof. $\qquad\square$

**Lemma D.6.** *Let $\{\rho_k\}$ and $\{\sigma_k\}$ be sequences such that $\rho_{k+1} \geq \rho_k > 0$, $\sigma_k \geq \sigma_{k+1} > 0$. Then, for any $x \in X$, we have*

$$\varphi_{k+1}(x) - \varphi_k(x) \leq (\sigma_k - \sigma_{k+1})M^2. \tag{44}$$

*Proof.* Recall the definition of $\varphi_k(x)$:

$$\varphi_k(x) := \max_{y \in Y}\psi_k(x, y) \geq \psi_k(x, y_{k+1}^*(x)).$$

Thus, we have the following inequality:

$$
\begin{aligned}
\varphi_{k+1}(x) - \varphi_k(x) &\leq \psi_{k+1}(x, y_{k+1}^*(x)) - \psi_k(x, y_{k+1}^*(x)) \\
&= -(\rho_{k+1} - \rho_k)[c(x, y_{k+1}^*(x))]_+^2 + (\sigma_k - \sigma_{k+1})\|y_{k+1}^*(x)\|^2 \\
&\leq (\sigma_k - \sigma_{k+1})M^2,
\end{aligned}
$$

which completes the proof. $\qquad\square$

**Lemma D.7.** *Let $\{\rho_k\}$ and $\{\sigma_k\}$ be sequences such that $\rho_{k+1} \geq \rho_k > 0$, $\sigma_k \geq \sigma_{k+1} > 0$. Suppose the step-size sequence $\{\beta_k\}$ satisfies $0 < \beta_k \leq \frac{1}{\bar{L}_k}$ for each $k$. Then, the following inequality holds:*

$$
\begin{aligned}
\mathbb{E}[\|y^{k+1} - y_{k+1}^*(x^{k+1})\|^2 \mid \mathcal{F}_k] \leq{} & (1 - \tfrac{1}{2}\beta_k\sigma_k)\|y^k - y_k^*(x^k)\|^2 + (1 + \tfrac{1}{2}\beta_k\sigma_k)\beta_k^2\delta^2 \\
& + 2(1 + \frac{2}{\beta_k\sigma_k})\frac{\bar{L}_k^2}{\sigma_k^2}\mathbb{E}[\|x^{k+1} - x^k\|^2 \mid \mathcal{F}_k] \\
& + 2(1 + \frac{2}{\beta_k\sigma_k})\left(\frac{2(\rho_{k+1} - \rho_k)^2}{\sigma_k^2}M^4 + \frac{2(\sigma_k - \sigma_{k+1})^2}{\sigma_k^2}M^2\right).
\end{aligned}
\tag{45}
$$

*Proof.* First, by applying Cauchy Schwarz inequality, for any $\delta > 0$, we have:

$$\mathbb{E}[\|y^{k+1} - y^*_{k+1}(x^{k+1})\|^2 \mid \mathcal{F}_k] \leq (1 + \hat{\delta})\mathbb{E}[\|y^{k+1} - y^*_k(x^k)\|^2 \mid \mathcal{F}_k] + (1 + \frac{1}{\hat{\delta}})\mathbb{E}[\|y^*_k(x^k) - y^*_{k+1}(x^{k+1})\|^2 \mid \mathcal{F}_k].$$

$$(46)$$

Further, by taking $\hat{\delta} = \frac{1}{2}\beta_k\sigma_k$ and using Lemma D.2, we can bound the first term as follows:

$$(1 + \delta)\mathbb{E}[\|y^{k+1} - y^*_k(x^k)\|^2 \mid \mathcal{F}_k] \leq (1 + \frac{1}{2}\beta_k\sigma_k)(1 - \beta_k\sigma_k)\|y^k - y^*_k(x^k)\|^2 + (1 + \frac{1}{2}\beta_k\sigma_k)\beta_k^2\delta^2$$

$$\leq (1 - \frac{1}{2}\beta_k\sigma_k)\|y^k - y^*_k(x^k)\|^2 + (1 + \frac{1}{2}\beta_k\sigma_k)\beta_k^2\delta^2.$$

Next, combining Lemma D.4 and D.3, we have:

$$(1 + \frac{1}{\hat{\delta}})\mathbb{E}[\|y^*_k(x^k) - y^*_{k+1}(x^{k+1})\|^2 \mid \mathcal{F}_k]$$

$$\leq 2(1 + \frac{2}{\beta_k\sigma_k})\mathbb{E}[\|y^*_k(x^k) - y^*_k(x^{k+1})\|^2 \mid \mathcal{F}_k] + 2(1 + \frac{2}{\beta_k\sigma_k})\mathbb{E}[\|y^*_k(x^{k+1}) - y^*_{k+1}(x^{k+1})\|^2 \mid \mathcal{F}_k]$$

$$\leq 2(1 + \frac{2}{\beta_k\sigma_k})\left[\frac{\bar{L}_k^2}{\sigma_k^2}\mathbb{E}[\|x^{k+1} - x^k\|^2 \mid \mathcal{F}_k] + \frac{2(\rho_{k+1} - \rho_k)^2}{\sigma_k^2}M^4 + \frac{2(\sigma_k - \sigma_{k+1})^2}{\sigma_k^2}M^2\right].$$

Combining these inequalities with (46), we arrive at the desired inequality. □

**Lemma D.8.** *Let $\{\rho_k\}$ and $\{\sigma_k\}$ be sequences such that $\rho_{k+1} \geq \rho_k > 0$, $\sigma_k \geq \sigma_{k+1} > 0$. Suppose the step-size sequence $\{\beta_k\}$ satisfies $0 < \beta_k \leq \frac{1}{L_k}$ for each $k$. Then, the following inequality holds:*

$$\mathbb{E}[\varphi_{k+1}(x^{k+1}) \mid \mathcal{F}_k] - \varphi_k(x^k) + \left(\frac{1}{4\alpha_k} - \frac{L_{\varphi_k}}{2}\right)\mathbb{E}[\|x^{k+1} - x^k\|^2 \mid \mathcal{F}_k]$$

$$\leq \frac{\alpha_k}{2}\bar{L}_k^2\|y^k - y^*_k(x^k)\|^2 + \frac{\alpha_k}{2}\bar{L}_k^2\beta_k^2\delta^2 + \alpha_k\mathbb{E}[\|e_x^k\|^2 \mid \mathcal{F}_k] + (\sigma_k - \sigma_{k+1})M^2.$$

$$(47)$$

*Proof.* First, we can reformulate the total difference as follows:

$$\mathbb{E}[\varphi_{k+1}(x^{k+1}) \mid \mathcal{F}_{k+\frac{1}{2}}] - \varphi_k(x^k) = \mathbb{E}[\varphi_{k+1}(x^{k+1}) - \varphi_k(x^{k+1}) \mid \mathcal{F}_{k+\frac{1}{2}}] + \mathbb{E}[\varphi_k(x^{k+1}) - \varphi_k(x^k) \mid \mathcal{F}_{k+\frac{1}{2}}]. \quad (48)$$

By applying Lemma D.6 with $x = x^{k+1}$, the first term can be bounded as:

$$\mathbb{E}[\varphi_{k+1}(x^{k+1}) - \varphi_k(x^{k+1}) \mid \mathcal{F}_{k+\frac{1}{2}}] \leq (\sigma_k - \sigma_{k+1})M^2. \quad (49)$$

For the second term, $\varphi_k(x^{k+1}) - \varphi_k(x^k)$, we use the $L_{\varphi_k}$-Lipschitz continuity of $\nabla\varphi_k(x)$ established in Lemma D.5. The standard descent inequality (Beck, 2017, Lemma 5.7) states:

$$\mathbb{E}[\varphi_k(x^{k+1}) \mid \mathcal{F}_{k+\frac{1}{2}}] - \varphi_k(x^k) \leq \langle\mathbb{E}[\nabla\varphi_k(x^k), x^{k+1} - x^k\rangle \mid \mathcal{F}_{k+\frac{1}{2}}] + \frac{L_{\varphi_k}}{2}\mathbb{E}[\|x^{k+1} - x^k\|^2 \mid \mathcal{F}_{k+\frac{1}{2}}]. \quad (50)$$

Next, recall that the $x$-update is the projected gradient step

$$x^{k+1} = \mathcal{P}_X(x^k - \alpha_k d_x^k) = \arg\min_{x \in X}\left\{\langle d_x^k, x\rangle + \frac{1}{2\alpha_k}\|x - x^k\|^2\right\}.$$

By the first-order optimality condition of the above projection problem, we have:

$$\left\langle d_x^k + \frac{1}{\alpha_k}(x^{k+1} - x^k), x - x^{k+1}\right\rangle \geq 0, \qquad \forall x \in X.$$

Taking $x = x^k \in X$ yields:

$$\frac{1}{\alpha_k}\|x^{k+1} - x^k\|^2 \leq \langle -d_x^k, x^{k+1} - x^k\rangle.$$

Combining this inequality with the (50), we obtain:

$$\mathbb{E}[\varphi_k(x^{k+1}) \mid \mathcal{F}_{k+\frac{1}{2}}] - \varphi_k(x^k) + \left(\frac{1}{\alpha_k} - \frac{L_{\varphi_k}}{2}\right)\mathbb{E}[\|x^{k+1} - x^k\|^2 \mid \mathcal{F}_{k+\frac{1}{2}}]$$

$$\leq \mathbb{E}[\langle \nabla_x \psi_k(x^k, y_k^*(x^k)) - d_x^k, x^{k+1} - x^k \rangle \mid \mathcal{F}_{k+\frac{1}{2}}]$$

$$= \mathbb{E}[\langle \nabla_x \psi_k(x^k, y_k^*(x^k)) - \nabla_x \psi_k(x^k, y^{k+1}), x^{k+1} - x^k \rangle \mid \mathcal{F}_{k+\frac{1}{2}}] + \mathbb{E}[\langle \nabla_x \psi_k(x^k, y^{k+1}) - d_x^k, x^{k+1} - x^k \rangle \mid \mathcal{F}_{k+\frac{1}{2}}].$$

$$(51)$$

Further, by applying Lemma D.2 and Cauchy Schwarz inequality, the first term in the right side of this inequality can be bounded by:

$$\mathbb{E}[\langle \nabla_x \psi_k(x^k, y_k^*(x^k)) - \nabla_x \psi_k(x^k, y^{k+1}), x^{k+1} - x^k \rangle \mid \mathcal{F}_{k+\frac{1}{2}}]$$

$$\leq \bar{L}_k \|y^{k+1} - y_k^*(x^k)\| \mathbb{E}[\|x^{k+1} - x^k\| \mid \mathcal{F}_{k+\frac{1}{2}}]$$

$$\leq \frac{\alpha_k}{2}\bar{L}_k^2\|y^{k+1} - y_k^*(x^k)\|^2 + \frac{1}{2\alpha_k}\mathbb{E}[\|x^{k+1} - x^k\|^2 \mid \mathcal{F}_{k+\frac{1}{2}}]$$

$$\leq \frac{\alpha_k}{2}\bar{L}_k^2\|y^k - y_k^*(x^k)\|^2 + \frac{1}{2\alpha_k}\mathbb{E}[\|x^{k+1} - x^k\|^2 \mid \mathcal{F}_{k+\frac{1}{2}}] + \frac{\alpha_k}{2}\bar{L}_k^2\beta_k^2\delta^2,$$

where the last inequality follows from Lemma D.2. For the second term in the right side of (51),

$$\mathbb{E}[\langle \nabla_x \psi_k(x^k, y^{k+1}) - d_x^k, x^{k+1} - x^k \rangle \mid \mathcal{F}_{k+\frac{1}{2}}]$$

$$\leq \mathbb{E}[\alpha_k\|\nabla_x \psi_k(x^k, y^{k+1}) - d_x^k\|^2 + \frac{1}{4\alpha_k}\|x^{k+1} - x^k\|^2 \mid \mathcal{F}_{k+\frac{1}{2}}]$$

$$= \alpha_k\mathbb{E}[\|e_x^k\|^2 \mid \mathcal{F}_{k+\frac{1}{2}}] + \frac{1}{4\alpha_k}\mathbb{E}[\|x^{k+1} - x^k\|^2 \mid \mathcal{F}_{k+\frac{1}{2}}].$$

Combining the above inequality with (49) and (50):

$$\mathbb{E}[\varphi_{k+1}(x^{k+1}) \mid \mathcal{F}_{k+\frac{1}{2}}] - \varphi_k(x^k) + \left(\frac{1}{4\alpha_k} - \frac{L_{\varphi_k}}{2}\right)\mathbb{E}[\|x^{k+1} - x^k\|^2 \mid \mathcal{F}_{k+\frac{1}{2}}]$$

$$\leq \frac{\alpha_k}{2}\bar{L}_k^2\|y^k - y_k^*(x^k)\|^2 + \frac{\alpha_k}{2}\bar{L}_k^2\beta_k^2\delta^2 + \alpha_k\mathbb{E}[\|e_x^k\|^2 \mid \mathcal{F}_{k+\frac{1}{2}}] + (\sigma_k - \sigma_{k+1})M^2.$$

$$(52)$$

The conclusion follows by taking conditional expectation on $\mathcal{F}_k$. $\qquad\square$

**Lemma D.9.** *For the stochastic estimator $d_x^k$, the following statement holds at $k$-th iteration with $k > 0$:*

$$\mathbb{E}[\|e_x^k\|^2 \mid \mathcal{F}_k] \leq (1 - \eta_k)^2\|e_x^{k-1}\|^2 + 2\eta_k^2\delta^2 + 18\bar{L}_k^2 \left(\|x^k - x^{k-1}\|^2 + 3\beta_k^2\delta^2 + 3\beta_k^2\bar{L}_k^2\|y^k - y_k^*(x^k)\|^2 \right.$$
$$\left. + 3\beta_k^2(\rho_k M + 1 + \sigma_k M)^2 M^2\right),$$

$$(53)$$

*Proof.* Recall that $\mathcal{F}_k$ is the $\sigma$-algebra generated by the history up to iteration $k$. And we denote the expected direction by

$$D_x^k := \nabla_x \psi_k(x^k, y^{k+1}),$$

which implies that $e_x^k = d_x^k - D_x^k$.

Since the oracle is unbiased, note that $\mathcal{F}_{k+\frac{1}{2}} \in \mathcal{F}_k$, we have

$$\mathbb{E}\left[\nabla_x \Psi_k(x^k, y^{k+1}; \xi_k^x) \mid \mathcal{F}_k\right] = D_x^k, \qquad \mathbb{E}\left[\nabla_x \Psi_{k-1}(x^{k-1}, y^k; \xi_k^x) \mid \mathcal{F}_k\right] = D_x^k.$$

Consequently,

$$\mathbb{E}\left[\langle e_x^{k-1}, \nabla_x \Psi_k(x^k, y^{k+1}; \xi_k^x) - D_x^k \rangle \mid \mathcal{F}_k\right] = 0, \qquad \mathbb{E}\left[\langle e_x^{k-1}, D_x^{k-1} - \nabla_x \Psi_{k-1}(x^{k-1}, y^k; \xi_k^x) \rangle \mid \mathcal{F}_k\right] = 0.$$

Thus, the variance at $k$-th iteration can be composited as :

$$\mathbb{E}[\|e_x^k\|^2 \mid \mathcal{F}_k]$$

$$= \mathbb{E}[\|\nabla_x \Psi_k(x^k, y^{k+1}; \xi_k^x) - D_x^k + (1 - \eta_k)(d_x^{k-1} - \nabla_x \Psi_{k-1}(x^{k-1}, y^k; \xi_k^x))\|^2 \mid \mathcal{F}_k]$$

$$= \mathbb{E}[\|\nabla_x \Psi_k(x^k, y^{k+1}; \xi_k^x) - D_x^k + (1 - \eta_k)(d_x^{k-1} - D_x^{k-1} + D_x^{k-1} - \nabla_x \Psi_{k-1}(x^{k-1}, y^k; \xi_k^x))\|^2 \mid \mathcal{F}_k]$$

$$= \mathbb{E}[\|\nabla_x \Psi_k(x^k, y^{k+1}; \xi_k^x) - D_x^k + (1 - \eta_k)(D_x^{k-1} - \nabla_x \Psi_{k-1}(x^{k-1}, y^k; \xi_k^x))\|^2 \mid \mathcal{F}_k] + (1 - \eta_k)^2\|e_x^{k-1}\|^2.$$

$$(54)$$

Further, by the bounded variance assumption, smoothness assumption and $0 \le \eta_{k+1} \le 1$,

$$
\begin{aligned}
&\mathbb{E}[\|\nabla_x \Psi_k(x^k, y^{k+1}; \xi_k^x) - D_x^k + (1 - \eta_k)(D_x^{k-1} - \nabla_x \Psi_{k-1}(x^{k-1}, y^k; \xi_k^x))\|^2 \mid \mathcal{F}_k] \\
&\le 2(1 - \eta_k)^2 \mathbb{E}[\|\nabla_x \Psi_k(x^k, y^{k+1}; \xi_k^x) - \nabla_x \Psi_{k-1}(x^{k-1}, y^k; \xi_k^x) + D_x^{k-1} - D_x^k\|^2 \mid \mathcal{F}_k] \\
&\quad + 2\eta_k^2 \mathbb{E}[\|\nabla_x \Psi_k(x^k, y^{k+1}; \xi_k^x) - D_x^k\|^2 \mid \mathcal{F}_k] \\
&\le 2\mathbb{E}[\|\nabla_x \Psi_k(x^k, y^{k+1}; \xi_k^x) - \nabla_x \Psi_{k-1}(x^{k-1}, y^k; \xi_k^x) + D_x^{k-1} - D_x^k\|^2 \mid \mathcal{F}_k] + 2\eta_k^2 \delta^2.
\end{aligned}
\tag{55}
$$

Since the stochasticity only comes from the objective term $f(\cdot, \cdot; \xi)$, while the penalty/regularization terms (and the parameters $\rho_{k-1}, \rho_k$) are deterministic and do not depend on the sample $\xi$. Consequently,

$$
\nabla_x \Psi_k(x^{k-1}, y^k; \xi_k^x) - \nabla_x \Psi_{k-1}(x^{k-1}, y^k; \xi_k^x) - \left(\nabla_x \psi_k(x^{k-1}, y^k) - \nabla_x \psi_{k-1}(x^{k-1}, y^k)\right) = 0.
$$

Combining the Lipschitz of $\nabla_x \psi_k$ and Assumption 3.2, using similar trick as proof in Lemma 3.3, we can get that

$$
\begin{aligned}
&\mathbb{E}[\|\nabla_x \Psi_k(x^k, y^{k+1}; \xi_k^x) - \nabla_x \Psi_k(x^{k-1}, y^k; \xi_k^x)\|^2 \mid \mathcal{F}_k] \\
&\le \mathbb{E}[2\|\nabla_x f(x^k, y^{k+1}; \xi_k^x) - \nabla_x f(x^{k-1}, y^k; \xi_k^x)\|^2 \mid \mathcal{F}_k] \\
&\quad + 2\rho_k \mathbb{E}[\|\sum_{i=1}^p [c_i(x^k, y^{k+1})]_+ \nabla_x c_i(x^k, y^{k+1}) - \sum_{i=1}^p [c_i(x^{k-1}, y^k)]_+ \nabla_x c_i(x^{k-1}, y^k)\|^2 \mid \mathcal{F}_k] \\
&\le (2L_f + 2\rho_k p M L_c + 2\rho_k p M^2)^2 \mathbb{E}[\|x^k - x^{k-1}\|^2 + \|y^{k+1} - y^k\|^2 \mid \mathcal{F}_k] \\
&\le 2\bar{L}_k^2 \|x^k - x^{k-1}\|^2 + 2\bar{L}_k^2 \mathbb{E}[\|y^{k+1} - y^k\|^2 \mid \mathcal{F}_k].
\end{aligned}
$$

Thus, combining the two previous inequalities, we have

$$
\begin{aligned}
&\mathbb{E}[\|\nabla_x \Psi_k(x^k, y^{k+1}; \xi_k^x) - \nabla_x \Psi_{k-1}(x^{k-1}, y^k; \xi_k^x) + d_x^{k-1} - d_x^k\|^2 \mid \mathcal{F}_k] \\
&= \mathbb{E}[\|\nabla_x \Psi_k(x^k, y^{k+1}; \xi_k^x) - \nabla_x \Psi_k(x^{k-1}, y^k; \xi_k^x) + \nabla_x \Psi_k(x^{k-1}, y^k; \xi_k^x) - \nabla_x \Psi_{k-1}(x^{k-1}, y^k; \xi_k^x) \\
&\quad - (\nabla_x \psi_k(x^k, y^{k+1}) - \nabla_x \psi_k(x^{k-1}, y^k) + \nabla_x \psi_k(x^{k-1}, y^k) - \nabla_x \psi_{k-1}(x^{k-1}, y^k))\|^2 \mid \mathcal{F}_k] \\
&\le 3\mathbb{E}[\|\nabla_x \Psi_k(x^{k-1}, y^k; \xi_k^x) - \nabla_x \Psi_{k-1}(x^{k-1}, y^k; \xi_k^x) - (\nabla_x \psi_k(x^{k-1}, y^k) - \nabla_x \psi_{k-1}(x^{k-1}, y^k))\|^2 \mid \mathcal{F}_k] \\
&\quad + 3\mathbb{E}[\|\nabla_x \Psi_k(x^k, y^{k+1}; \xi_k^x) - \nabla_x \Psi_k(x^{k-1}, y^k; \xi_k^x)\|^2 \mid \mathcal{F}_k] \\
&\quad + 3\mathbb{E}[\|\nabla_x \psi_k(x^k, y^{k+1}) - \nabla_x \psi_k(x^{k-1}, y^k)\|^2 \mid \mathcal{F}_k] \\
&\le 9\bar{L}_k^2 \|x^k - x^{k-1}\|^2 + 9\bar{L}_k^2 \mathbb{E}[\|y^{k+1} - y^k\|^2 \mid \mathcal{F}_k],
\end{aligned}
\tag{56}
$$

where the last inequality follows from the $\bar{L}_k$ gradient Lipschitz of $\psi_k$. Combining the results in (55) and (56), we have:

$$
\begin{aligned}
&2\mathbb{E}[\|\nabla_x \Psi_k(x^k, y^{k+1}; \xi_k^x) - D_x^k + (1 - \eta_k)(D_x^{k-1} - \nabla_x \Psi_{k-1}(x^{k-1}, y^k; \xi_k^x))\|^2 \mid \mathcal{F}_k] \\
&\le 18\bar{L}_k^2 (\mathbb{E}[\|x^k - x^{k-1}\|^2 \mid \mathcal{F}_k] + \mathbb{E}[\|y^{k+1} - y^k\|^2 \mid \mathcal{F}_k]) + 2\eta_k^2 \delta^2.
\end{aligned}
$$

Furthermore, by the non-expansiveness of projection and boundedness of $y_k^*(x^k)$, we have

$$
\begin{aligned}
&\mathbb{E}[\|y^{k+1} - y^k\|^2 \mid \mathcal{F}_k] \\
&= \mathbb{E}[\|\mathcal{P}_Y(y^k + \beta_k d_k^y) - \mathcal{P}_Y(y^k)\|^2 \mid \mathcal{F}_k] \\
&\le \beta_k^2 \mathbb{E}[\|d_k^y\|^2 \mid \mathcal{F}_k] \\
&\le \beta_k^2 \mathbb{E}[\|d_k^y - \nabla_y \psi_k(x^k, y^k) + \nabla_y \psi_k(x^k, y^k) - \nabla_y \psi_k(x^k, y_k^*(x^k)) \\
&\quad + \nabla_y \psi_k(x^k, y_k^*(x^k))\|^2 \mid \mathcal{F}_{k+\frac{1}{2}}] \\
&\le 3\beta_k^2 \left(\mathbb{E}[\|d_k^y - \nabla_y \psi_k(x^k, y^k)\|^2 \mid \mathcal{F}_k] + \mathbb{E}[\|\nabla_y \psi_k(x^k, y_k^*(x^k))\|^2 \mid \mathcal{F}_k] \right. \\
&\quad \left. + \mathbb{E}[\|\nabla_y \psi_k(x^k, y^k) - \nabla_y \psi_k(x^k, y_k^*(x^k))\|^2 \mid \mathcal{F}_k]\right) \\
&\le 3\beta_k^2 \mathbb{E}[\|d_k^y - \nabla_y \psi_k(x^k, y^k)\|^2 \mid \mathcal{F}_k] \\
&\quad + 3\beta_k^2 \mathbb{E}[\|\nabla_y f(x^k, y_k^*(x^k)) + \rho_k [c(x^k, y_k^*(x^k))]_+ \nabla_y c(x^k, y_k^*(x^k))\|^2 \mid \mathcal{F}_k + \sigma_k y_k^*(x^k)] \\
&\quad + 3\beta_k^2 \bar{L}_k^2 \|y^k - y_k^*(x^k)\|^2 \\
&\le 3\beta_k^2 \delta^2 + 3\beta_k^2 (\rho_k M + 1 + \sigma_k M)^2 M^2 + 3\beta_k^2 \bar{L}_k^2 \|y^k - y_k^*(x^k)\|^2.
\end{aligned}
$$

Substituting this into the previous inequality, and using (54), we get:

$$\mathbb{E}[\|e_x^k\|^2 \mid \mathcal{F}_k] \le (1-\eta_k)^2 \|e_x^{k-1}\|^2 + 2\eta_k^2 \delta^2 + 12\bar{L}_k^2 (\|x^k - x^{k-1}\|^2 + 3\beta_k^2 \delta^2 + 3\beta_k^2 \bar{L}_k^2 \|y^k - y_k^*(x^k)\|^2$$
$$+ 3\beta_k^2 (\rho_k M + 1 + \sigma_k M)^2 M^2),$$

(57)

where the last inequality follows from Lemma D.2. This completes the proof. □

### D.3. Proof for Proposition 3.4

**Proposition D.10.** *(Proposition 3.4) Let $\{(x^k, y^k)\}$ be generated by SPACO (Algorithm 1) with penalty and regualrization parameters selected as:*

$$\sigma_k = \sigma_0 k^{-t}, \quad \rho_k = \rho_0 k^t,$$

*and the stepsizes and momentum parameter selected as:*

$$\alpha_k = \alpha_0 k^{-6t-s}, \quad \beta_k = \beta_0 k^{-t-s}, \quad \eta_k = \eta_0 k^{-s}$$

*where $\alpha_0, \beta_0, \sigma_0, \rho_0, t, s > 0$. Set the varying coefficients in $V_k$ as $a_k = k^{-2t}, b_k = k^{-3t}, c_k = k^{-7t}, d_k = k^{-4t}$. If $0 < t, s < 1$, $s > 3t$ and $8t + s < 1$, and if $\beta_0/\sigma_0$ is sufficiently small, then for all sufficiently large $k$, the following inequality holds:*

$$\mathbb{E}[V_{k+1} \mid \mathcal{F}_k] - V_k$$
$$\le -\frac{a_k \alpha_k}{24} \mathbb{E}[\|\mathcal{G}_k(x^k)\|^2 \mid \mathcal{F}_k] - \frac{b_k \beta_k \sigma_k}{4} \|y^k - y_k^*(x^k)\|^2$$
$$+ 2b_k \beta_k^2 \delta^2 + \zeta_k.$$

*where $\mathcal{G}_k(x) = \frac{1}{\alpha_k}(x - P_X(x - \alpha_k \nabla \varphi_k(x)))$ is the generalized gradient residual for $\min_{x \in X} \varphi_k(x)$ and $\{\zeta_k\}$ is a summable nonnegative sequence.*

*Proof for Proposition 3.4.* Recall the parameter schedules:

$$a_k = k^{-2t}, b_k = k^{-3t}, c_k = k^{-7t}, d_k = k^{-4t}, \alpha_k = \alpha_0 k^{-6t-s}, \beta_k = \beta_0 k^{-t-s}, \eta_k = \eta_0 k^{-s}, \sigma_k = \sigma_0 k^{-t}, \rho_k = \rho_0 k^t.$$

(58)

Recalling that $\bar{L}_k = L_f + \rho_k p M L_c + \rho_k p M^2 + \sigma_k$, the constant ratio $\beta_0/\sigma_0$ can be chosen sufficiently small to ensure that for all $k \ge 1$, the following inequality holds:

$$0 < \beta_k < \frac{1}{\bar{L}_k}.$$

Applying D.7, Lemma D.8 and Lemma D.9, and using the facts that $a_{k+1} < a_k$, $b_{k+1} < b_k$, $\eta_k < 1$, $\beta_{k+1} < \beta_k$ and

$\bar{L}_{k+1} > \bar{L}_k$, we obtain:

$$
\mathbb{E}[V_{k+1} \mid \mathcal{F}_k] - V_k
$$

$$
\leq a_k(\mathbb{E}[\varphi_{k+1}(x^{k+1}) \mid \mathcal{F}_k] - \varphi_k(x^k)) + b_k(\mathbb{E}[\|y^{k+1} - y_{k+1}^*(x^{k+1})\|^2 \mid \mathcal{F}_k] - \|y^k - y_k^*(x^k)\|^2)
$$

$$
+ c_k(\mathbb{E}[\|e_x^{k-1}\|^2 \mid \mathcal{F}_k] - \|e_x^{k-1}\|^2) + d_k(\|x^{k+1} - x^k\|^2 - \|x^k - x^{k-1}\|^2)
$$

$$
\leq - a_k(\frac{1}{4\alpha_k} - \frac{L_{\varphi_k}}{2})\mathbb{E}[\|x^{k+1} - x^k\|^2 \mid \mathcal{F}_k] + \frac{a_k\alpha_k}{2}\bar{L}_k^2\|y^k - y_k^*(x^k)\|^2 + \frac{a_k\alpha_k}{2}\bar{L}_k^2\beta_k^2\delta^2 + a_k\alpha_k\|e_x^k\|^2
$$

$$
+ a_k(\sigma_k - \sigma_{k+1})M^2 - \frac{1}{2}b_k\beta_k\sigma_k\|y^k - y_k^*(x^k)\|^2 + b_k(1 + \frac{1}{2}\beta_k\sigma_k)\beta_k^2\delta^2
$$

$$
+ 2b_k(1 + \frac{2}{\beta_k\sigma_k})\frac{\bar{L}_k^2}{\sigma_k^2}\mathbb{E}[\|x^k - x^{k+1}\|^2 \mid \mathcal{F}_k] + 2b_k(1 + \frac{2}{\beta_k\sigma_k})\left(\frac{2(\rho_{k+1} - \rho_k)^2}{\sigma_k^2}M^4 + \frac{2(\sigma_k - \sigma_{k+1})^2}{\sigma_k^2}M^2\right)
$$

$$
+ (-2c_k\eta_k + c_k\eta_k^2)\|e_x^{k-1}\|^2 + 2c_k\eta_k^2\delta^2 + 18c_k\bar{L}_k^2(\|x^k - x^{k-1}\|^2 + 3\beta_k^2\delta^2 + 3\beta_k^2\bar{L}_k^2\|y^k - y_k^*(x^k)\|^2
$$

$$
+ 3\beta_k^2(\rho_k M + 1 + \sigma_k M)^2 M^2) + d_k(\mathbb{E}[\|x^{k+1} - x^k\|^2 \mid \mathcal{F}_k] - \|x^k - x^{k-1}\|^2)
$$

$$
\leq -\underbrace{(\frac{a_k}{4\alpha_k} - \frac{a_k L_{\varphi_k}}{2} - 2b_k(1 + \frac{2}{\beta_k\sigma_k})\frac{\bar{L}_k^2}{\sigma_k^2} - d_k)\mathbb{E}[\|x^{k+1} - x^k\|^2 \mid \mathcal{F}_k]}_{(i)} \tag{59}
$$

$$
-\underbrace{(\frac{1}{2}b_k\beta_k\sigma_k - \frac{a_k\alpha_k}{2} - 54c_k\beta_k^2\bar{L}_k^4)\|y^k - y_k^*(x^k)\|^2}_{(ii)}
$$

$$
-\underbrace{(2c_k\eta_k - c_k\eta_k^2 - a_k\alpha_k)\|e_x^{k-1}\|^2}_{(iii)} -\underbrace{(d_k - 18c_k\bar{L}_k^2)\|x^k - x^{k-1}\|^2}_{(iv)}
$$

$$
+\underbrace{a_k(\sigma_k - \sigma_{k+1})M^2 + 2b_k(1 + \frac{2}{\beta_k\sigma_k})\left(\frac{2(\rho_{k+1} - \rho_k)^2}{\sigma_k^2}M^4 + \frac{2(\sigma_k - \sigma_{k+1})^2}{\sigma_k^2}M^2\right)}_{(v)}
$$

$$
+\underbrace{\frac{a_k\alpha_k}{2}\bar{L}_k^2\beta_k^2\delta^2 + b_k(1 + \frac{1}{2}\beta_k\sigma_k)\beta_k^2\delta^2 + 2c_k\eta_{k+1}^2\delta^2 + 54\bar{L}_{k+1}^2(c_k\beta_k^2\delta^2 + c_k\beta_k^2(\rho_k M + 1 + \sigma_k M)^2 M^2)}_{(vi)}
$$

*Part (i):* Recalling that $L_{\varphi_k} := \frac{(L_f + \rho_k M L_c + \rho_k M^2)(L_f + \rho_k M L_c + \rho_k M^2 + 2\sigma_k)}{\sigma_k}$, according to the criterion in (58), we have that:

$$
\frac{a_k}{4\alpha_k} = \mathcal{O}(k^{4t+s}), \ \frac{a_k L_{\varphi_k}}{2} = \mathcal{O}(k^{5t}), \ 4b_k(1 + \frac{2}{\beta_k\sigma_k})\frac{\bar{L}_k^2}{\sigma_k^2} = \mathcal{O}(k^{3t+s}), \ d_k = \mathcal{O}(k^{-4t}).
$$

Thus, for sufficiently large $k$, we have:

$$
\frac{a_k}{4\alpha_k} - \frac{a_k L_{\varphi_k}}{2} - 4b_k(1 + \frac{2}{\beta_k\sigma_k})\frac{\bar{L}_k^2}{\sigma_k^2} - d_k \geq \frac{a_k}{8\alpha_k},
$$

Equivalently,

$$
(i) \leq -\frac{a_k}{8\alpha_k}\mathbb{E}[\|x^{k+1} - x^k\|^2 \mid \mathcal{F}_k], \tag{60}
$$

Next, let $\mathcal{G}_k(x^k) = \frac{1}{\alpha_k}(x_k - P_X(x^k - \alpha_k \nabla \varphi_k(x^k)))$, it follows from the projection inequality that

$$\mathbb{E}[\|\mathcal{G}_k(x^k)\|^2 \mid \mathcal{F}_k]$$

$$= \frac{1}{\alpha_k^2} \mathbb{E}[\|x^k - P_X(x^k - \alpha_k \nabla \varphi_k(x^k)) + P_X(x^k - \alpha_k \nabla_x \psi_k(x^k, y^{k+1})) - P_X(x^k - \alpha_k \nabla_x \psi_k(x^k, y^{k+1}))$$

$$+ x^{k+1} - x^{k+1}\|^2 \mid \mathcal{F}_k]$$

$$\leq \frac{3}{\alpha_k^2} \left( \mathbb{E}[\|x^{k+1} - x^k\|^2 \mid \mathcal{F}_k] + \mathbb{E}[\|P_X(x^k - \alpha_k \nabla \varphi_k(x^k)) - P_X(x^k - \alpha_k \nabla_x \psi_k(x^k, y^{k+1}))\|^2 \mid \mathcal{F}_k] \right.$$

$$\left. + \mathbb{E}[\|P_X(x^k - \alpha_k \nabla_x \psi_k(x^k, y^{k+1})) - x^{k+1}\|^2 \mid \mathcal{F}_k] \right)$$

$$= \frac{3}{\alpha_k^2} \left( \mathbb{E}[\|x^{k+1} - x^k\|^2 \mid \mathcal{F}_k] + \mathbb{E}[\|P_X(x^k - \alpha_k \nabla \varphi_k(x^k)) - P_X(x^k - \alpha_k \nabla_x \psi_k(x^k, y^{k+1}))\|^2 \mid \mathcal{F}_k] \right.$$

$$\left. + \mathbb{E}[\|P_X(x^k - \alpha_k \nabla_x \psi_k(x^k, y^{k+1})) - P_X(x^k - \alpha_k d_x^k)\|^2) \mid \mathcal{F}_k] \right)$$

$$\leq \frac{3}{\alpha_k^2} \left( \mathbb{E}[\|x^{k+1} - x^k\|^2 \mid \mathcal{F}_k] + \alpha_k^2 \bar{L}_k^2 \|y^k - y_k^*(x^k)\|^2 + \alpha_k^2 \mathbb{E}[\|e^k\|^2 \mid \mathcal{F}_k] + \alpha_k^2 \bar{L}_k^2 \beta_k^2 \delta^2 \right),$$

where the last inequality follows from Lemma D.2. Thus, by applying Lemma D.9, we have that

$$-2\mathbb{E}[\|x^{k+1} - x^k\|^2 \mid \mathcal{F}_k] \leq -\frac{\alpha_k^2}{3} \mathbb{E}[\|\mathcal{G}_k(x^k)\|^2 \mid \mathcal{F}_k] + 2\alpha_k^2 \bar{L}_k^2 \|y^k - y_k^*(x^k)\|^2 + \alpha_k^2 \|e^{k-1}\|^2 + 2\alpha_k^2 \bar{L}_k^2 \beta_k^2 \delta^2.$$

Combining this with (60), (59) can be reformulated as

$$\mathbb{E}[V_{k+1} \mid \mathcal{F}_k] - V_k$$

$$\leq \underbrace{-\frac{a_k \alpha_k}{24} \mathbb{E}[\|\mathcal{G}_k(x^k)\|^2 \mid \mathcal{F}_k]}_{(i)} \underbrace{-(\frac{1}{2} b_k \beta_k \sigma_k - \frac{a_k \alpha_k}{2} - 54 c_k \beta_k^2 \bar{L}_k^4 - \frac{a_k \alpha_k \bar{L}_k^2}{8})\|y^k - y_k^*(x^k)\|^2}_{(ii)}$$

$$\underbrace{-(2c_k \eta_k - c_k \eta_k^2 - \frac{9}{8} a_k \alpha_k)\|e_x^{k-1}\|^2}_{(iii)} \underbrace{-(d_k - 18 c_k \bar{L}_k^2)\|x^k - x^{k-1}\|^2}_{(iv)} \tag{61}$$

$$\underbrace{+ a_k(\sigma_k - \sigma_{k+1})M^2 + 2b_k(1 + \frac{2}{\beta_k \sigma_k}) \left( \frac{2(\rho_{k+1} - \rho_k)^2}{\sigma_k^2} M^4 + \frac{2(\sigma_k - \sigma_{k+1})^2}{\sigma_k^2} M^2 \right)}_{(v)}$$

$$\underbrace{+ \frac{a_k \alpha_k}{2} \bar{L}_k^2 \beta_k^2 \delta^2 + b_k(1 + \frac{1}{2}\beta_k \sigma_k)\beta_k^2 \delta^2 + 2c_k \eta_{k+1}^2 \delta^2 + 54 \bar{L}_{k+1}^2(c_k \beta_k^2 \delta^2 + c_k \beta_k^2(\rho_k M + 1 + \sigma_k M)^2 M^2)}_{(vi)}$$

*Part (ii), (iii) and (iv):* According to the criterion in (58) again,

$$\frac{1}{2} b_k \beta_k \sigma_k = \mathcal{O}(k^{-5t-s}), \ \frac{a_k \alpha_k}{2} = \mathcal{O}(k^{-8t-s}), \ 54 c_k \beta_k^2 \bar{L}_k^4 = \mathcal{O}(k^{-5t-2s}), \ \frac{a_k \alpha_k \bar{L}_k^2}{8} = \mathcal{O}(k^{-6t-s}),$$

$$2c_k \eta_k = \mathcal{O}(k^{-7t-s}), \ c_k \eta_k^2 = \mathcal{O}(k^{-7t-2s}), \ \frac{9a_k \alpha_k}{8} = \mathcal{O}(k^{-8t-s}).$$

$$18 c_k \bar{L}_k^2 = \mathcal{O}(k^{-5t}), \ d_k = \mathcal{O}(k^{-4t}).$$

Therefore, for sufficiently large $k$, we have:

$$\frac{1}{2} b_k \beta_k \sigma_k - \frac{a_k \alpha_k}{2} - 54 c_k \beta_{k+1}^2 \bar{L}_{k+1}^4 - \frac{a_k \alpha_k \bar{L}_{k+1}^2}{8} \geq \frac{1}{4} b_k \beta_k \sigma_k,$$

$$2c_k \eta_{k+1} - c_k \eta_{k+1}^2 - \frac{9a_k \alpha_k}{8} \geq c_k \eta_{k+1},$$

$$d_k - 18 c_k \bar{L}_k^2 \geq 0.$$

This implies that:

$$(ii) \leq -\frac{1}{4}b_k\beta_k\sigma_k\|y^k - y_k^*(x^k)\|^2,$$
$$(iii) \leq 0, \tag{62}$$
$$(iv) \leq 0.$$

*Part (v):* According to the criterion in (58), there exist a constant $C$ such that :

$$2b_k(1 + \frac{2}{\beta_k\sigma_k})\left(\frac{2(\rho_{k+1} - \rho_k)^2}{\sigma_k^2}M^4 + \frac{2(\sigma_k - \sigma_{k+1})^2}{\sigma_k^2}M^2\right) \leq Ck^{3t+s-2}.$$

Noting that $3t + s < 8t + s < 1$, we have that

$$\sum_{k=1}^{\infty} a_k(\sigma_k - \sigma_{k+1})M^2 \leq \sum_{k=1}^{\infty}(\sigma_k - \sigma_{k+1})M^2 \leq \sigma_0 M^2 < \infty,$$

$$\sum_{k=1}^{\infty} 2b_k(1 + \frac{2}{\beta_k\sigma_k})\left(\frac{2(\rho_{k+1} - \rho_k)^2}{\sigma_k^2}M^4 + \frac{2(\sigma_k - \sigma_{k+1})^2}{\sigma_k^2}M^2\right) < \infty.$$

By denoting $\zeta_k = a_k(\sigma_k - \sigma_{k+1})M^2 + 2b_k(1 + \frac{2}{\beta_k\sigma_k})\left(\frac{2(\rho_{k+1}-\rho_k)^2}{\sigma_k^2}M^4 + \frac{2(\sigma_k-\sigma_{k+1})^2}{\sigma_k^2}M^2\right)$, we have that $\zeta_k = (iv) > 0$ and

$$\sum_{k=1}^{\infty} \zeta_k = \sum_{k=1}^{\infty}(iv) < \infty. \tag{63}$$

*Part (vi):* According to the criterion in (58) again:

$$\frac{3a_k\alpha_k}{2}\bar{L}_k^2\beta_k^2\delta^2 = \mathcal{O}(k^{-8t-4s}), \ b_k(1 + \frac{1}{2}\beta_k\sigma_k)\beta_k^2\delta^2 = \mathcal{O}(k^{-5t-2s}), \ 2c_k\eta_k^2\delta^2 = \mathcal{O}(k^{-7t-2s}),$$

$$54\bar{L}_k^2 c_k\beta_k^2\delta^2 = \mathcal{O}(k^{-7t-3s}), \ 54\bar{L}_k^2 c_k\beta_k^2(\rho_k M + 1 + \sigma_k M)^2 M^2 = \mathcal{O}(k^{-5t-3s}).$$

Since $1 + \frac{1}{2}\beta_k\sigma_k < \frac{3}{2}$ for sufficiently large $k$ and $s > 3t$, we can summarize:

$$(vi) \leq 2b_k\beta_k^2\delta^2. \tag{64}$$

Combining (62),(63), (64) and (61), the proof is completed. $\square$

## D.4. Proof for Theorem 3.5

**Theorem D.11.** *(Theorem 3.5) Let $\{(x^k, y^k)\}$ be the sequence generated by SPACO (Algorithm 1) with parameters selected as in Proposition 3.4. Then,*

$$\min_{0 \leq k \leq K}\{\mathbb{E}[\|\mathcal{G}_k(x^k)\|^2]\} = \mathcal{O}(\frac{1}{K^{1-8t-s}}) + \mathcal{O}(\frac{1}{K^{s-3t}}),$$

$$\min_{0 \leq k \leq K}\{\mathbb{E}[\|y^k - y_k^*(x^k)\|^2]\} = \mathcal{O}(\frac{1}{K^{1-5t-s}}) + \mathcal{O}(\frac{1}{K^s}).$$

*Proof for Theorem 3.5.* Taking expectation on the whole probability space on both side of inequality in Proposition 3.4, there exists an integer $k_0 \geq 1$ such that for all $k \geq k_0$,

$$\frac{a_k\alpha_k}{24}\mathbb{E}[\|\mathcal{G}_k(x^k)\|^2] + \frac{1}{4}b_k\beta_k\sigma_k\mathbb{E}[\|y^k - y_k^*(x^k)\|^2] \leq \ \mathbb{E}[V_k] - \mathbb{E}[V_{k+1}] + b_k\beta_k^2\delta^2 + \zeta_k. \tag{65}$$

Summing (65) from $k = k_0$ to $K$ yields

$$\sum_{k=k_0}^{K}\frac{a_k\alpha_k}{24}\mathbb{E}[\|\mathcal{G}_k(x^k)\|^2] + \sum_{k=k_0}^{K}\frac{1}{4}b_k\beta_k\sigma_k\mathbb{E}[\|y^k - y_k^*(x^k)\|^2] \leq \ \mathbb{E}[V_{k_0}] - \mathbb{E}[V_{K+1}] + \sum_{k=k_0}^{K}b_k\beta_k^2\delta^2 + \sum_{k=k_0}^{K}\zeta_k.$$

Since $V_k \geq 0$ for all $k$, it follows that

$$\sum_{k=k_0}^{K} \frac{a_k \alpha_k}{24} \mathbb{E}[\|\mathcal{G}_k(x^k)\|^2] + \sum_{k=k_0}^{K} \frac{1}{4} b_k \beta_k \sigma_k \mathbb{E}[\|y^k - y_k^*(x^k)\|^2] \leq \mathbb{E}[V_{k_0}] + \sum_{k=k_0}^{\infty} b_k \beta_k^2 \delta^2 + \sum_{k=k_0}^{\infty} \zeta_k.$$

Applying (58), we have $a_k \alpha_k = k^{-(8t+s)}$ and $b_k \beta_k \sigma_k = k^{-(5t+s)}$. Since both sequences are non-increasing, for any $k \leq K$ we have $a_k \alpha_k \geq a_K \alpha_K = K^{-(8t+s)}$ and $b_k \beta_k \sigma_k \geq b_K \beta_K \sigma_K = K^{-(5t+s)}$. Thus, combining with (65) yields

$$\frac{1}{K} \sum_{k=k_0}^{K} \mathbb{E}[\|\mathcal{G}_k(x^k)\|^2] \leq \mathcal{O}\left(\frac{1}{K^{s-3t}}\right) + \mathcal{O}\left(\frac{1}{K^{1-8t-s}}\right),$$

$$\frac{1}{K} \sum_{k=k_0}^{K} \mathbb{E}[\|y^k - y_k^*(x^k)\|^2] \leq \mathcal{O}\left(\frac{1}{K^s}\right) + \mathcal{O}\left(\frac{1}{K^{1-5t-s}}\right),$$

which completes the proof. $\square$

### D.5. Proof for Theorem 3.6

**Lemma D.12.** *(Robbins & Siegmund, 1971, Theorem 1) Let $(\Omega, \mathcal{F}, P)$ be a probability space and $\mathcal{F}_1 \subset \mathcal{F}_2 \subset \cdots$, suppose that $X_n$, $Y_n$ and $Z_n$ are non-negative $\mathcal{F}_n$ measurable random variables such that*

$$\mathbb{E}[X_{n+1}|\mathcal{F}_n] \leq X_n + Y_n - Z_n,$$

*if further $\sum_{n=1}^{\infty} Y_n < \infty$, then $\lim_{n \to \infty} X_n$ exists and $\sum_{n=1}^{\infty} Z_n < \infty$ almost surely.*

**Lemma D.13.** *Let $\{(x^k, y^k)\}$ be the sequence generated by Algorithm 1 with parameters selected as in (58). Assume further that $\alpha_0, \beta_0, \sigma_0, \rho_0, t, s > 0$. Let $0 < t, s < 1$, $8t + s < 1$ and $2s + 5t > 1$. Then, we have, almost surely, that*

$$\liminf_{k \to \infty} \|\mathcal{G}_k(x^k)\|^2 = \liminf_{k \to \infty} \|y^k - y_k^*(x^k)\|^2 = 0.$$

*Proof.* We provide the proof for $\liminf_{k \to \infty} \|\mathcal{G}_k(x^k)\|^2 = 0$, the proof for $\|y^k - y_k^*(x^k)\|^2$ can be derived similarly. By applying Proposition 3.4, by rearranging the inequality in Proposition 3.4, for sufficiently large $k$ we have

$$\mathbb{E}[V_{k+1}|\mathcal{F}_k] \leq V_k - \frac{a_k \alpha_k}{24} \|\mathcal{G}_k(x^k)\|^2 + b_k \beta_k^2 \delta^2 + \zeta_k.$$

As $b_k \beta_k^2 \delta^2 = O(\frac{1}{k^{2s+5t}})$, we have that both $b_k \beta_k^2 \delta^2$ and $\zeta_k$ are summable, it follows from Lemma D.12 with $X_k = V_k$, $Y_k = b_k \beta_k^2 \delta^2 + \zeta_k$ and $Z_k = \frac{a_k \alpha_k}{24} \|\mathcal{G}_k(x^k)\|^2$ that

$$\sum_{k=1}^{\infty} \frac{a_k \alpha_k}{24} \|\mathcal{G}_k(x^k)\|^2 < \infty \ a.s..$$

Since $\sum_{k=1}^{\infty} \frac{a_k \alpha_k}{24} = \infty$, we have that

$$\liminf_{k \to \infty} \|\mathcal{G}_k(x^k)\|^2 = 0 \text{ a.s.},$$

which completes the proof. $\square$

**Theorem D.14.** *(Theorem 3.6) Let $\{(x^k, y^k)\}$ be the sequence generated by SPACO (Algorithm 1) with parameters selected as in Proposition 3.4, and further assume that $2s + 5t > 1$. Then, almost surely, there exists a convergent subsequence $\{(x^{k_j}, y^{k_j})\}$ such that $\|\mathcal{G}_{k_j}(x^{k_j})\| \to 0$ and its limit point $(\bar{x}, \bar{y})$ is a stationary (KKT) point of the MCC (1) defined in (7), provided GPŁCQ holds at $(\bar{x}, \bar{y})$.*

*Proof for Theorem 3.6.* By applying Lemma D.13, we have almost surely that, we can find a subsequence $k_i$ such that

$$\lim_{i \to \infty} \|\mathcal{G}_{k_i}(x^{k_i})\| = \lim_{i \to \infty} \|y^{k_i} - y_{k_i}^*(x^{k_i})\| = 0.$$

As $(x_{k_i}, y_{k_i})$ are bounded, suppose that $(\bar{x}, \bar{y})$ is an accumulation point of a subsequence $(x^{k_j}, y^{k_j})$ and satisfies GPŁCQ. Then, without loss of generality, by re-index the sequence, we assume that $\lim_{k \to \infty}(x^k, y^k) = (\bar{x}, \bar{y})$.

We first establish boundedness of the multiplier $\lambda_k = \rho_k[c(x_k, y_k^*(x_k))]_+$. Note that the optimality condition of $y_k^*(x^k)$ implies

$$0 = y_k^*(x^k) - \mathcal{P}_Y(y_k^*(x^k) + \alpha_k \nabla_y \psi_k(x^k, y_k^*(x^k))).$$

Combining Cauchy-Schwarz inequality and the non-expansiveness of the projection operator, we have that for sufficiently large $k$:

$$\frac{1}{\alpha_k \rho_k} \left\| \begin{bmatrix} x^k \\ y_k^*(x^k) \end{bmatrix} - \mathcal{P}_{X \times Y}\left( \begin{bmatrix} x^k \\ y_k^*(x^k) \end{bmatrix} - \alpha_k \rho_k \begin{bmatrix} \nabla_x p(x^k, y_k^*(x^k)) \\ -\nabla_y p(x^k, y_k^*(x^k)) \end{bmatrix} \right) \right\|$$

$$\leq \frac{1}{\alpha_k \rho_k} \left\| \begin{bmatrix} x^k \\ y_k^*(x^k) \end{bmatrix} - \mathcal{P}_{X \times Y}\left( \begin{bmatrix} x^k \\ y_k^*(x^k) \end{bmatrix} - \alpha_k \rho_k \begin{bmatrix} \nabla_x p(x^k, y_k^*(x^k)) \\ -\nabla_y p(x^k, y_k^*(x^k)) \end{bmatrix} \right) \right.$$

$$\left. - \left( \begin{bmatrix} x^k \\ y_k^*(x^k) \end{bmatrix} - \mathcal{P}_{X \times Y}\left( \begin{bmatrix} x^k \\ y_k^*(x^k) \end{bmatrix} - \alpha_k \begin{bmatrix} \nabla_x \psi_k(x^k, y_k^*(x^k)) \\ -\nabla_y \psi_k(x^k, y_k^*(x^k)) \end{bmatrix} \right) \right) \right\|$$

$$+ \frac{1}{\alpha_k \rho_k} \left\| \begin{bmatrix} x^k \\ y_k^*(x^k) \end{bmatrix} - \mathcal{P}_{X \times Y}\left( \begin{bmatrix} x^k \\ y_k^*(x^k) \end{bmatrix} - \alpha_k \begin{bmatrix} \nabla_x \psi_k(x^k, y_k^*(x^k)) \\ -\nabla_y \psi_k(x^k, y_k^*(x^k)) \end{bmatrix} \right) \right\|$$

$$\leq \frac{1}{\rho_k} \left\| \begin{bmatrix} -\rho_k \nabla_x p(x^k, y_k^*(x^k)) - \nabla_x \psi_k(x^k, y_k^*(x^k)) \\ \rho_k \nabla_y p(x^k, y_k^*(x^k)) + \nabla_y \psi_k(x^k, y_k^*(x^k)) \end{bmatrix} \right\| + \frac{1}{\alpha_k \rho_k} \|x^k - \mathcal{P}_X(x^k - \alpha_k \nabla_x \psi_k(x^k, y_k^*(x^k)))\|$$

$$\leq \frac{1}{\rho_k} \left\| \begin{bmatrix} \nabla_x f(x^k, y_k^*(x^k)) \\ -\nabla_y f(x^k, y_k^*(x^k)) + \sigma_k y_k^*(x^k) \end{bmatrix} \right\| + \frac{1}{\rho_k} \|\mathcal{G}_k(x^k)\|$$

$$\leq \frac{3}{\rho_k} M.$$

Consequently, since $(x_k, y_k^*(x_k)) \to (\bar{x}, \bar{y})$, $\alpha_k \rho_k \to 0$ by the stepsizes criterion in (58) and GPŁCQ holds at $(\bar{x}, \bar{y})$, for sufficiently large $k$, we have $x^k \in B_\delta(\bar{x}), y_k^*(x_k) \in B_\delta(\bar{y}), \alpha_k \rho_k \leq \beta$. As a result, the following inequality holds:

$$\|\lambda_k\| = \|[\rho_k c(x_k, y_k^*(x_k))]_+\| = \sqrt{2}\rho_k \|p(x_k, y_k)\|^{\frac{1}{2}}$$

$$\leq \sqrt{2}v\rho_k\beta \left\| \begin{bmatrix} x_k \\ y_k^*(x_k) \end{bmatrix} - \mathcal{P}_{X \times Y}\left( \begin{bmatrix} x_k \\ y_k^*(x_k) \end{bmatrix} - \beta \begin{bmatrix} \nabla_x p(x_k, y_k^*(x_k)) \\ -\nabla_y p(x_k, y_k^*(x_k)) \end{bmatrix} \right) \right\|$$

$$\leq \sqrt{2}v\rho_k \frac{1}{\alpha_k \rho_k} \left\| \begin{bmatrix} x_k \\ y_k^*(x_k) \end{bmatrix} - \mathcal{P}_{X \times Y}\left( \begin{bmatrix} x_k \\ y_k^*(x_k) \end{bmatrix} - \alpha_k \rho_k \begin{bmatrix} \nabla_x p(x_k, y_k^*(x_k)) \\ -\nabla_y p(x_k, y_k^*(x_k)) \end{bmatrix} \right) \right\|$$

$$\leq 3\sqrt{2}vM.$$

Hence, the sequence $\{\lambda_k\}$ is bounded. By using the same trick as in Theorem 2.9, we show that any accumulation point of the multipliers $\lambda_k$ corresponds to a multiplier $\bar{\lambda}$ for $(\bar{x}, \bar{y})$. Without loss of generality, we assume that $\lambda_k \to \bar{\lambda}$. Further, by applying the projection inequality, we that

$$\langle x^k - \alpha_k \nabla \varphi_k(x^k) - \mathcal{P}_X(x^k - \alpha_k \nabla \varphi_k(x^k)), x - \mathcal{P}_X(x^k - \alpha_k \nabla \varphi_k(x^k)) \rangle \leq 0, \ \forall x \in X.$$

Recalling the definition of $\mathcal{G}_k(x^k) = \frac{1}{\alpha_k}(x_k - P_X(x_k - \alpha_k \nabla \varphi_k(x_k)))$, we have that

$$\langle \alpha_k \mathcal{G}_k(x^k) - \alpha_k \nabla \varphi_k(x^k), x - \mathcal{P}_X(x^k - \alpha_k \nabla \varphi_k(x^k)) \rangle$$

$$= \langle \alpha_k \mathcal{G}_k(x^k) - \alpha_k \nabla \varphi_k(x^k), x - x^k + x^k - \mathcal{P}_X(x^k - \alpha_k \nabla \varphi_k(x^k)) \rangle$$

$$= \langle \alpha_k \mathcal{G}_k(x^k) - \alpha_k \nabla \varphi_k(x^k), x - x^k + \alpha_k \mathcal{G}_k(x^k) \rangle$$

$$= \langle -\alpha_k \nabla \varphi_k(x^k), x - x^k \rangle + \langle \alpha_k \mathcal{G}_k(x^k), x - x^k + \alpha_k \mathcal{G}_k(x^k) \rangle - \langle -\alpha_k \nabla \varphi_k(x^k), \alpha_k \mathcal{G}_k(x^k) \rangle$$

$$\leq 0, \ \forall x \in X.$$

which implies that $\langle -\nabla \varphi_k(x^k), x - x^k \rangle + \langle \mathcal{G}_k(x^k), x - x^k + \alpha_k \mathcal{G}_k(x^k) \rangle - \langle -\nabla \varphi_k(x^k), \alpha_k \mathcal{G}_k(x^k) \rangle \leq 0$. Note that $\nabla \varphi_k(x^k) = \nabla_x f(x^k, y_k^*(x^k)) - \lambda_k^\top \nabla_x c(x^k, y_k^*(x^k))$ is bounded since $\lambda_k$, $X$ and $Y$ are bounded. Taking limitation to inf

and applying the fact that $\|\mathcal{G}_k(x^k)\| \to 0$, we have:

$$\lim_{k\to\infty} \langle -\nabla\varphi_k(x^k), x - x^k \rangle + \langle \mathcal{G}_k(x^k), x - x^k + \alpha_k\mathcal{G}_k(x^k) \rangle - \langle -\nabla\varphi_k(x^k), \alpha_k\mathcal{G}_k(x^k) \rangle$$

$$= \lim_{k\to\infty} \langle -\nabla_x f(x^k, y_k^*(x^k)) + \lambda_k^\top \nabla_x c(x^k, y_k^*(x^k)), x - x^k \rangle$$

$$= \langle -\nabla_x f(\bar{x}, \bar{y}) + \bar{\lambda}^\top \nabla_x c(\bar{x}, \bar{y}), x - \bar{x} \rangle$$

$$\leq 0, \ \forall x \in X,$$

which is equally to $-\nabla_x f(\bar{x}, \bar{y}) + \bar{\lambda}^\top \nabla_x c(\bar{x}, \bar{y}) \in N_X(\bar{x})$. On the other hand, recalling the optimality condition of $y_k^*(x_k)$, we have that

$$\nabla_y f(x_k, y_k^*(x_k)) - \lambda_k \nabla_y c(x_k, y_k^*(x_k)) - \sigma_k y_k^*(x_k) \in N_Y(y_k^*(x_k)).$$

As $Y$ is a convex compact set, $N_Y(\cdot)$ is outer semi-continuous (see, e.g., (Rockafellar & Wets, 2009, Proposition 6.6)), which implies that

$$\nabla_y f(\bar{x}, \bar{y}) - \bar{\lambda}^\top \nabla_y c(\bar{x}, \bar{y}) = \lim_{k\to\infty} \nabla_y f(x_k, y_k^*(x_k)) - \lambda_k^\top \nabla_y c(x_k, y_k^*(x_k)) - \sigma_k y_k^*(x_k) \in N_Y(\bar{y}).$$

Finally, the feasibility and complementarity follows exact the same as Theorem 2.9. Thus, the proof is completed. □

## E. Proof for Section 4

Here we present several examples beyond (11) to illustrate that penalty-based algorithms can help avoid spurious stationary points in many cases. These examples are summarized in Table 5. Specifically, the first example considers an objective function different from that in (11), the second example modifies the constraint structure in (11), and the third example extends (11) to a higher-dimensional setting.

| Problem | | Min–Min–Max reformulation (2) | Penalty-based approximation(3) |
|---|---|---|---|
| $\min\limits_{x \in X} \max\limits_{y \in Y, y \leq x^2} (x^2 - 1)^2 - (y - 1)^2 + y\ln(1+x)$ | | $[0, 0, 2]$ | |
| where $X = [-\frac{3}{4}, \frac{3}{4}]$, $Y = [-1, 1]$ | | $[-\frac{3}{4}, 1 - \ln 2, 0]$ | $[-\frac{3}{4}, 1 - \ln 2]\ (\rho > 1, \sigma = 0)$ |
| $\min\limits_{x \in X} \max\limits_{y \in Y, y \leq x^4} (x^4 - 1)^2 - (y - 1)^2 + y\ln(1+x)$ | | $[0, 0, 2]$ | |
| where $X = [-\frac{3}{4}, \frac{3}{4}]$, $Y = [-1, 1]$ | | $[-\frac{3}{4}, 1 - \ln 2, 0]$ | $[-\frac{3}{4}, 1 - \ln 2]\ (\rho > 1, \sigma = 0)$ |
| $\min\limits_{x \in X} \max\limits_{y \in Y, e^\top y - \|x\|^2 \leq 0} \frac{n}{2}(\frac{\|x\|^2}{n} - 1)^2 - \frac{1}{2}\|y - e\|^2 + \frac{1}{2}x^\top y$ | | $[\mathbf{0}, \mathbf{0}, 2]$ | |
| where $X = [-\frac{3}{4}, \frac{5}{4}]^n$, $Y = [-10, 10]^n$ | | $[-\frac{3}{4}e, \frac{9}{16}e, 0]$ | $[-\frac{3}{4}e, \frac{10+9\rho n}{16(1+\rho n)}e]\ (\rho > n, \sigma = 0)$ |

*Table 5.* Illustrative examples where the min–min–max reformulation admits non-optimal stationary minimax points, while the penalty-based reformulation does not exhibit such points for penalty-based smooth approximation.

### E.1. Experiment Setup for Figure 1

We solve the min–min–max reformulation using a nested optimization scheme implemented by the solver *Scipy* (Virtanen et al., 2020). The outer minimization over the joint variable $(x, \lambda)$ and the inner maximization over $y$ are solved using *scipy.optimize.minimize*. The solver terminates when its standard stopping criteria are met. For SPACO, we adopt the same parameter settings as the nonlinear constrained problem (12): initial penalty $\rho_0 = 10$, primal stepsizes $\alpha_0 = \beta_0 = 0.1$, proximal parameter $\sigma_0 = 10^{-4}$, and update factors $t = 0.05, s = 0.2$. It terminates when the KKT residual falls below $10^{-6}$ or the iteration count reaches $10^4$.

To analyze the convergence properties, we initialize $x^0$ on a uniform grid over $[-\frac{3}{4}, \frac{3}{4}]^2$ with fixed $y^0 = \mathbf{0}$. We categorize the convergence of each initialization based on the proximity of the final iterate $(x, y)$ to the known stationary points. $(x^*, y^*) = (-\frac{3}{4}e, \frac{9}{16}e)$ denotes the true solution and $(x', y') = (\mathbf{0}, \mathbf{0})$ denotes the spurious stationary point. We define the convergence errors as:

$$E_{\text{opt}} := \max \|x - x^*\|, \|y - y^*\|, \ E_{\text{spur}} := \max \|x - \mathbf{0}\|, \|y - \mathbf{0}\|. \tag{66}$$

In Figure 1, green regions ■ represent initializations converging to the true solution ($E_{\text{opt}} \leq 0.1$), while orange regions ■ indicate convergence to the spurious stationary point ($E_{\text{spur}} \leq 0.1$).

### E.2. Proof for Lemma 4.1

**Lemma E.1.** *() Consider the penalty-based smooth approximation of* (11)*:*

$$\min_{x \in X} \max_{y \in Y} \psi_{\rho,\sigma}(x,y).$$

*Since the objective is strongly concave in $y$, we take $\sigma = 0$. Then, for any $\rho_k > 1$, this approximation admits a unique stationary point $(x_k^*, y_k^*) = \left(-\frac{3}{4}\mathbf{e}, \frac{10+18\rho_k}{16+32\rho_k}\mathbf{e}\right)$, which converges to $(-\frac{3}{4}\mathbf{e}, \frac{9}{16}\mathbf{e})$ as $\rho_k \to 0$.*

*proof for Lemma 4.1.* Note that

$$\psi_{\rho,0}(x,y) := (\frac{\|x\|^2}{2} - 1)^2 - \frac{1}{2}(y-e)^2 + \frac{1}{2}x^\top y - \frac{\rho_k}{2}[e^\top y - \|x\|^2]_+^2, \qquad (x,y) \in X \times Y,$$

where $X = [-\frac{3}{4}, \frac{5}{4}]^2$ and $Y = [-10, 10]^2$. For each fixed $x \in X$, the mapping $y \mapsto \psi_{\rho_k,0}(x,y)$ is strictly concave on $Y$ (since it is the sum of a concave quadratic $-\frac{1}{2}(y-e)^2$, a linear term $\frac{1}{2}x^\top y$, and a concave penalty $-\frac{\rho_k}{2}[e^\top y - \|x\|^2]_+^2$). Hence the inner maximization admits a unique maximizer, denoted by $y_{\rho_k}^*(x)$. For points on the boundary of $X, Y$, we can easily check that only $(-\frac{3}{4}e, \frac{10+18\rho_k}{16+32\rho_k}e)$ is the first-order minimax point.

Suppose $(x,y)$ lies in the interior of $X \times Y$. Then the normal cones vanish and the first-order optimality conditions for the minimax point reduce to

$$\begin{cases} 2(\frac{\|x\|^2}{2} - 1)x + \frac{1}{2}y + 2\rho_k tx = 0, \\ -(y-e) + \frac{1}{2}x - \rho_k te = 0, \\ t = [e^\top y - \|x\|^2]_+. \end{cases} \tag{67}$$

We show that (67) admits in the interior of $X \times Y$ when $\rho_k$ is sufficiently large.

*Case 1: $t = 0$ (i.e., $e^\top y \leq \|x\|^2$).* Then (67) becomes

$$2(\frac{\|x\|^2}{2} - 1)x + \frac{1}{2}y = 0, \qquad -(y-e) + \frac{1}{2}x = 0 \iff y = e + \frac{1}{2}x.$$

On the other hand, $2 + \frac{1}{2}e^\top x = e^\top y \leq \|x\|^2$, which is a contradiction since $x \in X = [-\frac{3}{4}, \frac{5}{4}]^2$. Hence no interior solution exists in this case.

*Case 2: $t > 0$ (i.e., $e^\top y > \|x\|^2$).* Then $t = e^\top y - \|x\|^2$ and the second equation in (67) gives

$$-(y-e) + \frac{1}{2}x - \rho_k(e^\top y - \|x\|^2)e = 0 \implies (1 + 2\rho_k)e^\top y = 2 + \frac{e^\top x}{2} + 2\rho_k\|x\|^2.$$

Note that $-y + \frac{1}{2}x = \rho_k(e^\top y - \|x\|^2 - 1)e$, which implies that $y = \frac{1}{2}x + ae$ with a constant $a$. Combining this with the previous equality, we have $a = \frac{2+2\rho_k\|x\|^2 - \rho_k e^\top x}{2(1+2\rho_k)}$ and $y = \frac{2+2\rho_k\|x\|^2 - \rho_k e^\top x}{2(1+2\rho_k)}e + \frac{x}{2}$. As a result,

$$t = e^\top y - \|x\|^2 = \frac{4 + e^\top x - 2\|x\|^2}{2(1+2\rho_k)} > 0.$$

Substituting and the above expression of $y$ and $t$ into the first equation of (67) yields the cubic equation

$$2(\frac{\|x\|^2}{2} - 1)x + \frac{2+2\rho_k\|x\|^2 - \rho_k e^\top x}{4(1+2\rho_k)}e + \frac{x}{4} + \rho_k\frac{4 + e^\top x - 2\|x\|^2}{1+2\rho_k}x$$

$$= \frac{1}{1+2\rho_k}\|x\|^2 x + (-\frac{7}{4} + \frac{4\rho_k + \rho_k e^\top x}{1+2\rho_k})x + \frac{2+2\rho_k\|x\|^2 - \rho_k e^\top x}{4(1+2\rho_k)}e \tag{68}$$

$$= 0,$$

which implies $x = \bar{a}e$ for some $\bar{a}$. As a result,

$$\frac{2\bar{a}^3}{1+2\rho_k}e + (-\frac{7}{4}\bar{a} + \bar{a}\frac{4\rho_k + 2\rho_k\bar{a}}{1+2\rho_k})e + \frac{2 + 4\rho_k\bar{a}^2 - 2\rho_k\bar{a}}{4(1+2\rho_k)}e = 0,$$

Equivalently,

$$p_{\rho_k}(\bar{a}) := 8\bar{a}^3 - 7(1+2\rho_k)\bar{a} + (16\rho_k + 8\rho_k\bar{a})\bar{a} + 2 + 4\rho_k\bar{a}^2 - 2\rho_k\bar{a} = 8\bar{a}^3 + 12\rho_k\bar{a}^2 + -7\bar{a} + 2 = 0$$

Let

$$\rho_0 := \sup_{\bar{a} \in [-\frac{3}{4}, \frac{5}{4}], \bar{a} \neq 0} \frac{7\bar{a} - 8\bar{a}^3 - 2}{12\bar{a}^2} \in (0, 1),$$

By construction, for any $\rho_k > \rho_0$ we have $p_{\rho_k}(\bar{a}) \geq 0$ for all $\bar{a} \in (-\frac{3}{4}, \frac{5}{4})$, and moreover $p_{\rho_k}(\bar{a}) > 0$ for all $a \in (-\frac{3}{4}, \frac{5}{4})$. Therefore, (68) admits no root in $(-\frac{3}{4}, \frac{5}{4})^2$ when $\rho_k > \rho_0$, and hence (67) has no interior solution in the case $t > 0$.

Combining the two cases, we conclude that for any $\rho_k > \rho_{k0}$ there is no first-order minimax point in the interior of $X \times Y$. This completes the proof. □

