# OpenReview forum: "A Single-Loop Stochastic Gradient Algorithm for Minimax Optimization with Nonlinear Coupled Constraints"
_ICML.cc/2026/Conference — Submitted to ICML 2026_

### Official Review · Reviewer_UX9Y · 2026-03-05

**Soundness:** 1
**Presentation:** 2
**Significance:** 3
**Originality:** 3
**Overall Recommendation:** 4
**Confidence:** 3

**Summary:**

The paper considers saddle-point problems with functional constraints in the stochastic setting. A quadratic-penalty-based approach is considered, and consistency between the original and penalized problem is established. A primal-dual algorithm is proposed, and its convergence rates are obtained. The work of the algorithm is demonstrated on synthetic and ML problems with superior performance.

**Compliance With Llm Reviewing Policy:**

Affirmed.

**Final Justification:**

I'm satisfied with the last response by the authors and am increasing the score to 4. My only doubt is that there were some typos in the proofs, and it is hard to check whether they can be fixed without affecting the final result. Yet, the final results look reasonable based on the correspondence principle.

**Key Questions For Authors:**

1. How does the algorithm work compared to the min-min-max reformulation in other experiments?
2. Maybe I overlooked something in the proofs, and the authors can clarify the proof steps I mentioned above?
3. Is it possible to obtain rates in terms of the feasibility $c(x,y)\leq0$?

**Limitations:**

I believe the authors should discuss more explicitly the question of feasibility w.r.t. $c(x,y)\leq0$.

**Strengths And Weaknesses:**

# Soundness

I have several concerns, mainly regarding the theoretical results.
1. The proof of the main theoretical result in Proposition 3.4 seems to be incorrect. Namely, the inequality in line 1777 has at least three problems. a. The two terms in the l.h.s., with $\|e_x^k\|^2$ and with $\|e_x^{k-1}\|^2$ are somehow combined into one term with just $\|e_x^{k-1}\|^2$. This is also amplified by the inconsistency of these $e$-terms in (10) and (28) and after the first inequality in (59). b. The $2c_k\eta_k^2\delta^2$ term in line 1774 abruptly changes to $2c_k\eta_{k+1}^2\delta^2$ term in line 1790. c. The term $-a_k\alpha_k/2$ in line 1781 should be $-a_k\alpha_k\bar{L}_k^2/2$. Further, the second asymptotic in line 1800 should be $O(k^t)$ rather than $O(k^{5t))$. Finally, it seems that many terms coming from Lemma D.9 in the r.h.s. of line 1834 are missing.
2. The paragraph after (20) was not clear to me, and I think more details should be provided to check the proof.
3. I think the paper is missing the rate for the feasibility gap w.r.t. the constraint $c(x,y)\leq 0$. On the one hand, the authors prove the asymptotic feasibility. On the other hand, a particular iterate may be arbitrarily bad w.r.t. the constraint $c(x,y)\leq 0$.
4. I can't agree with the authors' claim about nonasymptotic rates since the result holds "for sufficiently large $k$" which comes from the asymptotics in line 1800.
5. Overall, the numerical experiments are quite impressive. Yet, sometimes the baselines seem not to be well motivated. For example, why EG is used in Fairness-aware Classification, given that there are many more recent algorithms for such problems? Further, in the constrained GAN experiments, I believe it is not fair to fix such a small penalty parameter for GAN-C. Finally, it is not clear why methods based on min-min-max reformulations are compared to only in a small synthetic example.


# Presentation

Overall, the presentation of the paper is quite good. The majority of the text is clearly written and easy to follow. As already mentioned, my main concern is about the proofs. Also, some parts of the appendix should be polished in terms of English grammar and style, and regarding the notation, typos, and explanations. For example, what is $(x^*,y^\ast)$ in Lemma C.4? The notation $\delta$ is used for variance and for ball radius in Def. 2.8 and that was confusing to me. Line 1003: close enough to what? Line 1046: Proposition->Lemma. Line 1638: I guess the second expression should be $D_x^{k-1}$. Line 1657: The sentence reads strangely. Line 1673: I think $d$ should be $D$ in both places. In line 1755, I could not see how this inequality can be guaranteed by choosing the ratio between $\beta_0$ and $\sigma_0$ -- I think it should actually be product rather than ratio, and there are also other constants, such as $\rho_k$ involved in $\bar{L}_k$.


# Significance

Despite the main idea being classical, I find the proposed algorithm interesting, and the whole paper tries to answer a timely question since such formulations appear in many ML problems. The numerical experiments show good performance of the algorithm, and I think it can be interesting for practitioners. One can think of many open questions in this direction, and I think the paper can motivate many follow-up works.


# Originality

The penalty approach is classical, but proving that it works in the setting of stochastic saddle-point problems requires some work, which I find valuable. I'm actually quite surprised that it has not been made so far. What is also a bit surprising to me is that it is sufficient to make just one step for the approximate solution of the inner problem, and the whole algorithm is single-loop.


# Summary

Overall, I think the paper considers an interesting problem well-motivated by ML applications. The proposed algorithm is interesting and performs well in the experiments. My main concern is about the theoretical results, which do not seem to be correct.

---

> ### Author Rebuttal · Authors · 2026-03-31
>
> We sincerely thank the reviewer for the insightful comments. We are grateful for the opportunity to clarify our work and address the concerns raised.
>
> Below, we address the specific concerns raised.
> ## 1. On the theoretical result (Regarding Soundness 1 and Question 2)
>
> We deeply appreciate the reviewer's insightful suggestions in helping us ensure the theoretical appendix is airtight. We confirm that the theoretical conclusions **remain valid**. The index typos identified by the reviewer do **not affect the final results**. We have corrected the derivations in the revision as follows:
>
> - **Resolution of the $e$-terms and Lemma D.9 (Point a):** The primary modifications center on the variance terms. By correcting the terms in Eq. (59) and combining them with line 1830, term (iii) in Eq. (61) takes the form:
>
>   $$(-2c_k\eta_k + c_k\eta_k^2)||e_x^{k-1}||^2 + \frac{9}{8}a_k\alpha_k\mathbb{E}[||e_x^k||^2\mid\mathcal{F}_k].$$
>
>   Substituting the upper bound for $\mathbb{E}[||e_x^k||^2\mid\mathcal{F_k}]$ from Lemma D.9 extracts the $||e_x^{k-1}||^2$ component and introduces expanded terms involving $||y^k-y_k^*(x^k)||^2$ and $||x^k-x^{k-1}||^2$. Under our step-size schedule, using the exact same bounding techniques applied in lines 1854, 1876, and 1894, these newly introduced terms are completely absorbed by the existing terms in Eq. (61). For instance, examining the coefficient of $||e_x^{k-1}||^2$, since $c_{k}\eta_{k}=\mathcal{O}(k^{-7t-s})$, $c_{k}\eta_{k}^2=\mathcal{O}(k^{-7t-2s})$, and $a_k\alpha_k=\mathcal{O}(k^{-8t-s})$, all positive lower-order terms are effectively absorbed by the dominant term $-2c_{k}\eta_{k}$, ensuring a strictly negative overall coefficient for $||e_x^{k-1}||^2$.
>
> - **Correction of localized typos (Points b, c, & line 1800):** We will fix the mismatched subscript to $2c_k\eta_k^2\delta^2$, restore the missing multiplier to $-a_k\alpha_k\bar{L}_k^2/2$, and correct the asymptotic rate typo to $\mathcal{O}(k^{t})$.
>
> ## 2. On the proof of the paragraph after (20) (Regarding Soundness 2)
>
> The steps after Eq. (20) aim to find a direction $d_y$  such that $\langle \nabla_y c_i(x^\*,y^\*), d_y\rangle<0$ for all $i\in\mathcal{A}$, which  is equivalent to MFCQ. For convex constraint systems, the Slater condition is a sufficient condition for MFCQ at any feasible point (see, e.g., Proposition 3.3.9 in [1]). Since $c_i(x^\*, \cdot)$ is convex in $y$ (Assumption 2.1) and satisfies the consistent Slater condition, MFCQ holds for the sliced system $c(x^\*,y)\le 0$ at $y^\*$. We will add this theoretical justification and citation to the revised appendix to make the subsequent algebraic steps transparent.
>
> [1] Bertsekas D. P. Nonlinear Programming. Athena Scientific, 1999.
>
> ## 3. On the feasibility rate (Regarding Soundness 3 and Question 3)
>
> We can indeed **establish a feasibility rate** of $\mathcal{O}(K^{-t})$ derived from the boundedness of $\lambda_k$ defined in line 2037. Due to space constraints, please refer to our reply to Reviewer WUnk for the detailed derivation.
>
> ## 4. On the claim of non-asymptotic rates  (Regarding Soundness 4)
>
> In our context, "for sufficiently large $k$" was intended to denote a finite, explicitly computable threshold $k_0$ after which our deterministic bounds take effect. Because this constant does not alter the order of the convergence rate, we omitted its explicit calculation for brevity, taking inspiration from similar simplified presentations in the literature (e.g., Theorem 4.9 [2]). We will  specify this in the revision.
>
> [2]  Nocedal, J. and Wright, S. J. Numerical optimization. Springer, 2006.
>
> ## 5. On the numerical experiments (Regarding Soundness 5 and Question 1)
>
> **Fairness Baselines:** We clarify that EG served as a classic baseline, and our comparison also included LEN, a **recent** state-of-the-art algorithm.
>
> **GAN-C Penalty and Min-Min-Max Methods:** We have evaluated GAN-C with larger penalties ($\rho \in \\{10, 20\\}$) and MGD on the high-dimensional AFHQ-v2 task. As reported below, increasing $\rho$ degrades the FID score, while MGD struggles with stable convergence across different dual step sizes ($\gamma$).
>
> |       Methods       |   FID    |   IS   |
> | :-----------------: | :------: | :----: |
> |   GAN-C ($\rho=5$)   | $26.46$  | $6.62$ |
> |  GAN-C ($\rho=10$)   | $27.96$  | $6.62$ |
> |  GAN-C ($\rho=20$)   | $28.05$  | $6.76$ |
> |  MGD ($\gamma=0.1$)  | $192.41$ | $3.61$ |
> | MGD ($\gamma=0.01$)  | $32.81$  | $6.23$ |
> | MGD ($\gamma=0.001$) | $35.12$  | $5.96$ |
> |  **SPACO** ($\rho_0=5$)  | $\mathbf{24.44}$  | $\mathbf{6.89}$ |
>
> ## 6. On presentation
>
> We will thoroughly address all notational concerns in the revision. Furthermore, we will clarify that in Lemma C.4 that $(x^\*,y^\*)$ denotes any feasible point, and explicitly add "close enough to $(x^\*,y^\*)$" in Line 1003 to ensure local geometric rigor.

---

> > ### Author Rebuttal · Reviewer_UX9Y · 2026-04-01
> >
> > I would like to thank the authors for their detailed rebuttals. Most of my concerns are addressed, but two still remain. (I use the same numbering as in the rebuttal)
> >
> > 1. The answer provides some hints on the proof corrections, but, unfortunately, it is hard for me to check the correctness of the new proof in full detail. Maybe  the authors may explain how their rates behave in some limiting cases for which the rates are already known. At least to check that the obtained rates satisfy the correspondence principle. For example, if there are no coupled constraints and $f$ is strongly concave in $y$, we have a mean-squared smooth stochastic optimization w.r.t. $x$, which, as far as I know, leads to sample complexity $1/\varepsilon^3$ to drive the expected norm of the gradient mapping below $\varepsilon$. What would be the corresponding bound in the proposed algorithm? What happens when there is no strong concavity in $y$? Then, introducing the regularization with parameter $\varepsilon$ in $y$, we reduce to the previous setting. But, now the complexity has additional factor depending on $\varepsilon$. What would be existing and new bound in this case? What happens if there is no stochasticity -- how the bounds compare to [1] Lu, Z. and Mei, S. mentioned by Reviewer WUnk? What is also confusing me in the final bounds is that there is no trade-off in $t$. In order to optimize the rates, the bound in Th. 3.5 suggests that $t$ should be taken as small as possible. I believe there should be price for that. Otherwise, why not just take $t=0$?
> >
> > 4. Unfortunately, I'm not so sure that $k_0$ can be explicitly computed since it will depend on the problem parameters such as $L_f,L_c,M$.
> >
> > To sum up, the authors resolved most of my concerns, but some points remain. Furthermore, I would qualify the changes in the paper as a major revision, and the updated manuscript should undergo one more review cycle. I've increased my score to 3, but the latter point makes me hesitate to further increase the score.

---

> > > ### Author Response · Authors · 2026-04-02
> > >
> > > We sincerely thank the reviewer for the constructive follow-up and for increasing the score. We provide the following clarifications to address the remaining concerns.
> > >
> > > ## 1. Correspondence Principle and Limiting Cases
> > >
> > > When reduced to certain simpler cases (such as the nonconvex–strongly-concave setting without coupled constraints), our rates indeed **recover classical complexities**. However, for some other scenarios, our general bound does not trivially reduce to the tightest problem-specific rates due to the necessary theoretical overhead.
> > >
> > > - **Nonconvex--Strongly Concave (NC-SC) without coupled constraints:** In this case, we set $t=0$ and $\rho_0=\sigma_0=0$, as penalty and regularization terms are no longer required. By choosing $s=1/2$, Theorem 3.5 yields a rate of $\min_{0\le k\le K} \{ \mathbb{E}[||\mathcal{G}_k(x^k)||^2] \} = \mathcal{O}(K^{-1/2})$. This corresponds to a complexity of $\mathcal{O}(\epsilon^{-4})$ (under the gradient norm measure), which is consistent with standard stochastic NC-SC optimization results (e.g., [1]).
> > > - **Nonconvex--Concave (NC-C) without coupled constraints:** In this case, we can take $t=1/15$ (to measure the decrease of $\sigma_k$) and $\rho_0=0$. Consequently, by taking $s=1/3$, the convergence rate in Theorem 3.5 is then $\min_{0\le k\le K} \{ \mathbb{E}[||\mathcal{G}_k(x^k)||^2] \}= \mathcal{O}(K^{-2/15})$, which results in a complexity of $\mathcal{O}(\epsilon^{-15})$. While this appears slower than the optimal rate for standard stochastic NC-C optimization, this discrepancy is expected; it is a natural consequence of our unified analytical framework, which encompasses a substantial amount of theoretical machinery dedicated to managing coupled constraints.
> > > - **Absence of Stochasticity (Deterministic Case):** As noted in our response to Reviewer WUnk, SPACO achieves $\mathcal{O}(\epsilon^{-15})$ with $t=1/15, s=1/3$. The deterministic complexity in [2] is $\tilde{\mathcal{O}}(\epsilon^{-4})$. However, since our algorithm and step-size schedules are strictly constructed for the stochastic regime, a direct comparison by simply "removing noise" is difficult, as the schedules are not optimized for the deterministic setting.
> > > - **The Trade-off in $t$:** Actually, $t$ cannot be arbitrarily small. We require $\rho_k \to \infty$ to ensure feasibility, with the feasibility error explicitly bounded by $\mathcal{O}(K^{-t})$ (please see our response to Reviewer WUnk). To balance stationarity and feasibility, we find $t=1/15, s=1/3$ to be a theoretically optimal choice. Empirically, as shown in our experiments, the algorithm is also robust to a range of $t$ and $s$ values.
> > >
> > >
> > > ## 2. On the Explicit Computation of $k_0$
> > >
> > > We clarify that $k_0$ can indeed be represented as a finite constant involving problem parameters and initial step sizes. For instance, to satisfy the descent condition $\frac{a_k}{8\alpha_k} > \frac{a_k L_{\varphi_k}}{2}$, it suffices to ensure:
> > > $$
> > > 16\sigma_0 \alpha_0 k^{3t+s} > (L_f + \rho_0 M L_c + \rho_0 M^2 + 2\sigma_0)^2 > (L_f k^{-t} + \rho_0 M L_c + \rho_0 M^2 + 2\sigma_0 k^{-2t})(L_f k^{-t} + \rho_0 M L_c + \rho_0 M^2).
> > > $$
> > > By defining $k_0 = \lceil (\frac{(L_f + \rho_0 M L_c + \rho_0 M^2 +2\sigma_0)^2}{16 \sigma_0 \alpha_0})^{1/(3t+s)} \rceil$, the condition holds for all $k > k_0$. Similar finite thresholds can be derived for all other "sufficiently large" steps in our proof. While these parameters ($L_f, L_c, M$) are often difficult to compute precisely in practice, we believe this type of non-asymptotic analysis is an effective convention in optimization theory. Importantly, **$k_0$ is a constant independent of** the target precision $\epsilon$, meaning it does not alter the fundamental convergence order of our algorithm.
> > >
> > >
> > > [1] Lin, T., Jin, C., and Jordan, M. I. On gradient descent ascent for nonconvex-concave minimax problems. In International Conference on Machine Learning, 2020.
> > >
> > > [2] Lu, Z. and Mei, S. A first-order augmented Lagrangian method for constrained minimax optimization. Mathematical Programming, 213:1063--1104, 2025.

---

### Official Review · Reviewer_WUnk · 2026-03-06

**Soundness:** 3
**Presentation:** 3
**Significance:** 2
**Originality:** 3
**Overall Recommendation:** 3
**Confidence:** 2

**Summary:**

The paper proposes a penalty-based smooth approximation for MCC by applying a quadratic penalty and adding a regularizer in $y$, which yields a smooth surrogate $\phi_{\rho,\sigma}(x)$. The framework connects to the original problem: under GPLCQ, accumulation points of stationary points of the smooth approximations are KKT points of the original MCC. Section 4 also gives a useful landscape intuition: in the toy example, the min-min-max reformulation has a spurious stationary point, while the penalty approximation follows a unique stationary path to the true solution and helps improve the landscape.

Section 3 develops SPACO, a single-loop stochastic method that tracks the inner maximizer with one ascent step and updates $x$ using an inexact gradient estimator, thereby avoiding nested inner solves. The analysis uses a Lyapunov function to control outer progress, inner tracking error, and estimator error. Theorem 3.5 gives non-asymptotic best-iterate bounds for the generalized gradient residual and the inner tracking error, while Theorem 3.6 gives almost-sure existence of a convergent subsequence whose limit is a KKT point under GPLCQ.

The experiments cover synthetic stochastic MCC problems, fairness-aware classification on Adult and CelebA, and constrained GAN training on CIFAR-10 and AFHQ-v2. SPACO is compared with benchmarks, and the reported results are generally favorable: faster error reduction on synthetic tasks and better fairness metrics.

**Compliance With Llm Reviewing Policy:**

Affirmed.

**Final Justification:**

The paper proposes a technically interesting penalty-based smooth approximation and a genuinely single-loop stochastic method for minimax optimization with nonlinear coupled constraints, together with meaningful convergence analysis.

The rebuttal and follow-up clarification fully addressed my questions about the comparison with prior work and the interpretation of the KKT convergence logic. However, these clarifications mainly sharpen the positioning of the work rather than materially changing my assessment of its contribution and overall strength relative to prior work. I therefore keep my score unchanged at 3.

**Key Questions For Authors:**

No.

**Limitations:**

Yes.

**Strengths And Weaknesses:**

Strengths:

1. A key strength is the penalty-based smooth approximation itself. It gives a clean surrogate with a principled connection back to MCC stationarity, and the Section 4 example makes the claimed landscape improvement by showing reduced attraction to spurious stationary points.

2. SPACO is truly single-loop and does not require exact inner solutions. The analysis is also substantive, providing explicit best-iterate rates together with an asymptotic stationarity result for the stochastic nonlinear-coupled setting.

Weakness:

1. The paper should benefit from a comparison of convergence results in Theorems 3.5 and 3.6 with the results in [1][2][3]. Although direct rate comparisons are difficult given differences in assumptions on problems, but some discussion would help readers judge the contributions of Theorems 3.5 and 3.6 more precisely.

2. The current guarantees also appear weaker than [1] if one compares the final convergence guarantees directly. Theorem 3.5 gives a best-iterate bound, and Theorem 3.6 proves subsequence convergence to a KKT point under GPLCQ, whereas [1] gives an $\epsilon$-KKT guarantee for the original constrained minimax problem with explicit complexity.

[1] Lu, Z. and Mei, S. A first-order augmented Lagrangian method for constrained minimax optimization. Mathematical Programming, 213:1063--1104, 2025.

[2] Tsaknakis, I., Hong, M., and Zhang, S. Minimax problems with coupled linear constraints: Computational complexity and duality. SIAM Journal on Optimization, 33(4):2675--2702, 2023.

[3] Hu, X., Toh, K.-C., Wang, S., and Xiao, N. A minimization approach for minimax optimization with coupled constraints. arXiv preprint arXiv:2408.17213, 2024.

---

> ### Author Rebuttal · Authors · 2026-03-31
>
> We appreciate the reviewer's thoughtful assessment. Below, we address the specific concerns.
>
> ## 1. On the comparison of convergence results (Regarding weakness 1)
>
> We appreciate this constructive suggestion. We will incorporate the following summary table into the revision.
>
> (Note: NC, SC, and C denote Non-Convex, Strongly-Convex/Strongly-Concave, and Convex/Concave, respectively. "Non-Asym" and "Asym" indicate whether the method provides a convergence rate for a specific stationarity measure, and the convergence of the iterate sequence to a limit point, respectively.)
>
> |Methods|Objective|Constraint|Loop Structure|Stochastic|Convergence|
> |:-:|:-:|:-:|:-:|:-:|:-:|
> |[1]|NC-C|Nonlinear|Triple-Loop|No|Non-Asym|
> |[2]|SC-SC|Linear|Double-Loop|No|Non-Asym|
> |[3]|NC-SC|Nonlinear|Single-Loop|No|Asym|
> |**Ours**|**NC-C**|**Nonlinear**|**Single-Loop**|**Yes**|**Non-Asym & Asym**|
>
> As highlighted, our primary contribution is an easy-to-implement **single-loop** algorithm for **stochastic** minimax optimization with **nonlinear** coupled constraints. Crucially, our analysis successfully characterizes **both the convergence rate and the limit points of the iterate sequence**. Given the compound challenges of stochasticity and nonlinear constraints, we currently focus on algorithmic feasibility. Developing accelerated variants and tighter complexity bounds remains an important direction for future research.
>
> ## 2. On the the convergence guarantee (Regarding weakness 2)
>
> While we respect the explicit $\epsilon$-KKT convergence guarantee in [1], we clarify that convergence guarantees are **structurally incomparable** due to their differing paradigms:
>
> - **Shared "best-iterate" bound:** Both frameworks rely on the same best-iterate bound technique. For instance,  Lemma 3 in [1] establishes its rate by bounding $\min_{0 \le k \le K} ||x^{k+1} - x^k|| = \mathcal{O}(1/K)$ and Theorem 2 outputs an $\epsilon$-primal-dual stationary point in at most $T+1$ outer iterations—which is **mathematically equivalent** to this exact bounding logic.
> - **Deterministic vs. Stochastic:** Deterministic methods in [1] use exact gradients for stopping criterion. In our stochastic setting, evaluating per-iteration KKT residuals is **computationally prohibitive**. The expected minimum translates [1]'s logic into the stochastic regime, thereby yielding a different convergence guarantee.
>
> Furthermore, by utilizing the non-asymptotic rates for the generalized gradient $\mathcal{G}_k(x^k)$ and tracking error $||y^k - y_k^*(x^k)||^2$ from Theorem 3.5, we can **naturally extend** existing results to an $\epsilon$-KKT complexity.
>
> Defining the multiplier $\bar{\lambda}_k := \rho_k [c(x^k, y^k)]\_\+$ and the standard Lagrangian $L(x,y,\lambda)$, we can define the KKT-residual similar as [1]:
> $$
> R_k := \max\Big\\{ ||x^k-\mathcal{P}_X(x^k -\nabla_x L(x^k, y^k, \bar{\lambda}_k))||, ||y^k - \mathcal{P}_Y(y^k + \nabla_y L(x^k, y^k, \bar{\lambda}_k))||, ||[c(x^k, y^k)]\_+||, |\bar{\lambda}_k^\top c(x^k, y^k)|\Big\\}.
> $$
>
> Employing similar techniques used in proof of Theorem 3.6, we can establish that $\min_{0 \le k \le K} \mathbb{E}[R_k] = \mathcal{O}(K^{-\tau})$, where $\tau:=\min\\{\frac{s-3t}{2},\frac{1-8t-s}{2},t\\}>0$.
>
> - **Feasibility ($||[c(x^k, y^k)]_+||$):** Analogous to line 2009, we establish that the expected exact multiplier is bounded ($\mathbb{E}[||\lambda_k||^2] = \mathcal{O}(1)$, where $\lambda_k=\rho_k [c(x^k, y_k^\*(x^k))]_+$), which implies the feasibility gap of $y_k^\*(x^k)$ is $\mathcal{O}(K^{-t})$. Further, the empirical constraint violation is naturally bounded by the feasibility gap of $y_k^\*(x^k)$ and the tracking error, achieving a rate of $\mathcal{O}(K^{-t})\le\mathcal{O}(K^{-\tau})$.
> - **Complementary Slackness ($|\bar{\lambda}_k^\top c(x^k, y^k)|$):** The slackness error is exactly $||\bar{\lambda}_k||^2 / \rho_k$. Combining the bounded multipiliers and the decay of the tracking error yields the $\mathcal{O}(K^{-\tau})$ convergence rate.
> - **Primal & Dual Stationarity:** As the gradient of the standard Lagrangian function $L$ structurally aligns with the gradient of our penalty-based approximation $\psi_k$, both stationarity gaps are controlled by the generalized gradient bound and tracking error. Using bounding techniques analogous to line 1988, this achieves the $\mathcal{O}(K^{-\tau})$ rates.
>
> We will incorporate this KKT convergence rate and the specific proof into the revised manuscript to elevate our theoretical completeness.
>
> [1] Lu, Z. and Mei, S. A first-order augmented Lagrangian method for constrained minimax optimization. Mathematical Programming, 213:1063--1104, 2025.
>
> [2] Tsaknakis, I., Hong, M., and Zhang, S. Minimax problems with coupled linear constraints: Computational complexity and duality. SIAM Journal on Optimization, 33(4):2675--2702, 2023.
>
> [3] Hu, X., Toh, K.-C., Wang, S., and Xiao, N. A minimization approach for minimax optimization with coupled constraints. arXiv preprint arXiv:2408.17213, 2024.

---

> > ### Author Rebuttal · Reviewer_WUnk · 2026-04-04
> >
> > Thank you for the helpful response. I appreciate the clarification on how Theorems 3.5 and 3.6 relate to [1][2][3], as well as the explanation of the stochastic, nonlinear, single-loop setting.
> >
> > The rebuttal also clarifies the comparison with [1] and provides a sketched KKT-style convergence argument, which answers my questions.
> >
> > However, the stronger KKT-style rate discussed in the rebuttal seems to reflect additional work beyond the current paper.

---

> > > ### Author Response · Authors · 2026-04-04
> > >
> > > We thank the reviewer for the prompt feedback and for acknowledging that our rebuttal effectively addressed your questions regarding the comparisons with exiting work and the KKT convergence logic. Below, we address the remaining concern regarding the KKT rates.
> > >
> > > We wish to clarify that the KKT convergence rate is **not a stronger result** than the penalty-based stationarity already established in our manuscript. As discussed in Section 4, KKT points may admit spurious saddle points, which renders a convergence guarantee based solely on KKT conditions somewhat unreliable. In contrast, our current theoretical framework rigorously characterizes the effectiveness of the minimizers and stationary points of our penalty-based approximation, thereby demonstrating the inherent reliability of these penalty-based stationary points.
> > >
> > > Therefore, while incorporating the explicit KKT convergence rate is certainly beneficial and enriches the theoretical discussion, we believe it acts as a natural corollary rather than a new core contribution that exceeds the established scope of the current paper.

---

### Official Review · Reviewer_FPrL · 2026-03-11

**Soundness:** 3
**Presentation:** 3
**Significance:** 3
**Originality:** 3
**Overall Recommendation:** 3
**Confidence:** 4

**Summary:**

This paper introduces SPACO, a single-loop stochastic gradient algorithm for solving nonconvex-concave minimax problems with nonlinear convex coupled constraints (MCC). The method employs a penalty-based smooth approximation framework, incorporating quadratic penalties and regularization to handle coupled constraints. The authors establish non-asymptotic complexity results and asymptotic stationarity under a newly proposed Generalized uniform Polyak-Łojasiewicz Constraint Qualification.

**Compliance With Llm Reviewing Policy:**

Affirmed.

**Final Justification:**

I appreciate the author's detailed response. The author's reply has addressed some of my concerns, and I am considering raising my score to 3.

**Key Questions For Authors:**

Given the extreme decay rate of k^{-(6t+s)}, the iterates may simply "freeze" numerically. Can you provide evidence that the gradient residual \mathcal{G}_k actually vanishes rather than the iterates just stopping due to a near-zero step size?

**Limitations:**

This convergence theory is limited to non-convex/concave regions with convex constraints, which restricts its application in fully non-convex/non-concave deep learning tasks. Furthermore, at high rho_k values, the extreme ill-conditioned nature of the penalized surface may hinder the algorithm's ability to obtain high-precision solutions in practice.

**Strengths And Weaknesses:**

Strengths:
The proposed SPACO algorithm addresses a technically challenging and highly relevant problem in stochastic minimax optimization. This approach offers significant computational advantages over traditional nested-loop methods by eliminating the need for exact subproblem solutions at each iteration.
Weaknesses:
1. The authors choose a penalty method to avoid increasing the problem's dimensionality via multipliers. However, low dimensionality does not necessarily mean that the problem is easier to solve; the augmented Lagrange method is now a commonly used method for solving constrained problems.
2. The paper's method relies on a standard quadratic penalty term combined with aggressive step decay. This approach uses decreasing step size to suppress numerical instability caused by increasing penalty term, and is a well-known heuristic in the optimization field, though its conceptual innovation is limited.

---

> ### Author Rebuttal · Authors · 2026-03-31
>
> We sincerely thank the reviewer for the valuable feedback. We address the specific concerns regarding the weaknesses and the key question below:
>
> ## 1. On the choice of Penalty Method (Regarding weakness 1)
>
> As highlighted in the abstract, the core contribution of our work is the development of an efficient **single-loop** algorithm specifically designed for **stochastic** minimax optimization with **nonlinear** coupled constraints.
>
> We respectfully clarify that our penalty-based framework is not merely a workaround for "low dimensionality". As illustrated in lines 92--95, we adopt the penalty method because it is a classical optimization approach that remains largely underexplored for MCC problems. Furthermore, Section 4 highlights a potential strength of the penalty method: its ability to **escape spurious stationary points**. While Augmented Lagrangian Method (ALM) is a powerful approach for classical constrained minimization, standard ALM inherently relies on nested loops and full gradients for its stopping criteria. Consequently, adapting ALM for stochastic MCC presents unique computational challenges under stochastic setting and remains an important direction for future research.
>
> ## 2. On "limited conceptual innovation" and heuristic step decay (Regarding weakness 2)
>
> While quadratic penalties and step-size decay are classical tools, integrating them into a rigorous **single-loop** stochastic minimax framework is **non-trivial**. Our decreasing step-size schedule is **not a mere heuristic** to suppress instability; instead, it is strictly and mathematically coupled with the penalty growth rate and momentum-based variance reduction.
>
> Specifically, Proposition 3.4 and Theorem 3.5 provide a rigorously quantified dynamic system, proving exactly how these schedules must coordinate to guarantee **convergence to true stationary points**.
>
> ## 3. On the potential "freezing" of iterates (Regarding the key question)
>
> We clarify that the algorithm does **not numerically "freeze"**. Theoretically, a premature freeze only occurs if the sum of the step sizes is finite. Our schedule explicitly requires $6t+s < 1$, ensuring $\sum \alpha_k = \infty$, which guarantees infinite travel capacity.
>
> Empirically, we additionally evaluated the gradient residual $\min_{i \leq k}||\mathcal{G}_i(x^i)||^2$ and stepsize dynamics on the Synthetic Example. As the table below demonstrates, the step sizes stabilize smoothly, allowing the residual to actively converge to near-zero in the noiseless setting. Under stochastic noise, while the gradient residual naturally decays more slowly, the relative error strictly reaches the exact same high-accuracy level. This explicitly refutes the premature stop hypothesis.
>
> | Iteration | Stepsize $\alpha_k$ | Gradient Residual (Noiseless) | Gradient Residual (Noisy) | Relative Error (Noiseless) | Relative Error (Noisy) |
> | :--: | :------: | :---------------------------: | :-----------------------: | :------------------------: | :--------------------: |
> |  0   |   1e-2   |            4.00e+2            |          4.00e+2          |          1.00e+0           |        1.00e+0         |
> | 1e3  |   3e-4   |            1.76e+1            |          1.74e+1          |          5.97e-2           |        5.88e-2         |
> | 1e4  |   1e-4   |            2.59e-2            |          3.13e-2          |          2.41e-4           |        2.42e-4         |
> | 4e4  |   5e-5   |            4.37e-6            |          2.11e-3          |          2.10e-4           |        2.11e-4         |
> | 1e5  |   3e-5   |            2.21e-9            |          1.23e-3          |          1.92e-4           |        1.92e-4         |
>
> ## 4. On the limitations (Regarding the limitaion)
>
> - **NC-NC limitation:** As noted in lines 46--52, stochastic NC-C problems with **nonlinear coupled** constraints **remain virtually unexplored**; **SPACO explicitly addresses this gap.** While fully NC-NC settings are important for deep learning, they lie beyond our current scope. As commented by reviewer Reviewer UX9Y in Significance,  we hope our framework can motivate future extensions into such regimes.
>
> - **ill-conditioning at high $\rho_k$:** Our approach **avoids** direct impacts from ill-conditioning through a coordinated dynamic schedule. Although penalty methods can theoretically face ill-conditioning as $\rho \to \infty$, we mitigate this by synchronizing penalty growth with decaying step sizes. This strategy effectively regulates the landscape curvature, as supported by the rigorous **convergence guarantees in Proposition 3.4 and Theorem 3.5**. Empirically, SPACO maintains high-precision performance across all tasks, demonstrating that ill-conditioning remains well-managed in our practical implementation.

---

> > ### Author Rebuttal · Reviewer_FPrL · 2026-04-06
> >
> > I appreciate the author's detailed response. The author's reply has addressed some of my concerns, and I am considering raising my score to 3.

---

> > > ### Author Response · Authors · 2026-04-07
> > >
> > > Thank you for reviewing our rebuttal and considering raising your score. We are glad that our response has addressed your concerns. For any aspects of our work requiring further clarification, we are very willing to provide additional details.

---

### Official Review · Reviewer_5ycb · 2026-03-11

**Soundness:** 3
**Presentation:** 4
**Significance:** 3
**Originality:** 2
**Overall Recommendation:** 3
**Confidence:** 5

**Summary:**

This paper studies a nonconvex–concave stochastic minimax optimization problem with nonlinear coupled convex constraints. Instead of relying on duality-based techniques, the authors propose a penalty-based regularized framework to handle the presence of constraints. To solve the resulting problem, they develop a single-loop stochastic gradient–based algorithm and establish non-asymptotic convergence rates. Under an additional constraint qualification assumption, the authors further characterize the optimality of the accumulation points of the iterates generated by the algorithm in the asymptotic sense. The paper also presents a synthetic example highlighting the benefits of the penalty-based approach compared to the aforementioned duality-based approach. Finally, two applications from deep learning are provided to further demonstrate the utility of the proposed framework and algorithm.

**Compliance With Llm Reviewing Policy:**

Affirmed.

**Final Justification:**

Thank the authors for all responses. However, my main concern on showing non-asymptotic convergence of the algorithm for the original problem was not addressed in a convincing way. I maintain my assessment.

**Key Questions For Authors:**

1. Is there a specific reason for the non utilisation of variance reduction in developing the gradient estimator for y?

2. GPŁCQ is a key assumption for asymptotic stationarity. Could the authors provide guidance on how one might verify or ensure GPŁCQ holds in practical ML tasks?

3. Could the authors clarify the relationship between the surrogate-based stationarity measure and the KKT conditions of the original problem, and show complexity result to achieve a near-KKT point of the original problem?

**Limitations:**

1. Theoretical assumptions (e.g. GPŁCQ) appear to be somewhat restrictive and verifying them could be challenging.

2. Limited interpretability of the results since the relationship between the surrogate-based stationarity measure and the KKT conditions of the original problem is not established.

**Strengths And Weaknesses:**

The submission appears sound and is generally easy to follow. The problem formulation, proposed framework, and algorithmic development are presented in a structured manner, and it is relatively straightforward to understand the motivation and contributions of the work.

While the paper considers a nontrivial extension of standard minimax optimization, the problem formulation is specialized. However, the demonstrated applications in fairness-aware classification and constrained GANs show that the framework is relevant to practical ML problems.

The surrogate function used to develop the algorithm is strongly concave in y, rendering the resulting objective nonconvex-strongly concave. There already exist single-loop algorithms with convergence guarantees for such problems. Additionally, the stationarity measure utilised is based on the penalty-based surrogate rather than the one based on the original constrained problem (i.e. KKT conditions). Since the authors do not establish a relationship between the two, the interpretation of the convergence results is weakened.

Also, one possible improvement would be to include a comparative summary table in the related work section that highlights key differences between the proposed approach and prior methods (e.g., assumptions, problem settings, convergence guarantees). Such a table would allow for a better comparison to existing works.

---

> ### Author Rebuttal · Authors · 2026-03-31
>
> We thank the reviewer for the constructive feedback. We address the specific comments below:
>
> ## 1. On the dynamic nature of the surrogate function (Regarding the weakness on existing single-loop algorithms)
>
> We clarify that our proposed stochastic algorithm **directly** solves **stochastic** minimax optimization with **nonlinear coupled constraints** rather than a static nonconvex-strongly-concave objective. This inherent difference necessitates our innovative dynamic design.
>
> As formulated in Section 3.2, our algorithm handles this structure by integrating an **iteration-varying penalty and regularization** scheme. While existing single-loop methods efficiently solve fixed nonconvex-strongly-concave problems, our penalty-based approximation $\varphi_{\rho_k, \sigma_k}(x,y)$ evolves dynamically across iterations. As illustrated in r.h.s. of lines 220--224, to guarantee convergence, the penalty parameter must increase to infinity while the regularization vanishes. This evolution causes the subproblem's geometry to progressively deteriorate, rendering existing algorithms inapplicable. Consequently, a core contribution of our work is the strictly coordinated **dynamic parameter schedule (Theorem 3.5)**, which successfully manages this dynamic evolution under stochastic noise.
>
> ## 2. On the comparative table and relationship with KKT conditions (Regarding Weakness 1, Key Question 3, and Limitation 2)
>
> We appreciate this constructive suggestion. We will incorporate a **comparative summary table** in the revised manuscript to contrast our approach with existing methods regarding assumptions, settings, and convergence guarantees. Due to space constraints in this response, please refer to our detailed reply to **Reviewer WUnk** for the full table.
>
> Furthermore, we can rigorously **establish the relationship** between our surrogate-based stationarity measure and the original KKT conditions, thereby bridging the gap between our theoretical analysis and classical optimality criteria. A detailed derivation is provided in our response to **Reviewer WUnk**.
>
> ## 3. On the specific reason for not utilizing variance reduction for $y$ (regarding key question 1)
>
> The asymmetric design of our algorithm (applying variance reduction to $x$ but not $y$) reflects the **asymmetric geometric properties** of the objective function, intentionally **balancing theoretical sufficiency with a lightweight algorithm design**.
>
> - **Theoretical Sufficiency:** Due to the constrained and non-convex nature of the value function $\phi_{\rho_k,\sigma_k}(x)$, variance reduction is required to mitigate oscillatory behavior caused by stochastic noise (lines 252--257). In contrast, the inner $y$-subproblem is strictly $\sigma_k$-strongly concave, meaning standard stochastic projected gradient ascent is already sufficient to smoothly track the optimal trajectory without severe oscillation and seamlessly provide the rigorous convergence guarantees established in our paper.
> - **Lightweight Algorithm Design:** Introducing variance reduction for $y$ necessitates additional gradient computations and momentum accumulation, increasing memory and computational overhead. To keep the algorithm lightweight, and since we have achieved satisfactory theoretical and numerical results without it, we leave further exploration for future work.
>
> ## 4. On the constraint qualifications (Regarding key question 2 and limitation 1)
>
> - **Practical Verifiability:** **Lemma C.4 has established a practical guideline** : GPŁCQ holds for all feasible points in the interior of $X \times Y$ if the **consistent Slater condition** is met. It also holds for all interior points in scenarios the coupled constraints are **linear**. Furthermore, we only require the accumulation point $(\bar{x}, \bar{y})$ to satisfy the GPŁCQ; essentially, the constraint qualification only needs to hold locally around the stationary points.
> - **Restrictiveness of GPŁCQ:** GPŁCQ naturally extends the classical PŁ condition to coupled constraint minimax settings. Since PŁ-type conditions are **widely accepted ML assumptions** relied upon by foundational works [1, 2], our framework aligns perfectly with mainstream theory.
> - **Value Beyond GPŁCQ:** Even in complex scenarios where GPŁCQ might be difficult to strictly verify, our penalty-based framework **remains highly valuable**. As guaranteed by Theorem 2.6, the global optimal solutions of our smooth approximations asymptotically **converge** to the exact optimal solution of the original MCC problem, regardless of local constraint qualifications.
>
>
> [1] Karimi, H., Nutini, J., and Schmidt, M. Linear convergence of gradient and proximal-gradient methods under the Polyak-Łojasiewicz condition. In *Joint European conference on machine learning and knowledge discovery in databases*, pp. 795--811, 2016.
>
> [2] Wang, R., Dvijotham, K., and Manchester, I. R. Monotone, Bi-Lipschitz, and Polyak-Łojasiewicz networks. In *International Conference on Machine Learning*, 2024.

---

> > ### Author Rebuttal · Reviewer_5ycb · 2026-04-02
> >
> > I thank the authors for their response. However, my biggest concern on establishing complexity result for $\epsilon$-KKT point of the **original** problem remains. I checked the authors' response to **Reviewer WUnk**. The authors responded there that they could obtain $\epsilon$-KKT point of the original problem by a similar proof technique to Theorem 3.6. I doubt this. I checked the proof of Theorem 3.6, it is all about asymptotic convergence. Also, in several places, the authors need sufficiently large $k$. Based on my knowledge to show a non-asymptotic result, such arguments are not enough. I suspect that addressing this issue will require a stronger assumption. For example, the GPLCQ may be needed at all iterates, and this is of course non-checkable before knowing the iterates. Therefore, even if the authors can finally establish a complexity result to obtain $\epsilon$-KKT point, it must require substantial change but not like the authors mentioned "by similar techniques used in proof of Theorem 3.6".
> >
> > **Follow-up comments**
> > Thank the authors for further response. However, your arguments rely on subsequence convergence. This is exactly asymptotic convergence but not non-asymptotic convergence. I can understand the multiplier is bounded. But for a non-asymptotic analysis, you will need an explicit bound that holds for ALL iterations. Otherwise, you can only claim a result when the iteration $k$ is large enough.

---

> > > ### Author Response · Authors · 2026-04-03
> > >
> > > We sincerely thank the reviewer for their rigorous scrutiny and for raising this highly insightful follow-up point.
> > >
> > > ## 1. On the $\epsilon$-KKT complexity results
> > >
> > > We clarify that the $\epsilon$-KKT complexity can indeed be established using techniques similar to those in Theorem 3.6. Importantly, we do **not require** the GPŁCQ to hold for all iterates; we only need it to hold at the accumulation points of the sequence $\{(x^k,y^k)\}$.
> > >
> > > In this context, the GPŁCQ is exclusively used to guarantee the boundedness of the expected multipliers $\mathbb{E}[\lambda_k]$. We can rigorously prove this via contradiction: Let $\{(x^j, y^j)\}$ be the specific subsequence that achieves the complexity bound in Theorem 3.5. Suppose there exists a further subsequence $\{(x^{j_i}, y^{j_i})\}$ converging to an accumulation point $(x^\*, y^\*)$ such that $\mathbb{E}[\lambda_{j_i}]$ is unbounded. Because $(x^\*, y^\*)$ satisfies the GPŁCQ, the multipliers $\mathbb{E}[\lambda_{j_i}]$ must be bounded in a neighborhood around $(x^\*, y^\*)$. This directly contradicts the assumption. Thus, the multipliers evaluated at the complexity-achieving subsequence must be bounded.
> > >
> > > Furthermore, as discussed in our paper, the GPŁCQ can be verified through the consistent Slater condition. We believe this provides a practical and easily checkable criterion for real-world applications.

---

### Decision · Program_Chairs · 2026-04-30

**Decision:**

Reject

**Comment:**

This submission proposes a single-loop algorithm for the minimax optimization problem with nonlinearly coupled constraints. The reviewers acknowledge that this problem is technically challenging. However, the major concern regarding the establishment of a complexity result for an $\epsilon$-KKT point of the original problem remains unresolved after the rebuttal. Since establishing this theoretical result is important for the proposed algorithm, this submission is not ready for publication at this time.